# Transient APC/C inactivation by mTOR boosts glycolysis during cell cycle entry

Debasish Paul[1], Derek L. Bolhuis[2,3], Hualong Yan[1], Sudipto Das[4], Xia Xu[4], Christina C. Abbate[5], Lisa M. M. Jenkins[6], Michael J. Emanuele[7], Thorkell Andresson[4], Jing Huang[1], John G. Albeck[5], Nicholas G. Brown[3,7] & Steven D. Cappell[1✉]

Mammalian cells entering the cell cycle favour glycolysis to rapidly generate ATP and produce the biosynthetic intermediates that are required for rapid biomass accumulation[1]. Simultaneously, the ubiquitin-ligase anaphase-promoting complex/cyclosome and its coactivator CDH1 (APC/C[CDH1]) remains active, allowing origin licensing and blocking premature DNA replication. Paradoxically, glycolysis is reduced by APC/C[CDH1] through the degradation of key glycolytic enzymes[2], raising the question of how cells coordinate these mutually exclusive events to ensure proper cell division. Here we show that cells resolve this paradox by transiently inactivating the APC/C during cell cycle entry, which allows a transient metabolic shift favouring glycolysis. After mitogen stimulation, rapid mTOR-mediated phosphorylation of the APC/C adapter protein CDH1 at the amino terminus causes it to partially dissociate from the APC/C. This partial inactivation of the APC/C leads to the accumulation of PFKFB3, a rate-limiting enzyme for glycolysis, promoting a metabolic shift towards glycolysis. Delayed accumulation of phosphatase activity later removes CDH1 phosphorylation, restoring full APC/C activity, and shifting cells back to favouring oxidative phosphorylation. Thus, cells coordinate the simultaneous demands of cell cycle progression and metabolism through an incoherent feedforward loop, which transiently inhibits APC/C activity to generate a pulse of glycolysis that is required for mammalian cell cycle entry.

The decision of a cell to enter the cell cycle is an energetically demanding process, requiring the generation of biosynthetic building blocks such as amino acids and nucleic acids to accumulate enough biomass to divide into two daughter cells. To meet these demands, cells initially rely on a relative shift towards glycolysis, rather than oxidative phosphorylation (OXPHOS), to generate bioenergetics and biosynthetic macromolecules[1,3,4]. Simultaneously, mitogen signalling induces the transcription of genes that are required for proper cell cycle progression and activates proteins to license origins of replication ahead of S phase. Cells must coordinate these diverse requirements in a timely manner to prepare for cell division[5] (Fig. 1a). The ubiquitin ligase APC/C associated with CDH1 (hereafter, APC/C) has an important role in coordinating both glycolysis and cell cycle events by targeting key proteins in both pathways for degradation. During quiescence, the APC/C remains active, degrading glycolytic enzymes such as 6-phosphofructo-2-kinase/fructose-2, 6-bisphosphatase 3 (PFKFB3)[2,6], to promote a balance between OXPHOS and glycolysis, and also DNA licensing regulators such as cyclin A2 and geminin, creating a permissive environment for formation of the prereplication complex[7]. During cell cycle re-entry, the APC/C is thought to remain fully active until the G1/S transition, at which time it rapidly inactivates in a switch-like manner[8,9], allowing for origin

firing while preventing relicensing of newly synthesized DNA. However, this series of events is apparently contradictory. If the APC/C remains active until the G1/S transition, it raises the question of how cells can shift to glycolysis to meet the energy and biosynthetic demands of the cell ahead of S phase. Similarly, if the APC/C is prematurely inactivated on exit from quiescence, it raises the question of how cells can license origins of replication ahead of S phase. This paradoxical signalling architecture calls into question the current model of cell cycle entry and the molecular mechanisms thought to coordinate these essential cellular processes. Thus, studies are needed to understand how cells resolve this paradox and coordinate mutually exclusive events that are both required for cell cycle entry.

## APC/C transiently inactivates at the G0/G1 transition

The current model of cell cycle entry includes three key components: (1) glycolysis is necessary for cell cycle entry[10,11]; (2) the APC/C inhibits glycolysis[2,6]; and (3) the APC/C is active until the end of G1 phase when it is inactivated at the G1/S transition[9,12] (Fig. 1a). However, these three statements cannot all be correct simultaneously, and we sought to test each of these model components during cell cycle entry. First, to test

[1]Laboratory of Cancer Biology and Genetics, Center for Cancer Research, National Cancer Institute, Bethesda, MD, USA. [2]Department of Biochemistry and Biophysics, University of North Carolina, Chapel Hill, NC, USA. [3]Lineberger Comprehensive Cancer Center, University of North Carolina, Chapel Hill, NC, USA. [4]Protein Characterization Laboratory, Cancer Research Technology Program, Frederick National Laboratory for Cancer Research, Frederick, MD, USA. [5]Department of Molecular and Cellular Biology, University of California, Davis, CA, USA. [6]Laboratory of Cell Biology, Center for Cancer Research, National Cancer Institute, Bethesda, MD, USA. [7]Department of Pharmacology, University of North Carolina, Chapel Hill, NC, USA. ✉e-mail: steven.cappell@nih.gov

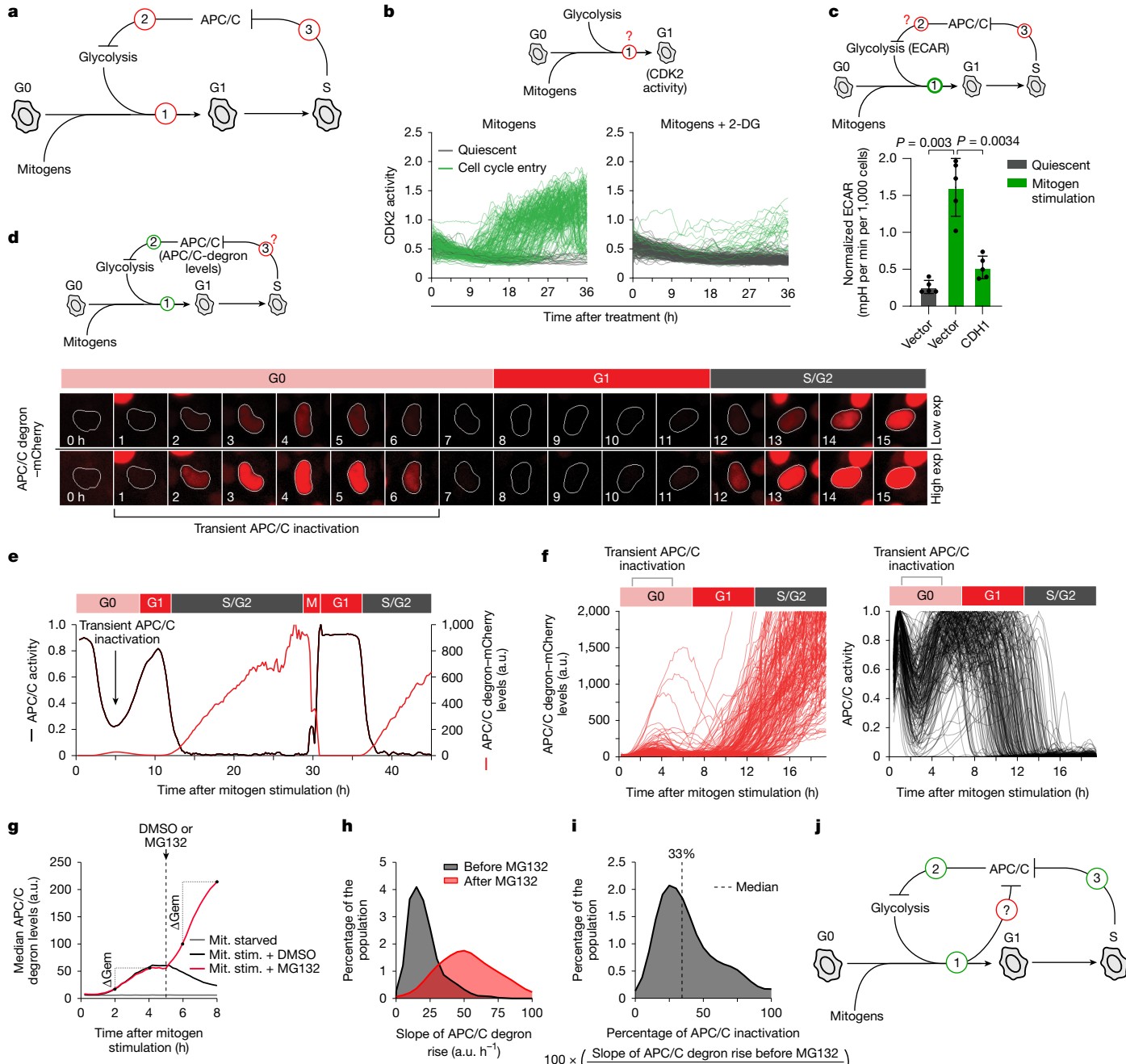

**Fig. 1 | APC/C transiently inactivates during the G0/G1 transition.**
**a**, Cells need mitogens and glycolysis to enter the cell cycle (1); APC/C inhibits glycolysis (2); and APC/C is reportedly fully active during quiescence and remains active until the G1/S transition (3). **b**, Test to determine whether glycolysis is required for cell cycle entry. CDK2 activity traces of single MCF-10A cells after mitogen stimulation with or without 2-DG after 48 h starvation-induced quiescence. Green, cells entering the cell cycle; grey, quiescent cells. $n$ = 100 cells per condition. **c**, Test to determine whether APC/C inhibits glycolysis. The ECAR levels in quiescent cells or 4 h after mitogen treatment in cells expressing empty vector or CDH1 are shown. Data are mean ± s.d. $n$ = 5. Statistical analysis was performed using one-way analysis of variance (ANOVA). **d**, Test to determine APC/C activity during the G1/S transition. Time-lapse images of a quiescent MCF-10A cell expressing an APC/C biosensor as it re-enters the cell cycle. Transient biosensor accumulation (red) occurs between 1 and 6 h, with sustained accumulation at G1/S. Bottom, higher exposure of the same cell. Exp, exposure.

**e**, Representative trace of APC/C biosensor levels (red) and APC/C activity (black) in a single MCF-10A cell re-entering the cycle after 48 h starvation and mitogen stimulation. **f**, Single-cell traces of APC/C biosensor levels (left) and APC/C activity (right) of cells entering the cell cycle. Cells were starved for 48 h to induce quiescence, followed by mitogen stimulation at time 0 h to promote cell cycle entry. $n$ = 200 cells for each condition. **g**, Median APC/C biosensor traces from cells with or without MG132 at the indicated times. MCF-10A cells were starved for 48 h, then stimulated (stim.) with mitogens (mit.). The pre-MG132 slope reflects APC/C inactivation; the post-MG132 slope reflects blocked biosensor degradation. **h**, APC/C biosensor-accumulation slopes from single cells before and after MG132 treatment from **g**. **i**, The percentage APC/C inactivation as measured by the ratio of APC/C biosensor-accumulation slopes before and after MG132 treatment from **g** and **h**. **j**, Updated model of cell cycle entry from **a**. Full growth medium promotes transient APC/C inactivation during cell cycle entry.

whether glycolysis is required for cell cycle entry, we serum-starved non-transformed MCF-10A breast epithelial cells for 48 h to induce quiescence (Extended Data Fig. 1a) and restimulated the cells with full growth medium to stimulate cell cycle entry, as assessed using a live-cell CDK2-activity biosensor[13] (Extended Data Fig. 1b). Treatment with the glycolysis inhibitor 2-deoxyglucose (2-DG) prevented CDK2 activation, but treatment with the OXPHOS inhibitor oligomycin had a limited effect[10] (Fig. 1b and Extended Data Fig. 1c,d). Furthermore, depleting glucose from the medium or inhibiting GLUT transporters blocked mitogen-induced cell cycle entry in MCF-10A cells (Extended Data Fig. 1e,f). We also found that depleting glucose from the medium or providing galactose as the only carbon source blocked cell cycle entry in MCF-10A cells, hTERT immortalized retinal pigment epithelial (RPE-1) cells and mouse embryonic fibroblasts (MEFs) (Extended Data Fig. 1g–i), confirming that glycolysis is required for cell cycle entry in diverse cell types. Second, to test whether the APC/C inhibits glycolysis, we exogenously expressed the APC/C co-activator protein CDH1 (encoded by *FZR1*) and measured glycolytic activity. CDH1 expression resulted in a significant reduction in basal extracellular acidification rate (ECAR), a glycolytic indicator (Fig. 1c), consistent with the APC/C inhibiting glycolysis[2,6]. Third, to test whether the APC/C remains active until the G1/S transition, we measured APC/C activity in single cells during cell cycle entry using an APC/C-activity biosensor[8,14] (Extended Data Fig. 2a). Notably, we observed that the APC/C partially and transiently inactivates during the G0/G1 transition (Fig. 1d–f, Extended Data Fig. 2b and Supplementary Video 1), approximately 6–8 h before the G1/S transition, when the APC/C fully inactivates[9]. We did not observe a similar transient APC/C inactivation in subsequent cell cycles, indicating that it is unique to the first cell cycle after an exit from quiescence (Extended Data Fig. 2c–e). Furthermore, we observed similar transient APC/C inactivation during cell cycle entry from quiescence induced by either MEK inhibition or contact inhibition (Extended Data Fig. 2f,g). We validated transient APC/C inactivation through immunoblotting and immunofluorescence analysis of endogenous APC/C substrates in multiple cell lines, including MCF-10A, RPE-1, primary human lung fibroblasts and MEFs (Extended Data Fig. 2h–l). Thus, we find that the third component of the cell cycle entry model is not supported by our experiments, and that the APC/C is transiently inactivated at the G0/G1 transition, hours before it is fully inactivated at the G1/S transition.

To further characterize this transient APC/C inactivation, we carefully measured the APC/C activity during cell cycle entry after the addition of full growth medium (hereafter, mitogen stimulation) when the geminin-based APC/C-activity biosensor had accumulated above background levels. We also measured the APC/C activity after treatment with the proteasome inhibitor MG132, which represents complete inhibition of the APC/C. We then determined the relative percentage of APC/C inhibition during the G0/G1 transition compared with complete inhibition after MG132 treatment. As the APC/C biosensor is under the control of a constitutive promoter, the slope of the increase in biosensor fluorescence is indicative of the relative APC/C activity in the cells[8]. We therefore compared the observed synthesis rate of the APC/C biosensor between 1 and 5 h after mitogen stimulation to the synthesis rate of the APC/C biosensor after treatment with MG132 in single cells. We found that the observed accumulation rate of the APC/C biosensor was approximately 33% less than when cells were treated with MG132, indicating that the APC/C is partially inactivated by approximately 33% during the G0/G1 transition (Fig. 1g–i and Extended Data Fig. 2m,n), in contrast to the full, switch-like inactivation at the G1/S transition[8]. Thus, cells appear to solve the paradoxical requirements for both active APC/C and glycolysis during cell cycle entry by transiently and partially inactivating the APC/C during cell cycle entry at the G0/G1 transition (Fig. 1j). The next question that arises is what causes this transient APC/C inactivation.

## mTORC1 inactivates APC/C via phosphorylation

We investigated the mechanism underlying transient APC/C inactivation by first testing the involvement of known negative regulators of the APC/C (Extended Data Fig. 3a). The cullin-based E3 ligases SCF[β-TrCP] and SCF[cyclin F] have been previously reported to degrade CDH1 (refs. 15,16), cyclin A2–CDK2 has been reported to phosphorylate CDH1 and inhibit its binding to the core APC/C scaffold[17] and EMI1 has been reported to function as a stoichiometric inhibitor of APC/C by binding to multiple regions of the complex[18–21]. Knockdown of these known APC/C regulators using small interfering RNA (siRNA) did not affect the transient APC/C inactivation during cell cycle entry (Extended Data Fig. 3b–g), suggesting that these negative regulators may tune APC/C activity in other phases of the cell cycle but do not have a role in transient APC/C inactivation during cell cycle entry. Given that transient APC/C inactivation is triggered by full growth medium, we hypothesized that growth-promoting kinases could be involved in transient APC/C inactivation during cell cycle entry (Fig. 2a). Inhibition of MEK, CDK1/2 or CDK4/6 did not affect transient APC/C inactivation. However, we found that inhibition of the mTOR signalling pathway using rapamycin, the mTORC1-selective inhibitor RMC6272 or knockdown of *RPTOR* but not *RICTOR* resulted in decreased transient APC/C inactivation during cell cycle entry (Fig. 2b,c and Extended Data Fig. 3h–m), indicating that mTORC1 or its downstream signalling targets regulate APC/C activity during cell cycle entry.

Immunoprecipitation of whole-cell lysates collected from MCF-10A cells entering the cell cycle (4 h after mitogen stimulation) using either a CDH1, an mTOR or an unrelated protein antibody confirmed specific binding of CDH1 to the mTORC1 complex (Fig. 2d and Extended Data Fig. 4a–c). To determine whether CDH1 is a substrate of mTOR, we used recombinant mTOR complex and purified CDH1 (Extended Data Fig. 4d) and assessed their binding in vitro. Pull-down assays showed binding between CDH1 and mTOR (Fig. 2e), and that this interaction leads to the phosphorylation of CDH1 (Fig. 2f and Extended Data Fig. 4e–i), establishing that CDH1 is a bona fide substrate of mTOR. To assess the kinetics of mTOR-mediated CDH1 phosphorylation, we immunoprecipitated CDH1 from MCF-10A whole-cell lysates entering the cell cycle in the presence or absence of the mTOR inhibitor rapamycin. CDH1 phosphorylation was detectable as early as 1 h after mitogen stimulation and increased over time but was reduced in the presence of rapamycin (Extended Data Fig. 4j). Furthermore, in support of our earlier measurements revealing that the APC/C is partially inactivated during cell cycle entry, we found that mTOR-mediated phosphorylation causes a partial dissociation of CDH1 from the APC/C, as determined by its binding to APC2 (Extended Data Fig. 4j).

Phosphoproteomic analysis of CDH1 using mass spectrometry (MS) identified threonine 129 (Thr129) or tyrosine 130 (Tyr130) as a potential site of mTOR-mediated phosphorylation (Extended Data Fig. 5a). Follow-up experiments confirmed that mTOR phosphorylates a threonine residue (Extended Data Fig. 5b,c), probably without affecting tyrosine residues (Extended Data Fig. 5d) of CDH1. Consistent with our live-cell imaging data, we found that phosphorylated CDH1 (p-CDH1) transiently accumulates with the same kinetics as transient APC/C inactivation (Extended Data Fig. 5e,f). In vitro binding analysis of mTOR with CDH1 revealed a significant decrease in binding when the Thr129 residue was mutated to a phospho-mimetic aspartic acid (CDH1(T129D)) compared with the wild type (Fig. 2g and Extended Data Fig. 5g). Furthermore, CDH1(T129A) was less phosphorylated by mTOR both in an in vitro kinase assay (Fig. 2h,i) as well as in vivo (Fig. 2j), indicating that mTOR primarily phosphorylates CDH1 during cell cycle entry.

We assessed the effect of mTOR-mediated phosphorylation of CDH1 and observed that CDH1(T129D) bound less effectively to the core APC/C (Extended Data Fig. 5h,i). Next, to assess the effect of CDH1 Thr129 phosphorylation on APC/C activity, we reconstituted substrate

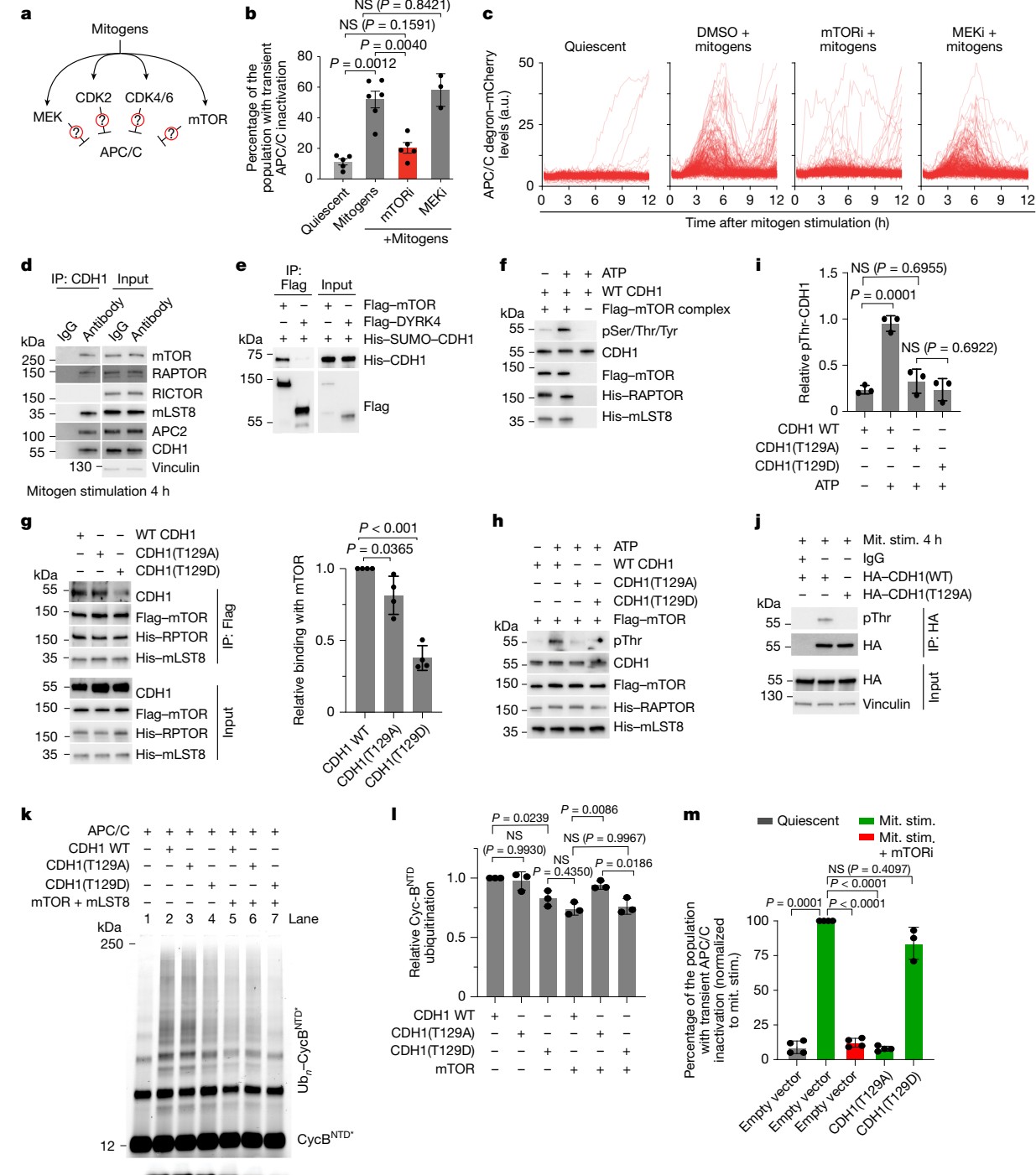

**Fig. 2 | mTORC1 phosphorylates APC/C during cell cycle entry. a**, Schematic showing how mitogen-activated kinases may transiently inactivate APC/C. **b,c**, APC/C inactivation during cell cycle entry. **b**, The fraction of cells. Data are mean ± s.e.m. $n = 5, 6, 5$ and 3 independent experiments. **c**, Single-cell biosensor traces. $n = 200$ cells per condition. MCF-10A cells were starved for 48 h, then stimulated with mitogens with or without kinase inhibitors. $P$ values were calculated using one-way ANOVA; NS, not significant. **d**, Immunoprecipitation analysis of CDH1 from MCF-10A cells (48 h starvation, 4 h mitogen stimulation) probed for the indicated proteins. Representative blot. $n = 3$. **e**, Co-immunoprecipitation of purified His–Sumo–CDH1 with DDK/Flag–mTOR or DDK/Flag–DYRK4 in a cell-free system. Representative blot. $n = 3$. **f**, In vitro kinase assay of purified CDH1 with recombinant mTOR complex, probed with a pan-phospho-serine/threonine/tyrosine antibody. Representative blot. $n = 3$. **g**, In vitro binding of CDH1 WT, T129A and T129D (200 ng each) to recombinant DDK–mTOR. Left, representative blot. Right, quantification. Data are mean ± s.d.

$n = 4$. $P$ values were calculated using one-way ANOVA. **h**, In vitro kinase assay of CDH1 mutants with recombinant DDK–mTOR, probed with a pan-phospho-threonine antibody. Representative blot. $n = 3$. **i**, Quantification of **h**. $P$ values were calculated using one-way ANOVA. **j**, Immunoprecipitation of HA–CDH1 WT or T129A from MCF-10A cells (48 h starvation, 4 h mitogen stimulation), probed for the indicated proteins. Representative blot. $n = 3$. **k**, In vitro ubiquitylation reactions using recombinant APC/C, UBE2C and CDH1 WT, CDH1(T129A) or CDH1(T129D), plus recombinant mTOR and mLST8. Ubiquitinated cyclin B[NTD] was detected by fluorescence scanning. Representative gel. $n = 3$. **l**, Quantification of relative cyclin B[NTD] ubiquitination from **k**. Data are mean ± s.e.m. $n = 3$. $P$ values were calculated using one-way ANOVA. **m**, Quantification of the percentage of the population with transient APC/C inactivation normalized to the control from Extended Data Fig. 5m. Data are mean ± s.d. from $n = 4$ (bars 1–4) or $n = 3$ (bar 5) independent experiments. $P$ values were calculated using one-way ANOVA.

polyubiquitination by APC/C using a recombinant system. We performed APC/C–UBE2C-dependent ubiquitination assays in the presence of wild-type CDH1, CDH1(T129A) or CDH1(T129D). We observed that CDH1(T129D) is less effective in promoting ubiquitination of substrates compared with either wild-type CDH1 or CDH1(T129A) (Fig. 2k,l and Extended Data Fig. 5j,k). Furthermore, adding active mTOR to the reaction reduced the effectiveness of wild-type CDH1 in promoting substrate ubiquitination to similar levels to the CDH1(T129D) mutant, but it did not affect the CDH1(T129A) mutant, indicating that mTOR-mediated modification leads to partial APC/C inactivation (Fig. 2k,l). Notably, phosphorylation of the unstructured N-terminal region of CDH1 by other kinases was previously shown to disrupt the binding of CDH1 to the APC/C and therefore inhibit APC/C activity in a similar manner[22] (Extended Data Fig. 5l). Specifically, it was shown[22] that four of the CDK2 target sites, Ser40, Thr121, Ser151 and Ser163, are necessary and sufficient to mediate complete CDH1 inactivation, but that each residue contributes to a degree of partial APC/C inactivation. Thus, our data are consistent with these previous findings and support a model in which mTOR phosphorylation of CDH1 disrupts the CDH1–APC/C interaction through electrostatic repulsion and steric clashes, leading to the partial inactivation of APC/C.

To determine the effect of mTOR-mediated CDH1 phosphorylation on APC/C activity during cell cycle entry, we introduced exogenous CDH1(T129A) and CDH1(T129D) mutants into MCF-10A cells that were entering the cell cycle and then measured transient APC/C inactivation. Expression of the T129A mutant reduced transient APC/C inactivation to similar levels as rapamycin treatment, acting in a dominant-negative manner (Fig. 2m and Extended Data Fig. 5m). Conversely, expression of the phosphomimetic T129D mutant did not affect transient APC/C inactivation compared with the empty vector control when exogenously expressed in the presence of wild-type CDH1 (Fig. 2m, Extended Data Fig. 5m and Supplementary Video 2). Furthermore, expression of CDH1(T129D) resulted in sustained APC/C-degron accumulation when exogenously expressed while also knocking down endogenous *CDH1* using an siRNA targeting the 3′-UTR of CDH1 (Extended Data Fig. 5n,o). Taken together, these data support a model in which, after mitogen stimulation, mTOR phosphorylates CDH1, causing it to partially dissociate from the core APC/C, resulting in partial APC/C inactivation during cell cycle entry.

## APC/C regulation by an incoherent feedforward loop

Having established that mTOR-mediated phosphorylation of CDH1 is responsible for transient APC/C inactivation, we next sought to understand how and why the APC/C gets reactivated to generate the transient APC/C inactivation that we observed in our live-cell imaging experiments. We considered two main hypotheses: either mTOR itself is activated transiently or a protein phosphatase removes the mTOR-mediated phosphorylation from CDH1 (Fig. 3a). We tested the first hypothesis by immunoblotting for p-mTOR and its direct substrate p-4EBP1. We assessed the ratio of p-mTOR to mTOR to quantify the mTOR activity and found a slight increase in the p-mTOR/mTOR ratio between 3 and 7 h after mitogen stimulation when APC/C transiently reactivates (3–5 h after mitogen stimulation) (Extended Data Fig. 6a,b), suggesting that regulation of mTOR could in part underlie the regulation of transient APC/C inactivation. We next tested the second hypothesis by treating MCF-10A cells with a pan-protein-phosphatase inhibitor during cell cycle entry[23–25]. Inhibition of protein phosphatases resulted in an initial APC/C inactivation, as indicated by accumulation of the APC/C biosensor. However, this accumulation was sustained, indicating that the APC/C did not reactivate (Fig. 3b, Extended Data Fig. 6c and Supplementary Video 3). Notably, sustained accumulation of the APC/C biosensor after treatment with protein phosphatase inhibitors could be rescued by expressing CDH1(T129A), which is insensitive to both phosphorylation and dephosphorylation (Extended Data Fig. 6d).

Consistent with these live-cell imaging observations, we found that inhibiting protein phosphatases leads to sustained CDH1 phosphorylation, sustained dissociation of CDH1 with the core APC/C components APC2 and APC6 (Fig. 3c), and reduced polyubiquitination of the APC/C biosensor (Fig. 3d). These data support the second hypothesis that protein-phosphatase-mediated dephosphorylation is required for APC/C reactivation during cell cycle entry.

We next investigated the timing and interplay between mTOR-mediated APC/C inactivation and protein-phosphatase-mediated APC/C reactivation. Inhibiting mTOR with rapamycin within 1 h of mitogen stimulation blocked the transient APC/C inactivation but, when rapamycin was added two or more hours after mitogen stimulation, the APC/C transiently inactivated to similar levels to the control (Fig. 3e). These data are consistent with our earlier observation that mTOR is rapidly activated within 1 h of mitogen stimulation (Extended Data Fig. 6a) and CDH1 phosphorylation is already detected 1 h after mitogen stimulation (Extended Data Fig. 4j), indicating that mTOR-mediated phosphorylation of CDH1 is a rapid/early process after mitogen stimulation (Fig. 3e). To investigate when protein phosphatase activity on CDH1 becomes relevant to APC/C activity, we treated cells with mitogens and a protein phosphatase inhibitor at time zero and measured when APC/C biosensor levels diverged from those of the cells treated with mitogens alone. Analysis of the median APC/C biosensor levels revealed that they co-accumulated with each other until 4 h from the addition of mitogens, at which point, the median traces diverged, with the phosphatase inhibitor-treated cells continuing to accumulate the APC/C biosensor (Fig. 3f). Furthermore, addition of the phosphatase inhibitor to quiescent cells did not induce APC/C biosensor accumulation, whereas addition of the phosphatase inhibitor at various timepoints more than 4 h after mitogen stimulation caused an immediate accumulation of APC/C biosensor levels (Extended Data Fig. 6e,f). These data indicate that protein phosphatases first start to affect APC/C activity 4 h after mitogen stimulation—a relatively delayed process compared with mTOR activity. Notably, this combination of an inhibitory regulation and an activating regulation generates an incoherent feedforward circuit—a signalling motif capable of generating transient pulses of activity[26,27]. To demonstrate how this incoherent feedforward loop, comprising rapid mTOR activation and delayed protein phosphatase activation, could affect the kinetics of APC/C activity, we used a mathematical model and incorporated the timing parameters that we measured experimentally. The APC/C activity levels generated by this model displayed transient and partial APC/C inactivation, matching our experimental observations for both APC/C activity and APC/C substrate levels, demonstrating how the proposed signalling architecture is consistent with the observed dynamics (Fig. 3g). Overall, we find an incoherent feedforward loop involving mTOR, APC/C and a protein phosphatase that generates transient and partial APC/C inactivation during cell cycle entry (Fig. 3h).

## Transient APC/C inactivation boosts glycolysis

Given that APC/C transiently inactivates during cell cycle entry and that APC/C inhibits glycolysis, we hypothesized that cells may transiently shift to glycolysis during the 1–6 h when the APC/C is partially inactivated. One of the APC/C substrates linked to glycolysis is PFKFB3, a key enzyme regulating glycolytic flux[1,2,28,29]. We confirmed previous reports that PFKFB3 is a substrate of APC/C[2,6] (Extended Data Fig. 7a–c). Moreover, we observed that both PFKFB3 and PFKFB2 transiently accumulate during cell cycle entry, with PFKFB2 exhibiting a prolonged accumulation compared with PFKFB3. By contrast, PFKFB1 did not show transient accumulation (Extended Data Fig. 7d). We confirmed that PFKFB3, but not PFKFB1 or PFKFB2, accumulates during cell cycle entry in an APC/C-dependent manner and is independent of β-TrCP, another ubiquitin ligase that is known to ubiquitinate PFKFB3 (ref. 6) (Extended Data Fig. 7e). The accumulation of PFKFB3 is inhibited by

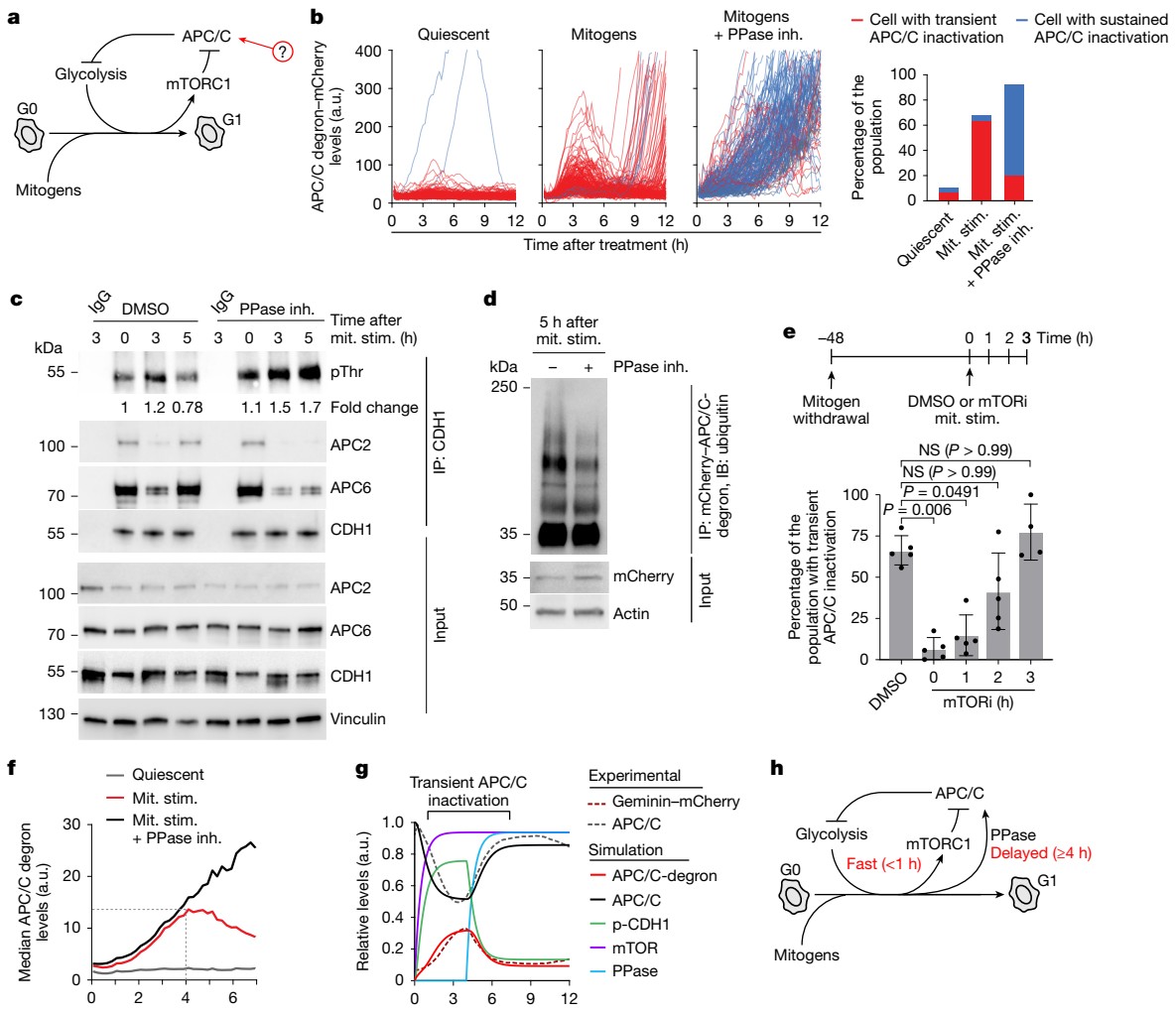

**Fig. 3 | An incoherent feedforward loop controls APC/C activity during cell cycle entry. a**, Schematic of the hypothesis for APC/C reactivation. **b**, MCF-10A cells were starved for 48 h, then stimulated with mitogens with or without sodium fluoride (NaF) and sodium orthovanadate ($Na_3VO_4$) phosphatase inhibitor (PPase inh.). Single-cell APC/C biosensor traces are shown. Left, quantification of cells with transient or sustained APC/C inactivation. **c**, MCF-10A cells were starved for 48 h, then stimulated with mitogens with or without phosphatase inhibitor for the indicated times. Lysates were immunoprecipitated with CDH1 antibodies and probed for the indicated proteins. Representative blot from $n$ = 3 experiments. **d**, MCF-10A cells were starved for 48 h, then stimulated with mitogens with or without phosphatase inhibitor for 5 h. MG132 (10 μM) was added at 2 h. At 5 h, lysates were immunoprecipitated with mCherry antibodies and probed for the indicated proteins. Representative blot from $n$ = 3 experiments. **e**, Top, schematic of the experimental timeline. MCF-10A cells were starved for 48 h and released with or without rapamycin at the indicated

times. Bottom, quantification of cells with transient APC/C inactivation. Data are mean ± s.d. from $n$ = 5, 5, 5, 5 and 4 experiments. Statistical analysis was performed using Kruskal–Wallis tests. **f**, MCF-10A cells were starved for 48 h, then stimulated with mitogens with or without phosphatase inhibitor. Median APC/C biosensor traces are shown. The dashed line marks the divergence between the mitogen (red) and mitogen + phosphatase inhibitor (black) conditions at 4 h. **g**, The output from mathematical modelling: an incoherent feedforward loop involving mTOR, a protein phosphatase and the APC/C through phosphorylation of CDH1. Experimentally derived APC/C activity and APC/C biosensor levels (geminin–mCherry) are plotted for comparison. **h**, Schematic showing an incoherent feedforward loop between mTOR, APC/C and protein phosphatase. The process of mTOR-mediated APC/C inactivation is relatively fast (<1 h) and the process of protein phosphatase-mediated APC/C reactivation is a relatively delayed process (>4 h).

rapamycin (Fig. 4a and Extended Data Fig. 7f) or by expression of the CDH1(T129A) mutant either transiently or using an inducible expression system (Fig. 4b and Extended Data Fig. 8a–g). Mutating PFKFB3's KEN box, a known APC/C degron sequence, resulted in higher levels of PFKFB3 after mitogen stimulation that could not be inhibited by expressing the CDH1(T129A) mutant (Extended Data Fig. 8h,i). The build-up of PFKFB3 is the result of post-transcriptional regulation, as the mRNA levels of *PFKFB3* are not affected by CDH1(T129A) expression (Extended Data Fig. 8j). During the time when APC/C is transiently inactivated, we observed relatively low levels of PFKFB3 polyubiquitination, but inhibiting mTOR with rapamycin was able to increase PFKFB3 polyubiquitination in a CDH1-dependent manner (Fig. 4c). Similarly,

expression of the CDH1(T129A) but not the CDH1(T129D) mutant led to increased PFKFB3 polyubiquitination and lower PFKFB3 protein levels (Fig. 4d and Extended Data Fig. 8k,l). Finally, treatment of cells with a protein phosphatase inhibitor 5 h after mitogen stimulation showed reduced PFKFB3 polyubiquitination (Fig. 4e), consistent with PFKFB3 ubiquitination being regulated by the mTOR-mediated CDH1 phosphorylation status.

To determine whether the PFKBF3 accumulation by transient APC/C inactivation leads to a pulse of bioenergetics[29], we first measured total ATP levels during cell cycle entry. We found a steady increase in total cellular ATP levels during cell cycle entry (Fig. 4f,g). Given that PFKFB3 is one of the rate-limiting enzymes of glycolysis[1,28,29]

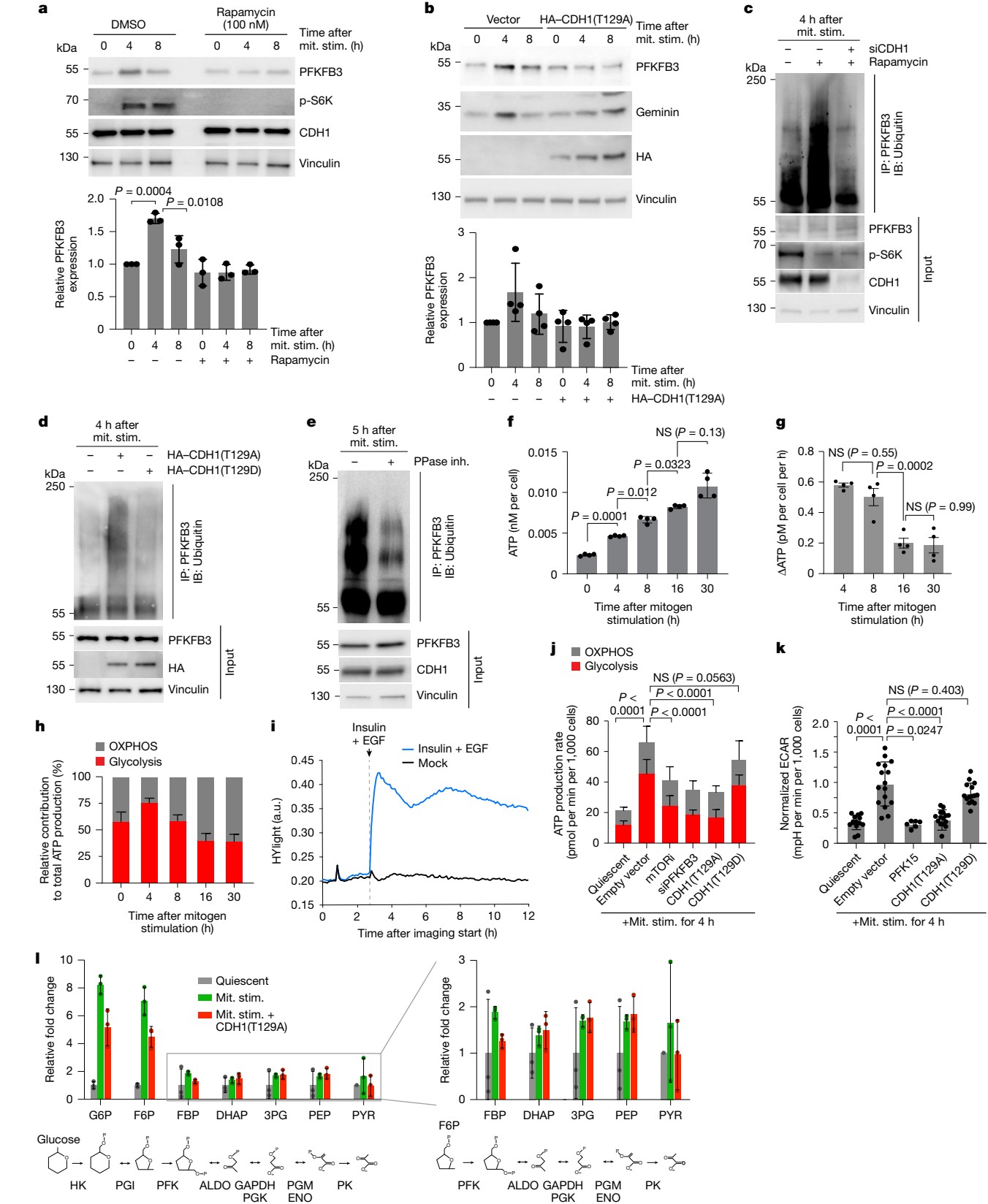

**Fig. 4 | See next page for caption.**

(Extended Data Fig. 9a), we considered whether the increase in total ATP levels we observed was due to an increase in the glycolytic rate during cell cycle entry. We therefore measured the contribution of glycolysis to the rate of ATP production during cell cycle entry using a

Seahorse ATP rate assay. While quiescent cells relied on a relative even balance between the rate of glycolysis and OXPHOS, we observed an increase in the relative rate of ATP production due to glycolysis 4 h after mitogen stimulation, followed by a relative decrease in the rate of ATP

**Fig. 4 | PFKFB3 transiently accumulates during cell cycle entry to promote a metabolic switch. a,b**, PFKFB3 levels in MCF-10A cells that were starved for 48 h, then stimulated with mitogens with or without DMSO or rapamycin (**a**) or transfected with vector or HA–CDH1(T129A) (**b**). Quantification is shown below the blots. Data are mean ± s.d. $n = 3$ (**a**) and $n = 4$ (**b**). P values were calculated using one-way ANOVA. **c,d**, Immunoprecipitation of PFKFB3 in starved MCF-10A cells that were stimulated with mitogens (0 h), and treated with MG132 (10 μM) at 1 h. Cells were collected at 4 h. Cells were treated with *CDH1* siRNA with or without rapamycin (**c**) or expressing HA–CDH1(T129A) or HA–CDH1(T129D) (**d**). Representative blots. $n = 3$. **e**, Immunoprecipitation of PFKFB3 in starved MCF-10A cells stimulated with mitogens with or without phosphatase inhibitor (0 h) and treated with MG132 (10 μM) at 1 h. Cells were collected at 5 h. Representative blot. $n = 3$. **f,g**, Total ATP (**f**) and change in ATP levels (**g**) in quiescent or mitogen-stimulated MCF-10A cells. Data are mean ± s.d. $n = 4$. P values were calculated using one-way ANOVA. **h**, Normalized ATP from glycolysis (red) or the OXPHOS pathway (grey) under the same conditions as in **f**. Data are mean ± s.d. from $n = 3$ independent experiments. **i**, The HYlight signals (GFP:Sapphire ratio) in MCF-10A cells that were cultured for 50 h without growth factors, then treated with or without EGF (20 ng ml⁻¹) and insulin (10 μg ml⁻¹). The mean traces represent $n \geq 2,500$ cells. **j**, Quantification of the ATP-generation rate from glycolysis or OXPHOS in quiescent MCF-10A cells or in MCF-10A cells after mitogen stimulation for 4 h, with the indicated treatments, ectopic expression or gene knockdown. Data are mean ± s.d. from $n = 3$ independent experiments. P values were calculated using one-way ANOVA. **k**, The normalized basal ECAR in quiescent MCF-10A cells or MCF-10A cells after 4 h of mitogen stimulation with the indicated treatments or ectopic gene expression. Data are mean ± s.d. $n = 3$ experiments; 5 technical replicates, except for PFK15, for which there were 2 technical replicates. P values were calculated using one-way ANOVA. **l**, The relative mean abundance of traced glycolytic metabolites in quiescent or mitogen-stimulated (4 h) MCF-10A cells with or without ectopic CDH1(T129A) expression. Isotopologues: M+6 (glucose 6-phosphate (G6P), fructose 6-phosphate (F6P), fructose 1,6-bisphosphate (FBP)); M+3 (dihydroxyacetone phosphate (DHAP), glyceraldehyde 3-phosphate (3PG), phosphoenolpyruvate (PEP), pyruvate (PYR)). Data are mean ± s.d. normalized to protein. $n = 3$. Inset: magnification of the indicated metabolites. Bottom, schematic of glucose-to-pyruvate carbon flow. ALDO, aldolase; ENO, enolase; GAPDH, glyceraldehyde 3-phosphate dehydrogenase; HK, hexokinase; PFK, phosphofructo-kinase-1; PGI, phosphoglucose isomerase; PGK, phosphoglycerate kinase; PGM, phosphoglycero-mutase; PK, pyruvate kinase.

production due to glycolysis 8 h after mitogen stimulation, coinciding with the time when APC/C is transiently inactivated (Fig. 4h and Extended Data Fig. 9b,c). To assess the changes in the glycolytic rate at a higher time resolution, we conducted live-cell imaging experiments using the HYlight reporter, which measures glycolytic flux through the fructose 1,6-bisphosphate concentration[30], and took measurements every 6 min. Similar to the Seahorse assays, we observed a transient increase in the HYlight signal, indicative of a transient increase in the rate of glycolysis after mitogen stimulation, coinciding with transient APC/C inactivation (Fig. 4i). The transient increase in the HYlight signal was suppressed by depriving cells of glucose and glutamine, or depriving cells of glucose, glutamine and pyruvate (Extended Data Fig. 9d,e). Consistently, we also observed a transient increase in glucose uptake after mitogen stimulation (Extended Data Fig. 9f), corroborating previously published reports that found increased labelled glucose uptake 2 h after mitogen stimulation[31]. This transient increase in the glycolytic rate during cell cycle entry can be inhibited by rapamycin, expressing the CDH1(T129A) mutant, depleting *PFKFB3* using siRNA (Fig. 4j and Extended Data Fig. 9g–i) or inhibiting PFKFB3 using the small-molecule inhibitor PFK15 (Fig. 4k). Furthermore, ectopic expression of CDH1(T129A) results in compartmentalization of early glycolytic steps, leading to a reduced flow of G6P/F6P, but relative build-up of lower glycolytic intermediates (Fig. 4l). This is consistent with reduced glycolytic flux, in agreement with previous studies[32,33]. By contrast, expressing PFKFB3^KEN-mut, which is resistant to APC/C-mediated degradation, rescues the lower glycolytic rate observed when expressing the CDH1(T129A) mutant (Extended Data Fig. 9j), and expressing the phosphomimetic CDH1(T129D) mutant does not affect the transient increase in the glycolytic rate (Fig. 4j,k). Protein translation is one of the most energetically demanding processes, largely depending on the availability of biosynthetic macromolecules and ATP[34]. Consistent with a reduction in the availability of these macromolecules and ATP, we observed that MCF-10A cells expressing the CDH1(T129A) mutant, but not the CDH1(T129D) mutant, showed a significant decrease in protein translation (Extended Data Fig. 9k,l). Thus, cells transiently increase the rate of glycolysis during cell cycle entry and, when the APC/C is prevented from transiently inactivating, cells cannot increase the glycolysis rate, resulting in lower levels of glycolytic products and reduced protein translation.

### Transient APC/C inactivation aids cell cycle entry

Having established that transient APC/C inactivation is needed for the proper management of bioenergetics and biosynthetic macromolecules during cell cycle entry, we hypothesized that transient APC/C inactivation is necessary for cell cycle entry. To test this hypothesis, we exogenously expressed the CDH1(T129A) and CDH1(T129D) mutants in MCF-10A cells entering the cell cycle and monitored cell cycle entry using the CDK2 biosensor. We observed that expression of the CDH1(T129A) mutant significantly reduced the number of cells that activate CDK2, enter the cell cycle and proliferate (Fig. 5a–d and Extended Data Fig. 10a). Conversely, expression of the CDH1(T129D) mutant that binds poorly to the APC/C core had little effect on the number of cells that activate CDK2, enter the cell cycle and proliferate compared with the control (Fig. 5a–d), suggesting that transient APC/C inactivation is needed for robust cell cycle entry. We further tested our hypothesis by treating MCF-10A cells with the PFKFB3 inhibitor PFK15 and also observed a reduced number of cells entering the cell cycle as measured by CDK2 activity (Extended Data Fig. 10b). Notably, transiently expressing the CDH1(T129A) mutant or treating cells with PFK15 for only the first 8 h or 6 h after mitogen stimulation, approximately when we observed transient APC/C inactivation and transient PFKFB3 accumulation, respectively, still significantly reduced the number of cells entering the cell cycle, indicating that the transient accumulation of PFKFB3, and the transient increase in glycolytic rate that happens as a result, is necessary for the activation of CDK2 and cell cycle entry (Extended Data Fig. 10c–f).

### Discussion

Here we show how cells coordinate the dual but mutually exclusive requirement for APC/C activity and glycolysis during cell cycle entry. Typically, the APC/C functions as a molecular switch by turning on or off to control progression through critical cell cycle phase transitions using signalling motifs involving positive and negative feedback loops[9,15,16,35]. Our study shows that, in addition to switch-like regulation by cyclin-dependent kinases at both the G1/S and the metaphase/anaphase transitions[7,9], mTOR mediates a transient and partial inactivation of the APC/C at the G0/G1 transition through phosphorylation of CDH1 (Fig. 5e). While we identified Thr129 as an important site phosphorylated by mTOR, it is possible that mTOR also phosphorylates additional amino acids on the N terminus of CDH1, similar to the multisite phosphorylation of CDH1 by CDK2 at the G1/S transition. Overall, we found that, during cell cycle entry, mitogens trigger an incoherent feedforward loop to generate a transient pulse of APC/C inactivation, resulting in a transient accumulation of PFKFB3 (including other APC/C substrates such as geminin) and a temporary increase in glycolysis. Thus, cells coordinate metabolism with the rest of the cell

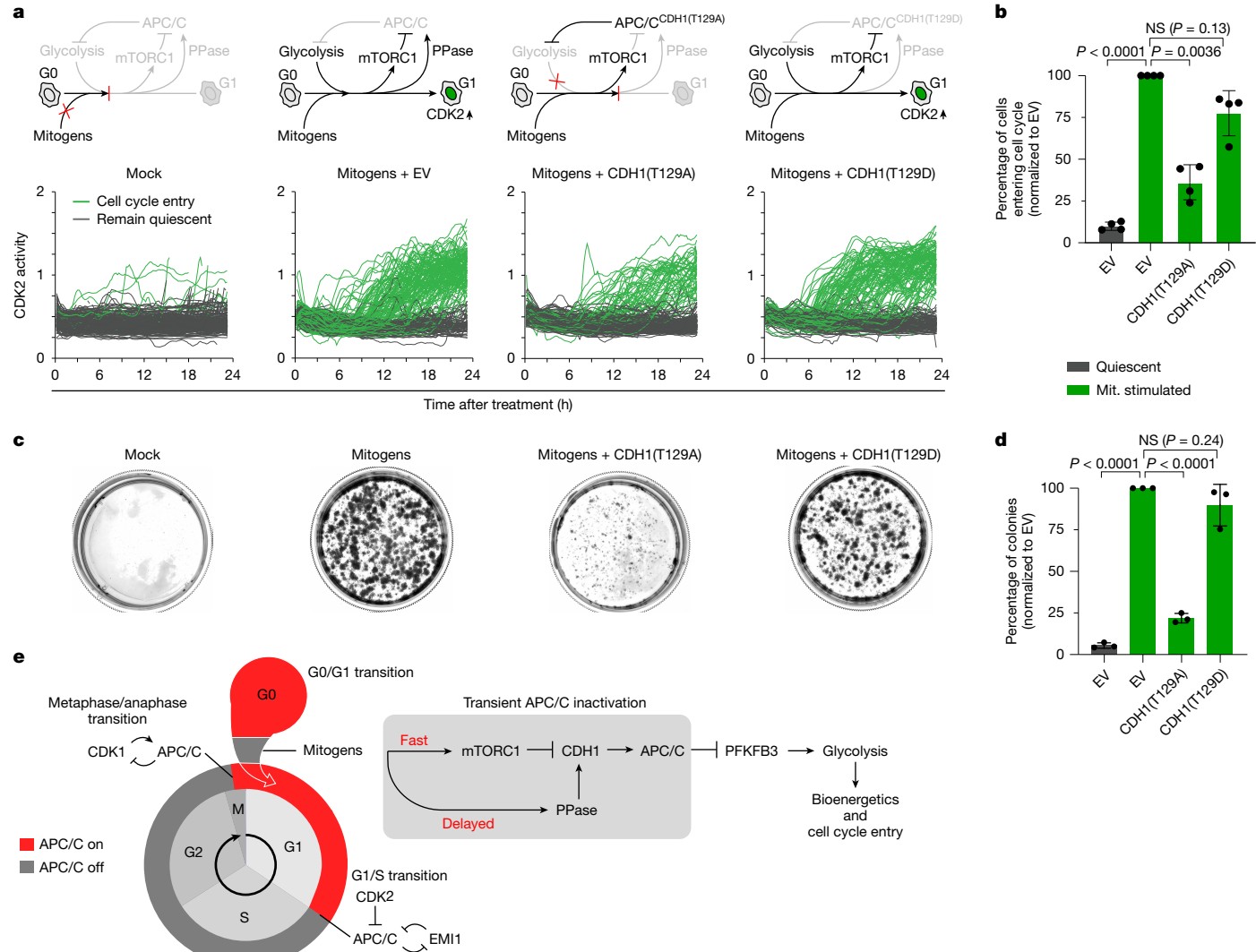

**Fig. 5 | Transient APC/C inactivation promotes cell cycle entry. a**, Schematic of the signalling pathway that results from each treatment (top). Data are single-cell CDK2 activity traces of MCF-10A cells treated as indicated. Cells were first mitogen-starved for 48 h to induce quiescence followed by ectopic overexpression of empty vector (EV), HA–CDH1(T129A) or HA–CDH1(T129D). $n$ = 200 cells per condition. **b**, Relative quantification of the percentage of cycling cells from **a** normalized to the empty vector control. Data are mean ± s.d. from $n$ = 4 independent experiments. $P$ values were calculated using one-way ANOVA. **c**, Long-term colony-formation assay of MCF-10A cells expressing inducible CDH1(T129A) or CDH1(T129D) protein 72 h after mitogen withdrawal. Cells were allowed to grow normally for 7 days with the induction of the indicated proteins, fixed and stained. Representative images from $n$ = 3 independent experiments. **d**, Quantification of the percentage of colonies from **c**. Data are mean ± s.d. from $n$ = 3 independent experiments. $P$ values were calculated using one-way ANOVA. **e**, Schematic of the events during cell cycle entry, glycolysis and APC/C activity. After cell cycle entry, mTOR phosphorylates CDH1 leading to transient APC/C inactivation. Protein phosphatases remove CDH1 phosphorylation leading to APC/C reactivation. Transient APC/C inactivation allows for a temporary switch to glycolysis, coordinating ATP production and the synthesis of macromolecules with the rest of the cell cycle machinery needed for cell cycle entry.

cycle machinery through the temporal and dynamic regulation of the APC/C and its substrates to ensure robust cell cycle entry. Notably, we observed transient APC/C inactivation only during the first cell cycle after exit from quiescence and not in the subsequent cell cycles, suggesting that cells require a metabolic jumpstart to ignite the cell cycle engine and leave quiescence. However, once running, the cell cycle metabolic machinery appears to be capable of sustaining its momentum through multiple cell cycles[36].

Our findings demonstrate that transient inactivation of APC/C during cell cycle entry triggers a metabolic switch favouring glycolysis dependent on PFKFB3 in normal cells. Notably, cancer cells exhibit a distinct metabolic phenotype, often relying on glycolysis for their sustenance and rapid proliferation. As our data indicate this APC/C-mediated metabolic shift promotes cell cycle entry, it may be deregulated in

diseases such as cancer, which often display perturbed cell cycle entry mechanisms[37]. Further exploration of the interplay between the mTOR–APC/C-protein phosphatase loop holds potential for identifying therapeutic targets and developing innovative strategies to intervene in disease. Elucidating the intricate dynamics of this molecular network will not only advance our understanding of cell cycle regulation but could also offer promising routes for targeted therapeutic interventions in cancer and related disorders.

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

## Methods

### Cell culture

Human MCF-10A, human retinal pigment epithelial (RPE-1), human lung fibroblasts (HLFs) and transformed human embryonic kidney (HEK293T) cells were obtained from ATCC. MEFs were gift from J. Vidigal. Cell lines not obtained from ATCC were not authenticated. MEFs, HLFs and HEK293T cells were cultured in Dulbecco's modified Eagles medium (DMEM) (Gibco, Life Technologies) containing 10% FBS (Gibco). MCF-10A cells were cultured in the following full-growth medium: phenol-red-free DMEM/F12 (Invitrogen) supplemented with 5% horse serum, 20 ng ml$^{-1}$ EGF, 10 μg ml$^{-1}$ insulin, 500 μg ml$^{-1}$ hydrocortisone, 100 ng ml$^{-1}$ cholera toxin and 1% penicillin–streptomycin. To induce quiescence, cells were mitogen-starved for 48–72 h with DMEM/F12 (Invitrogen) supplemented with 0.3% bovine serum albumin, 100 ng ml$^{-1}$ cholera toxin and 1% penicillin–streptomycin. For starvation experiments longer than 48 h, starvation medium was replaced in each 48 h interval. For contact inhibition mediated quiescence, cells were seeded at 90% confluence and allowed to grow in full growth medium for 48 h to induce quiescence. Where indicated, quiescent cells were transfected with 20% OPTI-MEM (Invitrogen) for 6 h, followed by replenishing with starvation medium for an additional 24 h before mitogen stimulation. To induce quiescence through MEK inhibition, cells in complete medium were treated with 100 nM trametinib for 48 h. RPE-1 cells were cultured in DMEM/F12 (Invitrogen) supplemented with 10% FBS and with 0.01 mg ml$^{-1}$ hygromycin B. For mitogen-removal experiments, RPE-1 cells were incubated with the above-described composition supplemented with 0.3% BSA and without FBS. MEFs were starved with DMEM containing 0% serum for 72 h. All tissue culture media were supplemented with 2 mM L-glutamine, 25 μg ml$^{-1}$ streptomycin and 25 U penicillin (Gibco). Cells were cultured in a humidified atmosphere with 5% CO$_2$ at 37 °C. For all pTet$_{on}$ experiments, cells were pretreated with doxycycline for 12 h in starved medium followed by induction with mitogen and doxycycline if not otherwise mentioned. Cells were routinely tested for mycoplasma.

### Constructs and stable cell lines

CSII-pEF1α-H2B-mTurquoise, CSII-pEF1α-mCherry-Geminin(amino acids 1–110) and CSII-pEF1α-DHB(amino acids 994–1087)-mVenus were described previously[8,13]. HA–CDH1 was a gift from M. Santra, HA–CDH1(T129A) and HA–CDH1(T129D) were subcloned using Gibson cloning. CMV-Hylight was a gift from R. Goodman (Addgene, 193447)[30]. Wild-type PFKFB3 and PFKFB3$^{KEN-mut}$ (KEN box mutant PFKFB3) were gifts from J. P. Bolanos[2]. Transduced cells were sorted on the BD Biosciences FACS-Aria Fusion system to obtain pure populations expressing the desired fluorescent biosensors. The plasmids were co-transfected with the packaging plasmid (pCAG-HIVgp) and the VSV-G- and Rev-expressing plasmid (pCMV-VSV-G-RSV-Rev) into HEK293T cells. High-titre viral solutions of the target proteins were prepared and used for co-transduction into several cell lines. Then, 72 h later, cells were FACS sorted to obtain a pure population of cells expressing the biosensors against the protein of interest. For making the inducible system, gene bodies were cloned in pSB-Tet on system (Addgene, 60496). Cells were transfected with sleeping beauty transposon, Tet-on gene and selected using puromycin as the selection marker.

### Treatment of cells

Cells were treated with vehicle (DMSO, Sigma-Aldrich, D2650), 100 nM of rapamycin (Tocris Biosciences, 1292/1), 10 nM of RMC6272 (Medchem Express, HY-134904), 100 nM of PD0325901 (MEKi) (Selleckchem, S1036), 10 μM of RO3306 (Calbiochem, SIG217714), 1 μM of palbociclib (Selleckchem, S1116), 2.5 μM PFK15 (Selleck Chem, S7289), 5 mM 2-deoxy glucose (Sigma-Aldrich, D8375), D-glucose (U-$^{13}$C$_6$, 99%) (Cambridge Isotope Laboratories, CLM-1396-2), dialysed horse serum 10 kDa (Bioivt, HSE00SRM-0108261), 30 μM DRB18 (Cayman Chemical, 38217),

5–10 μM MG132 (Calbiochem, 133407-82-6) or a pan-phosphatase inhibitor (20 mg ml$^{-1}$ of sodium fluoride (Sigma-Aldrich, 7681-49-4) and 40 mg ml$^{-1}$ of sodium orthovanadate (Sigma-Aldrich, 13721-39-6)) for the indicated timepoints. For doxycycline induction of CDH1(T129A) or CDH1(T129D), cells were incubated with 5 μM doxycycline for 8 h in the starvation medium followed by mitogen stimulation with doxycycline. Similarly, for PFK15 and 2-DG treatment, cells were pretreated with respective drugs for 6 h in the starvation medium followed by mitogen stimulation with respective drugs. The cells were either collected after treatment and whole-cell extracts were prepared or imaging was continued. For experiments without glucose, cells were starved as described above with starvation medium, followed by replacement with starvation medium without glucose for the last 12–16 h. Cells were either stimulated with starvation medium containing 1,000 pg ml$^{-1}$ of EGF with or without glucose. For RPE-1 cells and MEFs, cells were starved for 48 h, followed by 12–16 h of starvation without glucose. Cells were stimulated to enter the cell cycle by replacing the starvation medium with 2% FBS containing DMEM-F12 or DMEM medium, respectively, with or without glucose.

### Antibodies

The following antibodies were used in this study; CDH1 (FZR1) antibody (Abcam, Ab217038, 1:1,000, IP: 2 μg; Santa Cruz, sc-56312, IB: 1:800; Sigma-Aldrich, CC43-100UG, 1:2,000), mTOR antibody (CST, 2972, 1:1,000), p-mTOR (CST, 2971, 1:1,000), mCherry (Abcam, ab167453, 1:1,000), geminin (CST, 5165, 1:1,000), cyclin D1 (Thermo Fisher Scientific, MA5 14512, 1:750), p-Rb (Ser807/811) (CST, 8516, 1:1,000), Rb (CST, 9309, 1:2,000), vinculin (Sigma-Aldrich, V9131, 1:10,000), Ki-67 (Abcam, ab8191, 1:1,000), p21 (BD Biosciences, 556430, 1:1,000), p27 (CST, 3686, 1:1,000), His tag (Santa Cruz, sc-8036, 1:800), anti-DDK/Flag (Sigma-Aldrich, F3165, 1:1,000), p-S6 kinase (CST, 9205, 1:1,000), APC2 (CST, 12301, 1:1,000), APC6 (CST, 9499, 1:1,000), APC11 (CST, 14090, 1:1,000), S6 kinase (CST, 9202, 1:1,000), p-4EBP1 (CST, 2855, 1:1000), cyclin A2 (Santa Cruz, sc-271682, 1:500), EMI1 (Santa Cruz, sc-365212, 1:500), cyclin F (Santa Cruz, sc-515207, 1:500), βTrCP (CST, 4394, 1:1,000), pSer/Thr/Tyr (Thermo Fisher Scientific, 61-8300, 1:1,000), pThr (Abcam, ab9337, 1:500), pTyr (Abcam, ab10321, 1:1,000), pan anti-phospho-serine/threonine antibody (Phospho Solutions, PP2551; 1:2,000), HA (Santa Cruz, sc-7392, 1:800, IP: 2 μg; and CST, 3724, 1:1,000), GST (Santa Cruz, sc-138, 1:750), PFKFB3 (MBS, 9604769, IB: 1:750, IP: 2 μg), PFKFB3 (Abcam, AB181861-1001, 1:4,000), PFKFB2 (CST, 13029, 1:1,000), PFKFB1 (Abcam, ab155564, 1:1,000), β-actin (Abcam, ab6276, 1:2,000), RPTOR (CST, 2280, 1:1,000), RICTOR (CST, 2114, 1:1,000), mLST8 (CST, 3274, 1:1,000), ubiquitin (Santa Cruz, sc-8017, 1:800), TSC1 (CST, 6935, 1:1,000), NPRL2 (CST, 37344, 1:1,000), histone H1 (Abcam, 11079, 1:1,000), EGFR (Santa Cruz, sc-373746, 1:800), AKT (CST, 9272, 1:1,000), p-AKT (CST, 4060, 1:1,000), mouse anti-goat IgG-HRP (Santa Cruz, sc-2354, 1:10,000), anti-rabbit IgG, HRP-linked antibody (CST, 7074, 1:10,000), mouse anti-rabbit secondary-HRP (Santa Cruz, sc-2357, 1:2,500) or recombinant anti-mouse (Santa Cruz, sc-516102, 1:2,500), anti-mouse IgG, HRP-linked antibody (CST, 7076, 1:10,000), normal rabbit IgG (CST, 2729, IP: 2 μg), normal mouse IgG (Santa Cruz, sc-2025, IP: 2 μg).

### siRNA transfection

The indicated cells were transfected using Dharmafect 1 (Horizon) according to the manufacturer's instructions. The following siRNAs were used: On-Target plus control siRNA (nontargeting, Dharmacon), On-Target plus pooled set of four siRNAs for *CCNA2* (M-003205-02-0005), *FZR1* (L-004086-00-0005), *EMI1* (M-012434-01-0005), *CCNF* (custom, 5′-GCACCCGGUUUAUCAGUAAUUUU-3′), *BTRC* (encoding βTrCP) (L-003463-00-0005 and L-003490-00-0005), *PFKFB3* (L-006763-00-0005), *TSC1* (L-003028-00-0005), *NPRL2* (L-015645-00-0005), *CDH1* 3′ UTR (Qiagen, SI04955265|S1 (1027417) at final concentrations of 20 nM unless noted. Then, 6 h after transfection,

the medium was replaced with starvation medium and imaging was started 24 h later.

## Time-lapse microscopy

Before imaging, cells were plated in a 96-well plate (Ibidi, 89626) and allowed to grow in full growth medium for 24 h. Time-lapse imaging was conducted in 200 µl full growth medium, with images captured in CFP, YFP and RFP channels every 12 min using a Nikon Ti2-E inverted microscope (Nikon) with NIS elements (V5.11.00) equipped with ×10/0.45 NA and ×20/0.75 NA Plan Apo objectives. To minimize phototoxicity, the total light exposure time was kept under 300 ms (30 ms for CFP, 200 ms for YFP and 300 ms for mCherry) for each timepoint. The cells were imaged in a humidified chamber at 37 °C with 5% $CO_2$. To ensure minimal overlap, four or six sites were imaged per well with their positions spaced apart. For experiments involving drug treatments, cells were first imaged without drugs, and the video was then paused to exchange fresh medium in each well with the desired drug concentration. For experiments with the HYlight biosensor, the following filter sets were used: Sapphire: Chroma ET395/25x excitation filter, Chroma ET525/50m emission filter, Chroma T425lpxr dichroic; GFP: Chroma ET480/20X excitation filter, Chroma ET510/20 m emission filter, Chroma T495lpxr dichroic. Cell tracking and data analysis were performed using custom MATLAB scripts.

## Cell lysate preparation and immunoblotting

The cells were washed twice with ice-cold PBS and collected. Whole-cell lysis buffer (50 mM Tris pH 7.4, 200 mM NaCl, 50 mM NaF, 1 mM $Na_3VO_4$, 0.5% Triton X-100 and protease inhibitor cocktail) was added to the cells, and the cells were lysed on ice for 30 min. The lysates were centrifuged at high speed (16,000$g$), and the clear supernatants were transferred into new tubes. The protein concentration was measured using the BCA method, with BSA used as a standard. The samples were prepared in SDS sample buffer and run in SDS–PAGE running buffer (Bio-Rad, 1610772). The separated proteins were transferred onto a PVDF membrane using the Bio-Rad semidry transfer system (Bio-Rad, 10026938). The membranes were incubated with the primary antibody overnight at 4 °C and subsequently with an HRP-conjugated secondary antibody for 1 h at room temperature. The blots were developed using the chemiluminescence method, and densitometry analysis of the immunoblots was performed using ImageJ software.

## Immunoprecipitation

The cells were lysed as described above (see the 'Cell lysate preparation and immunoblotting' section). A total of 300–500 µg of whole-cell lysate was co-immunoprecipitated with 2 µg of antibody and IgG control in 500 µl of modified IP lysis buffer (50 mM Tris pH 7.4, 200 mM NaCl, 50 mM NaF, 1 mM $Na_3VO_4$, 0.1% Triton X-100 and protease inhibitor cocktail). The protein–antibody mixture was kept at 4 °C in a rotor with gentle rocking for 12–16 h. The next day, the mixture was allowed to bind to protein G agarose beads for 2 h at 4 °C with gentle rocking. The immunoprecipitated proteins were eluted from the beads using Laemmli buffer for 3–5 min and boiled before resolving on SDS–PAGE. In all immunoprecipitation assays, 3% of the proteins taken in the immunoprecipitation experiment were used as input.

For co-IP of purified protein, 200 ng of His–CDH1 was allowed to bind to the beads overnight followed by washing and addition of purified mTOR. The whole complex was kept at 4 °C with rotation for 1 h followed by washing and elution using 1× SDS-Laemmli buffer. For untagged purified protein co-IPs, magnetic Flag beads (A36797, Thermo Fisher Scientific) were used and DDK–mTOR (Sigma-Aldrich, SRP0364)/ DDK–DYRK4 (Carna Biosciences, 04-434) was allowed to bind to the beads overnight at 4 °C followed by washing and addition of purified WT or mutant CDH1 or His–SUMO–CDH1 (Mybiosource, MBS1437931). The whole complex was kept at 4 °C with rotation for 1 h followed by washing and elution using 1× SDS-Laemmli buffer.

## In vitro kinase assay

A total of 200 ng of untagged human CDH1 or His–SUMO–CDH1 purified protein was incubated with 200 ng of DDK-tagged human mTOR protein (Carna Biosciences or Sigma-Aldrich), GST–CDK1–cyclin B (Carna Biosciences, 04-102), GST–ERK (Reaction Biology, 0883-0000-1) or Flag–DYRK4 (Carna Biosciences, 04-434) in 1× kinase buffer (Cell Signaling Technology, 9802). The assay was performed with 100 µM of ATP (Cell Signaling Technology, 9804) at 37 °C for 30 min, followed by addition of 4×-reducing SDS sample buffer and heating at 95 °C for 10 min. The reaction was resolved by SDS–PAGE and immunoblotting was performed as stated above.

## Ubiquitination assay

Before collection, cells were treated with 10 µM of MG132 for 3–4 h (except where otherwise mentioned). After collection, whole-cell lysates were prepared and 300–500 µg of lysate was subjected to immunoprecipitation using the indicated antibodies. The immunoprecipitate obtained was separated using SDS–PAGE and probed with anti-ubiquitin antibodies to determine the levels of ubiquitylation.

## In vitro phosphorylation of CDH1 for ubiquitination assay

CDH1 purified from baculoviral infected insect cells was subjected to phosphorylation by recombinant Flag–mTOR + mLST8 (Carna Biosciences, 11-431). Then, 1 µM CDH1 WT, T129A or T129D was mixed with buffer or 10 ng µl$^{-1}$ mTOR and 200 µM of $MgCl_2$-ATP for 30 min at 30 °C, after which the reactions were stored on ice. All of the reactions (control or containing mTOR) were treated equally). All further steps were performed either on ice or at 4 °C unless otherwise stated. The reactions were quenched by removal of ATP through desalting with a Zeba Spin column (Thermo Fisher Scientific, PI89882) into 20 mM HEPES pH 7.0, 400 mM AmSO$_4$, 100 mM NaCl with or without 2.5% glycerol according to the manufacturer's specifications. Flag–mTOR was removed from the reactions, desalted into glycerol-containing buffer through addition of 40 µl of 50:50 Flag-resin slurry (GenScript, L00432) equilibrated with glycerol-containing buffer. The slurry was rotated in batch for 1 h on ice, then Flag resin was pelleted by centrifugation at 1,500$g$. The supernatant was removed and used for further reactions. The concentration of reacted CDH1 was checked by Nanodrop and confirmed with Coomassie staining. Phosphorylation was confirmed by western blotting with pan-anti-phospho-serine/threonine antibody (Phospho Solutions, PP2551; 1:2,000) or CDH1 (Sigma-Aldrich, CC43-100UG; 1:2,000), mouse anti-rabbit secondary-HRP (Santa Cruz, sc-2357; 1:2,500) or recombinant anti-mouse (Santa Cruz, sc-516102; 1:2,500) and visualized with Clarity ECL (Bio-Rad, 1705060).

## In vitro ubiquitination assays

Ubiquitination assays were performed with recombinant APC/C, UBE2C, UBA1, substrates, ubiquitin and CDH1 as previously described[38,39]. Then, 100 nM of fluorescently labelled cyclin A2 (FAM-cyclin A) or the N-terminal domain (NTD) of cyclin B (FAM-cyclin B$^{NTD}$) was incubated on ice with 100 nM UBA1, 300 nM UBE2C, 5 mM Mg-ATP, 30 nM APC/C and 30 nM CDH1. The components were equilibrated to room temperature for 5 min before reactions were started with addition of 100 µM ubiquitin. The reactions were quenched after 20 min with 4× SDS–PAGE loading buffer, separated on 4–12% Bis-Tris gels, and imaged on the Amersham Typhoon imager. For in vitro ubiquitination assays with recombinant phosphorylated CDH1, 40 nM CDH1 was mixed with 30 nM APC/C, 100 nM UBA1, 300 nM UBE2C, 250 nM CycB-NTD* and 5 mM $MgCl_2$-ATP on ice. The reaction was allowed to come to room temperature, and 100 µM Ub was added. The reactions were quenched after 20 min with 4× SDS buffer. Proteins were separated on 4–12% Bis-Tris SDS–PAGE gels (GenScript, M00654) and CycB-NTD* ubiquitination was visualized on the Typhoon fluorescence scanner. Ubiquitination was quantified in ImageQuant software and normalized to CDH1 WT without mTOR.

## Image analysis

Custom MATLAB scripts were used for all image analyses, according to previously described methods[8]. In brief, optical illumination bias was empirically determined by sampling background areas in all wells during an imaging session, and this information was then used to flatten all images. This enabled a global background to be measured and subtracted from each image. Cells were segmented based on their nuclei using either Hoechst staining for fixed-cell imaging or H2B–mTurquoise for live-cell imaging. The procedure for determining APC/C and CDK2 activity was previously reported[13].

## OPP assay

Cells were seeded and starved for 48 h. After 24 h of the starvation, empty vector, CDH1(T129A) or CDH1(T129D) mutant was ectopically expressed in the cells and after 6 h of transfection medium was replaced with starvation medium. Cells were released as described above and incubated with 20 μM of *O*-propargylpuromycin (OPP) for 30 min followed by fixation with 4% paraformaldehyde for 15 min at room temperature in the dark. Further procedures were performed according to the manufacturer's instructions (Thermo Fisher Scientific, C10458).

## Glycolysis and ATP rate measurements by extracellular flux assays

To measure glycolytic function of cells, the Glycolysis Stress Test Kit (Agilent technologies, 103020-100) was used. To measure the glycolytic bioenergetics of cells, the Glycolytic Rate Assay Kit (Agilent technologies, 103344-100) was used. To distinguish between the fraction of ATP produced from mitochondrial OXPHOS and glycolysis, the ATP Rate Assay Kit (Agilent technologies, 103591-100) was used. For all of the assays, cells were initially seeded at a density of 10,000 cells per well in an XF96-well plate precoated with poly-lysine. For ectopic overexpression or silencing, Lipofectamine 3000 or Dharmafect 1 (Horizon) were used to transfect cells according to the manufacturer's instructions. After 24 h of transfection, cells were released with complete medium with or without drugs for another 3 h. After release, for the glycolysis stress test, cells were washed and incubated with Seahorse XF DMEM medium, pH 7.4 (Agilent Technologies, 103575-100) supplemented with 2 mM glutamine. Glucose, oligomycin and 2-DG were added to cells sequentially at the specific timepoint with final concentrations of 10 mM, 1 μM and 50 mM, respectively. For the glycolytic rate assay, cells were washed and incubated with Seahorse XF DMEM medium, pH 7.4 (Agilent Technologies, 103575-100) supplemented with 10 mM glucose, 1 mM pyruvate and 2 mM glutamine. Antimycin/rotenone and 2-DG were added to cells sequentially at the specific timepoint with final concentrations of 0.5 μM and 50 mM, respectively. For the ATP rate test, cells were washed and incubated with Seahorse XF DMEM medium, pH 7.4 (Agilent Technologies, 103575-100) supplemented with 10 mM glucose, 1 mM pyruvate and 2 mM glutamine. Oligomycin and antimycin/rotenone were applied with final concentrations of 1.0 μM and 0.5 μM, respectively. The oxygen-consumption rate and ECAR was collected using the Agilent Seahorse Wave 2.6.1 desktop software. Data were normalized to the cell number determined using the crystal violet assay and analysed using the Wave software.

## ATP measurements

Cells were seeded at a density of 5,000 cells per well in a 96-well plate and incubated with complete growth medium for 24 h. The medium was then replaced with starvation medium and incubated for another 24 h. The cells were then transfected with either siControl or siPFKFB3 (using Dharmafect, Horizon) or CDH1(T129A) and CDH1(T129D) (using Lipofectamine 3000, Invitrogen). Then, 6 h after transfection, the medium was replaced with starvation medium. After another 24 h, the cells were mitogen-stimulated to promote cell cycle entry. Then, 4 h after stimulation, the medium was replaced with chilled PBS and the sample was kept on an iron slab on ice to stop or slow down the metabolic turnover[40,41]. The cells were collected and the assays were performed according to the manufacturer's protocols (Abcam, ab113849).

## qPCR with reverse transcription

Total RNA was extracted from cultured cells using the RLT RNeasy 96 Qiacube HT Kit (Qiagen, 74171) and on-column DNA digestion was performed using the Qiagen DNA digestion kit (79254). The cDNA was synthesized using the Bio-Rad cDNA preparation kit according to the manufacturer's instructions, starting with 1 μg of total RNA. Quantitative PCR (qPCR) was performed using the Bio-Rad SYBR green qPCR Mix. The expression of *GAPDH* was used as a reference gene to normalize the mRNA level of the gene of interest. The relative mRNA levels were determined by comparing the treated or transfected samples to the untreated or vector control, which was considered as 1. The following primers were used: *GAPDH* forward, AATCCCATCACCATCTTCCA; *GAPDH* reverse, TGGACTCCACGACGTACTCA; *PFKFB3* forward, GGTACC GAATCAAGCAGAGC; *PFKFB3* reverse, GCAGTAGGAGGACGAGTTGG.

## Mathematical model

The incoherent feedforward model involving mTOR and a delayed protein phosphatase activity towards the APC/C was mathematically modelled using a set of ordinary differential equations (ODEs) as shown below:

(1) $dMTOR/dt = 1/\tau_{mTOR}(S^{(n_1)}/(k_1^{(n_1)} + S^{(n_1)}) - mTOR)$

(2) $dPP/dt = 1/\tau_{PP}(S^{(n_2)}/(k_2^{(n_2)} + S^{(n_2)}) - PP)$

(3) $dpCDH1/dt = 1/\tau_{pCDH1}((mTOR^{(n_3)}/(k_3^{(n_3)} + mTOR^{(n_3)}))(k_4^{(n_4)}/(k_4^{(n_4)} + PP^{(n_4)})) - pCDH1)$

(4) $dAPC/dt = 1/\tau_{APC}(k_5^{(n_5)}/(k_5^{(n_5)} + pCDH1^{(n_5)}) - APC)$

(5) $dGEMININ/dt = 1/\tau_{GEMININ}(k_6^{(n_6)}/(k_6^{(n_6)} + APC^{(n_6)}) - GEMININ)$.

The parameters for the model were as follows: $k_1 = 0.2, n_1 = 1, \tau_{mTOR} = 0.5$; $k_2 = 0.2, n_2 = 1, \tau_{PP} = 0.5$; $k_3 = 0.3, n_3 = 1, \tau_{pCDH1} = 0.5$; $k_4 = 0.2, n_4 = 2, \tau_{APC} = 0.5$; $k_5 = 0.8, n_5 = 1, \tau_{GEMININ} = 0.3$; $k_6 = 0.4, n_6 = 3$.

The initial conditions for the model were as follows: mTOR = 0, PP = 0, pCDH1 = 0, APC = 1, GEMININ = 0.

The following notation is used: S, mitogens (for example, serum); mTOR, mTOR activity; PP, protein phosphatase activity; pCDH1, phosphorylated CDH1; APC, APC/C activity; GEMININ, APC/C substrate levels (for example, geminin).

Each system of ODEs was solved in RStudio (v.1.3.1093) using the ode function (from the deSolve R package) using the LSODA algorithm. Most parameters were chosen on the basis of previous models[9,42], but mTOR and phosphatase activation kinetics were chosen to match data from this study. To model the effect of mitogen stimulation, the initial conditions of the model were set as listed above and the model was solved for S = 3. To model the delay in protein phosphatase activity as measured experimentally, the PP term was kept at zero until 4 h after mitogen stimulation.

## In vitro purification of CDH1

Baculovirus-induced expression of 3×Myc–6×His–CDH1 wild type and variants was performed by infecting *Trichoplusia ni* cells in ESF921 medium (Expression Systems). All purification steps were completed at 4 °C. Cells were collected about 2.5 days later, pelleted and resuspended in 20 ml of lysis buffer (20 mM HEPES pH 8.0, 100 mM AmSO₄, 2.5% glycerol, 10 mM imidazole supplemented with 10 μg ml⁻¹ leupeptin, 20 μg ml⁻¹ aprotinin, 5 U ml⁻¹ benzonase, 2 mM benzamidine, 2.5 mM PMSF and 1 Roche EDTA-free protease inhibitor tablet per 50 ml buffer) per litre. Cells were lysed by sonication and clarified by centrifugation at 30,000$g$ for 1 h. The lysates were incubated in batch with Ni-NTA resin (Genesee Scientific) for 1 h before centrifugation of beads at 500$g$ for 10 min. After resuspending and washing the resin with 10 column volumes of wash buffer (20 mM HEPES pH 7.0, 100 mM AmSO₄, 2.5% glycerol and 10 mM imidazole), CDH1 was eluted with wash buffer

supplemented with 300 mM AmSO$_4$ and 250 mM imidazole. The affinity tags were removed by the addition of HRV-3C protease for 1 h on ice. The protein solution was diluted back to 100 mM AmSO$_4$ with wash buffer lacking salt or imidazole and further purified by cation-exchange chromatography with SP-Sepharose resin (Thermo Fisher Scientific). Protein was eluted with 20 mM HEPES pH 7.0, 100 mM AmSO$_4$, 400 mM NaCl, 2.5% glycerol and 2 mM dithiothreitol and flash frozen with liquid nitrogen in small aliquots.

### Protein digestion
Recombinant protein (500 ng) was digested by Arg-C (Promega) at a ratio of 1:50 (Promega) and incubated overnight at 37 °C. The digested peptide samples were acidified by formic acid to a final concentration of 1% and desalted using Pierce peptide desalting columns according to the manufacturer's protocol (Thermo Fisher Scientific). Peptides were eluted from the columns using 50% acetonitrile/0.1% formic acid, vacuum centrifuged to dry and stored at −80 °C until analysed by MS.

### MS acquisition and data analysis
Dried peptide fractions were reconstituted in 0.1% TFA and subjected to nanoflow liquid chromatography (Thermo Ultimate 3000 RSLC nano LC system, Thermo Fisher Scientific) coupled to an Orbitrap Eclipse mass spectrometer (Thermo Fisher Scientific). Peptides were separated using a low-pH gradient using a 5–50% acetonitrile over 120 min in mobile phase containing 0.1% formic acid at a flow rate of 300 nl min$^{-1}$. Full MS1 scans were performed in the Orbitrap at a resolution of 120,000 with an ion accumulation target set at $4 \times 10^5$ and a max IT set at 50 ms over a mass range of 400–1,600 $m/z$. Ions with determined charge states between 2 and 5 were selected for MS2 scans in the orbitrap with HCD fragmentation (NCE 30%; maximum injection time, 22 ms; AGC $5 \times 10^4$) at a resolution of 15,000.

Acquired MS/MS spectra were searched against FZK-Human fasta along with a contaminant protein database, using a SEQUEST and Fixed Value PSM Validator in the Proteome Discoverer v.2.4 software (Thermo Fisher Scientific). The precursor ion tolerance was set at 10 ppm and the fragment ions tolerance was set at 0.02 Da along with methionine oxidation and phosphorylation of serine, threonine and tyrosine included as dynamic modification. Arg-C was specified as the proteolytic enzyme, with up to two missed cleavage sites allowed. The HCD fragmentation pattern of the peptide [KGLFT*YSLSTKR] is shown. The red colour peaks shown in Extended Data Fig. 5 represent the matched b+ ion from the peptide. The blue colour peaks represent the y+ ions from the fragmented peptide. Peptide fragmentation ladder confirms the sequence of the peptide along with phosphorylation site of Thr129. Furthermore, the neutral loss of the labile phosphate group from Thr129 confirms the phosphorylation site, shown as the yellow highlighted peak in the spectra.

### Sample preparation for metabolomics
Cells were serum-starved for 48 h to induce quiescence. After starvation, cells were transfected with either empty vector or CDH1(T129A). The medium was replaced with fresh starvation medium at 6 h after transfection. Cells were washed twice with 1× PBS 24 h after transfection and the medium was replaced with glucose-free starvation medium for 1 h. Cells were then incubated with either standard starvation medium or mitogen-rich medium containing dialysed horse serum supplemented with labelled glucose (Cambridge Isotope Laboratories, CLM-1396-2). Cells were washed with 1× PBS, placed on ice, and collected by scraping followed by quick freezing in liquid nitrogen for subsequent analysis.

### Reversed-phase ion-pairing LC–MS/MS analysis for cell $^{13}C_6$-glucose isotope tracing
The unlabelled glycolytic metabolite reference compounds were purchased from Sigma-Aldrich. OmniSolv LC–MS grade acetonitrile and methanol were obtained from EMD Millipore. Tributylamine (TBA), LC–MS grade acetic acid and formic acid were purchased from Thermo Fisher Scientific. All chemicals and solvents used in this study were HPLC or reagent grade unless otherwise noted.

Metabolite extraction was performed by adding 200 µl chilled 80% methanol–water solution to the cell pellet as previously described[43]. The sample was vortexed vigorously for 30 s and centrifuged at 14,000g for 10 min. Then, 50 µl supernatant was transferred to an autosampler vial. The sample was dried with the SpeedVac vacuum concentrator (Thermo Fisher Scientific) and then reconstituted in 60 µl 3% (v/v) methanol in water. Then, 10 µl sample was injected for reversed-phase ion-pairing LC–MS/MS analysis. The LC–MS/MS analysis was performed using the Thermo TSQ Quantiva triple quadrupole mass spectrometer (Thermo Fisher Scientific) coupled to the NexeraXR LC system (Shimadzu Scientific Instruments). Both the HPLC and mass spectrometer were controlled by the Xcalibur software v.4.1 (Thermo Fisher Scientific). Reversed-phase ion-pairing liquid chromatography was carried out on a 100-mm-long, 2.1-mm-inner-diamter Synergi Hydro-RP C18 column with 2.5 µm particles and a 100 Å pore size (Phenomenex) and kept at 40 °C. The mobile phase, operating at a flow rate of 200 µl min$^{-1}$, consisted of 10 mM TBAA in water as solvent A and methanol as solvent B. For the current analysis, a linear gradient was held at a B/A solvent ratio 3:97 for 3 min, followed by a B/A solvent ratio of 80:20 for 14 min. After washing with 98% B for 3 min, the column was re-equilibrated with a mobile-phase composition B/A of 3:97 for 10 min before the next injection. The general MS conditions were as follows: source: ESI; ion polarity: negative; spray voltage: 2,500 V; sheath and auxiliary gas: nitrogen; sheath gas pressure: 40 arbitrary units; auxiliary gas pressure: 5 arbitrary units; ion transfer capillary temperature, 350 °C; scan type: selected reaction monitoring (SRM); collision gas: argon; collision gas pressure: 2 mTorr. The $^{13}C_6$-glucose isotope tracing of the targeted cell glycolysis isotopomer/isotopolog analysis peak identifications and integrations was carried out using Xcalibur Quan Browser (Thermo Fisher Scientific). The SRM conditions and natural isotope abundance corrections analyses were adapted from the previous publications[44,45]. The final peak area data were normalized to the sample protein concentration.

### Statistical analysis and reproducibility
Statistical analyses were performed in MATLAB (MathWorks, vR2020b) and Prism (GraphPad 9, v.9.2.0). Specific statistical tests used are noted in the figure legends. No statistical method was used to determine sample size. No data were excluded from the analyses. The experiments were not randomized, but unbiased, automated analysis was used to ensure that observer bias did not influence the experimental results.

### Reporting summary
Further information on research design is available in the Nature Portfolio Reporting Summary linked to this article.

## Data availability
All data supporting the findings of this study are available from the corresponding author on reasonable request. Data from the glucose tracing experiments are available at Massive with the unique identifier MSV000098175. Source data are provided with this paper.

## Code availability
All original code has been deposited at GitHub (https://github.com/scappell/Cell_tracking) and is publicly available.

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

**Acknowledgements** We thank the members of the Flow Cytometry Core Facility of the Center for Cancer Research at the National Cancer Institute (NCI) for technical support; M. Koivomagi (NCI) for providing us with GST–ERK1 (Reaction Biology, 0883-0000-1) and H1 (Millipore Sigma, 14-155) purified protein; J. P. Bolanos (IBFG) for providing WT and KEN box mutant PFKFB3 construct; J. Vidigal (NCI) for providing the MEF cell line; M. Santra (NCCS, Pune) for providing the HA–CDH1 construct; and all of the members of the Cappell laboratory for comments and support. This research was supported by the Intramural Research Program of the NIH (grant ZIA BC 011830 to S.D.C.) and the Extramural Research Program of the NIH (grant T32GM008570 to D.L.B. and grant R35GM128855 and UCRF to N.G.B.).

**Author contributions** D.P. and S.D.C. conceived the project and designed all of the experiments. D.P., D.L.B., H.Y., C.C.A., X.X., S.D., L.M.M.J. and S.D.C. performed all of the experiments and analysed data. D.P. performed all of the live-cell imaging, IPs, in vitro kinase assays, western blots and colony-forming assays. D.L.B., M.J.E. and N.G.B. designed and performed the in vitro APC/C ubiquitination assays. C.C.A. and J.G.A. designed and performed the HYlight experiments. L.M.M.J., S.D. and T.A. designed and performed the MS experiments. D.P., H.Y. and J.H. designed and performed the Seahorse-based assays. D.P., X.X. and T.A. designed and performed the glucose tracing experiments. D.P. and S.D.C. wrote the manuscript with help from all of the authors. S.D.C. supervised and funded the project.

**Competing interests** The authors declare no competing interests.

**Additional information**
**Correspondence and requests for materials** should be addressed to Steven D. Cappell.

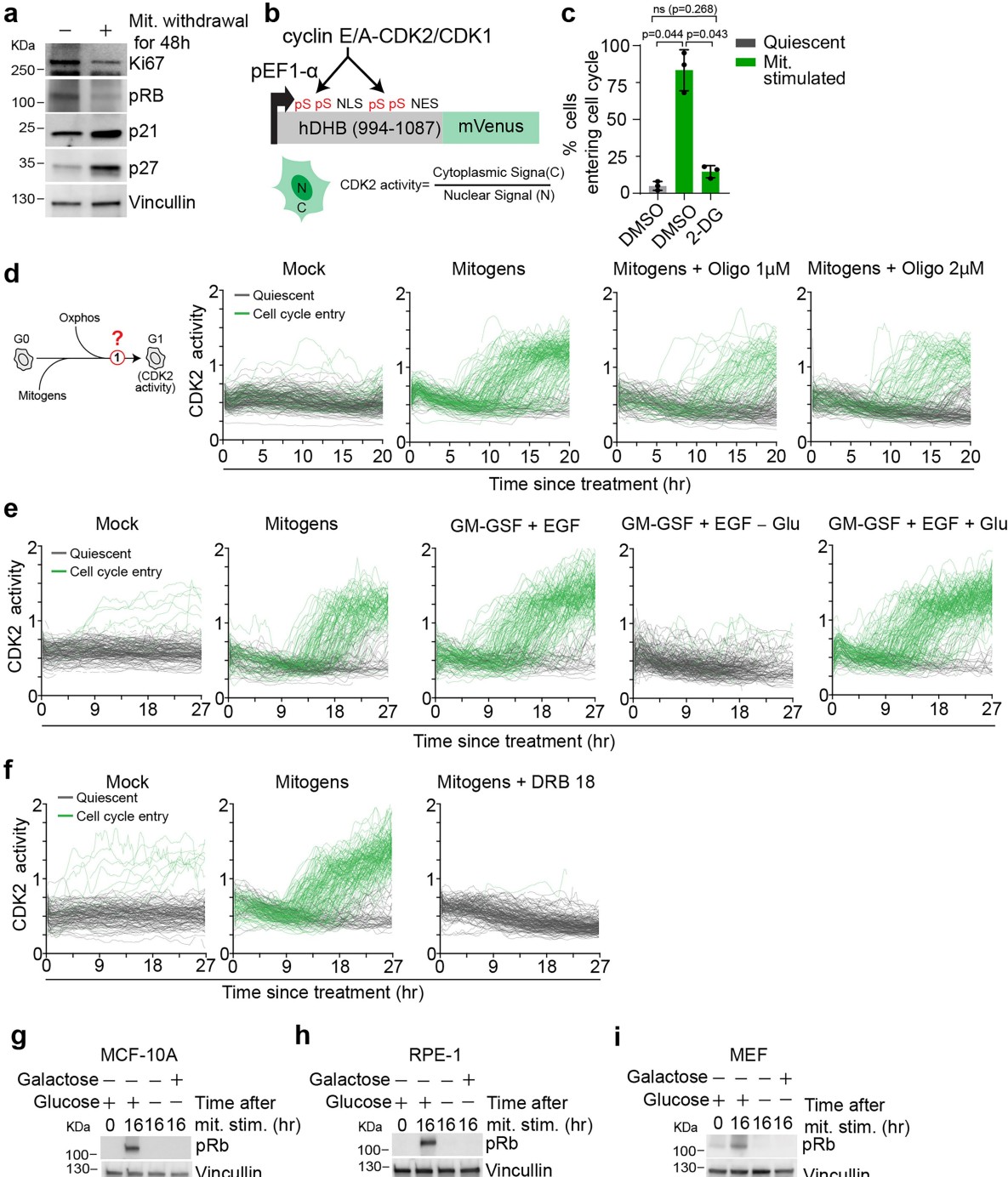

**Extended Data Fig. 1 | See next page for caption.**

**Extended Data Fig. 1 | Glycolysis is required for cell cycle entry.**
**a**, Immunoblot analysis of the indicated proteins from either cycling or mitogen starved MCF-10A cells. Representative blot of n = 3 independent experiments. **b**, Schematic of the cyclin E/A-CDK1/2 (CDK2) biosensor. A fragment of human DNA helicase B (amino acids 994-1087) fused to mVenus under the control of the constitutive promoter EF1α. **c**, MCF-10A cells were starved for 48 h to induce quiescence followed by mitogen stimulation with DMSO or 2-deoxy glucose (2-DG). Percent of cells turning CDK2 on were quantified. Data are mean ± SD from n = 3 independent experiments. P-values were calculated using a one-way ANOVA. ns, Not significant. **d**, Experiment to test whether OXPHOS is needed to start the cell cycle (left). CDK2 activity traces of single cells for the indicated treatments since the time of mitogen addition (right). MCF-10A cells were starved for 48 h to induce quiescence followed by mitogen stimulation with or without the addition of oligomycin (Oligo) at the indicated concentration. Green traces depict cells that entered the cell cycle. Grey traces depict cells that remain quiescent and failed to enter cell cycle. N = 200 cells per condition. **e**, Experiment to test whether glucose is needed to start the cell cycle. CDK2 activity traces of single cells for the indicated treatments since the time of mitogen or EGF addition. MCF-10A cells were starved for 48 h to induce quiescence by switching the growth media to growth media minus growth factors (GM-GFS). At time zero, the media was switched to either full growth media, GM-GFS supplemented with EGF and with and without glucose (Glu). Green traces depict cells that entered the cell cycle. Grey traces depict cells that remain quiescent and failed to enter cell cycle. N = 200 cells per condition. **f**, Experiment to test whether glucose transport is needed to start the cell cycle. CDK2 activity traces of single cells for the indicated treatments since the time of mitogen addition. MCF-10A cells were starved for 48 h to induce quiescence followed by mitogen stimulation with or without the addition of 30 μM DRB18, a pan GLUT inhibitor. Green traces depict cells that entered the cell cycle. Grey traces depict cells that remain quiescent and failed to enter cell cycle. N = 200 cells per condition. **g-i**, MCF-10A (g), retinal pigment epithelial cells (RPE-1) (h), or mouse embryonic fibroblasts (MEF) (i) were grown in serum free medium for 48 h to induce quiescence. At time 0 h, cells were stimulated with growth medium containing 5% fetal bovine serum with either glucose, galactose, or neither. After 16 h, cells were harvested and cell lysates were subjected to immunoblot analysis for phosphorylated Rb, a marker for cell cycle entry. Representative blot of n = 2 independent experiments.

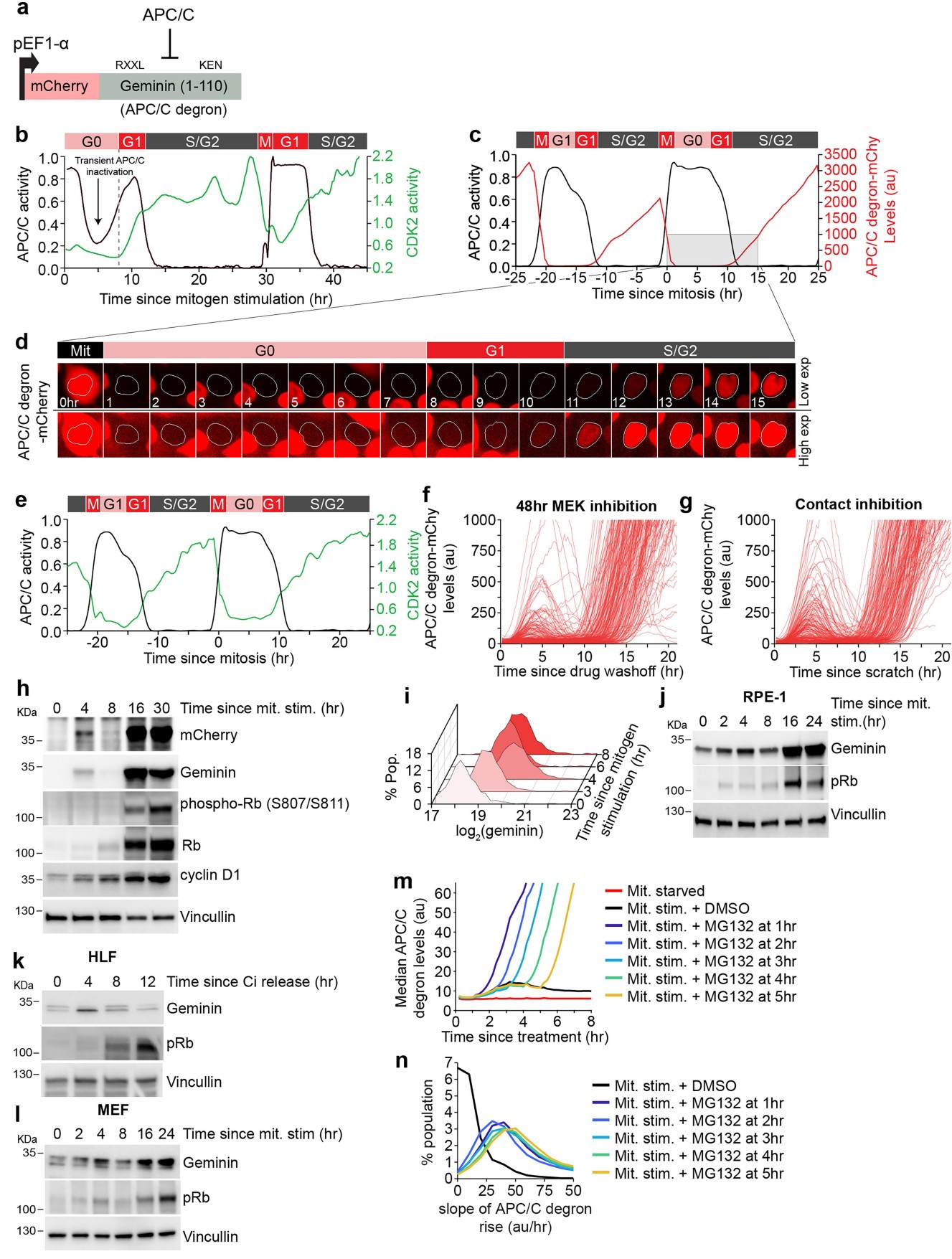

**Extended Data Fig. 2 |** See next page for caption.

**Extended Data Fig. 2 | APC/C transiently inactivates during cell cycle entry.**
**a**, Schematic of the APC/C biosensor. A fragment of human geminin (amino acids 1-110) fused to mCherry under the control of the constitutive promoter EF1α. **b**, Representative trace of APC/C activity (black line) and CDK2 activity (green line) of a single MCF-10A cell entering the cell cycle from quiescence. Cells were starved for 48 h to induce quiescence, followed by mitogen stimulation to promote cell cycle entry. Note the transient dip in APC/C activity during early G0 phase as noted by the arrow. **c**, Representative trace of APC/C biosensor (red line) and APC/C activity (black line) of a single cycling MCF-10A cell. Note there is no transient accumulation of the APC/C biosensor or a dip in APC/C activity during transient G0 phase of the cell cycle. **d**, Single-cell image montage of the APC/C biosensor post-mitosis going through a transient G0 phase as trace shown in (c). **e**, Representative trace of APC/C activity (black line) and CDK2 activity (green line) of the same MCF-10A cell as in (c). **f**, **g**, MCF-10A cells were treated with MEK inhibitor (f) or contact inhibited (g) for 48 h to induce quiescence followed by removal of perturbation to promote cell cycle entry. Single-cell APC/C biosensor traces were plotted. N = 200 cells for each condition. **h**, MCF-10A cells were starved for 48 h to induce quiescence followed by mitogen stimulation to promote cell cycle entry and were collected at the indicated time points. Whole cell lysates were resolved on SDS-PAGE and probed for the indicated proteins. Representative blot of n = 3 independent experiments. **i**, MCF-10A cells were starved for 48 h to induce quiescence followed by mitogen stimulation for indicated time points, fixed, immunostained with a geminin antibody, and imaged using a fluorescent microscope. Geminin levels were quantified, and data represent histograms showing the relative geminin expression in MCF-10A cells at indicated time points. **j**, RPE-1 cells were starved for 48 h to induce quiescence followed by mitogen stimulation to promote cell cycle entry and were collected at the indicated time points. Whole cell lysates were resolved on SDS-PAGE and probed for the indicated proteins. Representative blot of n = 3 independent experiments. **k**, **l**, Human lung fibroblasts (k) or mouse embryonic fibroblasts (l) were either contact inhibited or starved for 48 h or 72 h to induce quiescence followed by contact inhibition release or mitogen stimulation to promote cell cycle entry and whole cell lysates were collected at the indicated time points. Representative blot of n = 3 independent experiments. **m**, Median traces of APC/C biosensor levels from cells treated with or without 10 μM MG132 at the indicated time. Cells were starved for 48 h to induce quiescence, followed by mitogen stimulation to promote cell cycle entry. The slope of the APC/C biosensor accumulation after MG132 addition reflects the synthesis rate of the biosensor. **n**, Histograms of the APC/C biosensor accumulation slopes from single-cells after either mitogen stimulation (black line) or mitogen stimulation plus MG132 at the indicated time from (m).

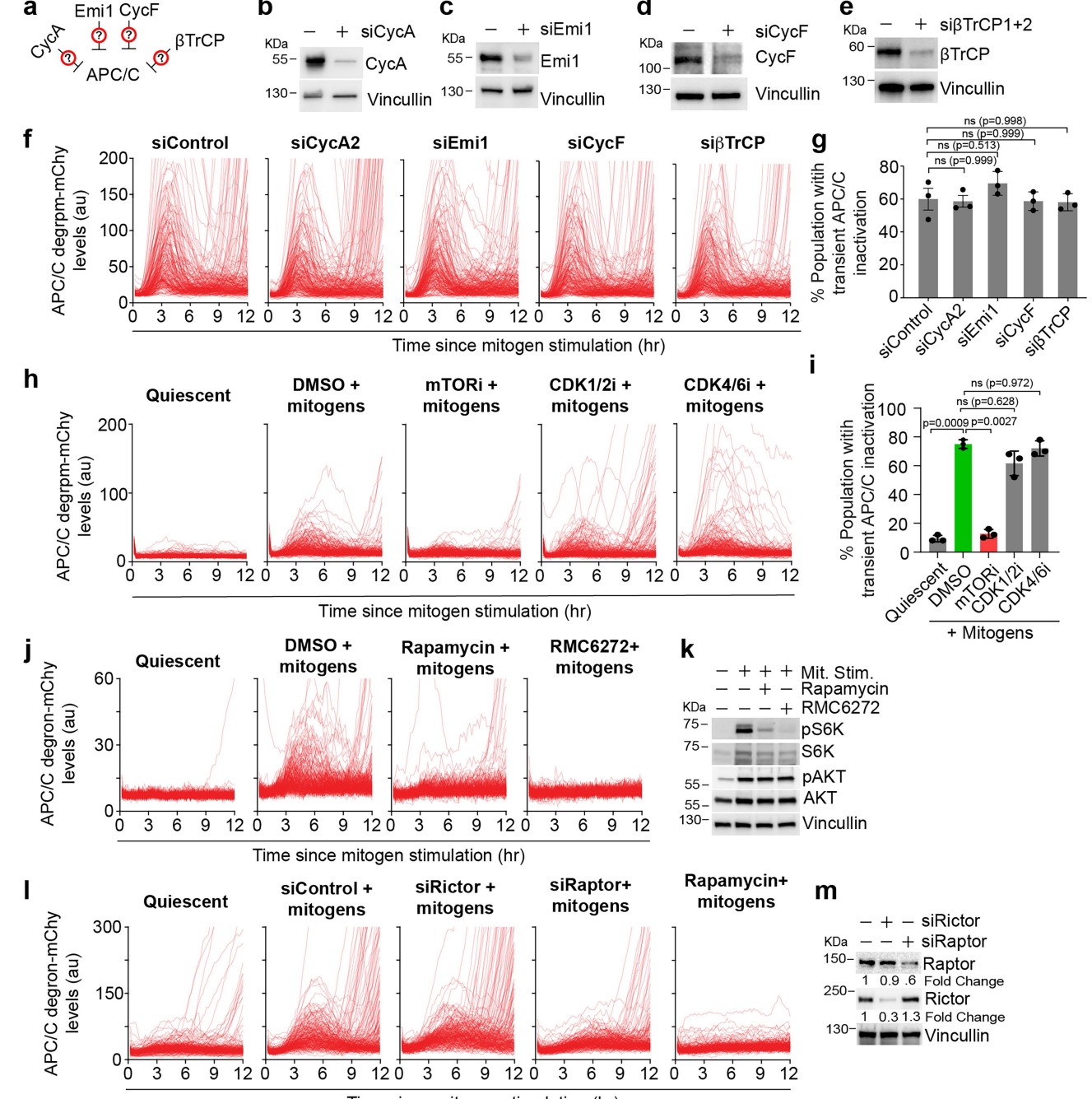

**Extended Data Fig. 3 | mTORC1 regulates transient APC/C inactivation.**
**a**, Schematic illustrating known negative regulators of APC/C^Cdh1 and their possible involvement in transient APC/C inactivation. **b-e** Immunoblot validation of cyclin A2, Emi1, Cyclin F, and βTrCP knockdown using siRNA. **f**, Single cell APC/C biosensor traces of MCF-10A cells entering the cell cycle transfected with the indicated siRNA. N = 200 cells per condition. **g**, Quantification of the percent of MCF-10A cells transiently inactivating APC/C during cell cycle entry as in (f). Data are mean ± SD from n = 3 independent experiments. P-values were calculated using a one-way ANOVA. ns, Not significant. **h**, Single cell APC/C biosensor traces of MCF-10A cells entering cell cycle treated with either DMSO or the indicated kinase inhibitors. N = 200 cells per condition. **i**, Quantification of the percent of MCF-10A cells transiently inactivating APC/C during cell cycle entry as in (h). Data are mean ± SD from n = 3 independent experiments. P-values were calculated using a one-way ANOVA. ns, Not significant. **j**, Single cell APC/C biosensor traces of MCF-10A cells entering cell cycle treated with either DMSO or the indicated mTOR inhibitors. N = 200 cells per condition. **k**, Immunoblot validation of mTOR inhibitors used in (**j**). **l**, Single cell APC/C biosensor traces of MCF-10A cells entering cell cycle treated with either control siRNA or siRNA targeting *RICTOR* or *RPTOR*. Cells were treated with the mTOR inhibitor Rapamycin as a positive control. N = 200 cells per condition. **m**, Immunoblot validation of Raptor and Rictor knockdown using siRNA.

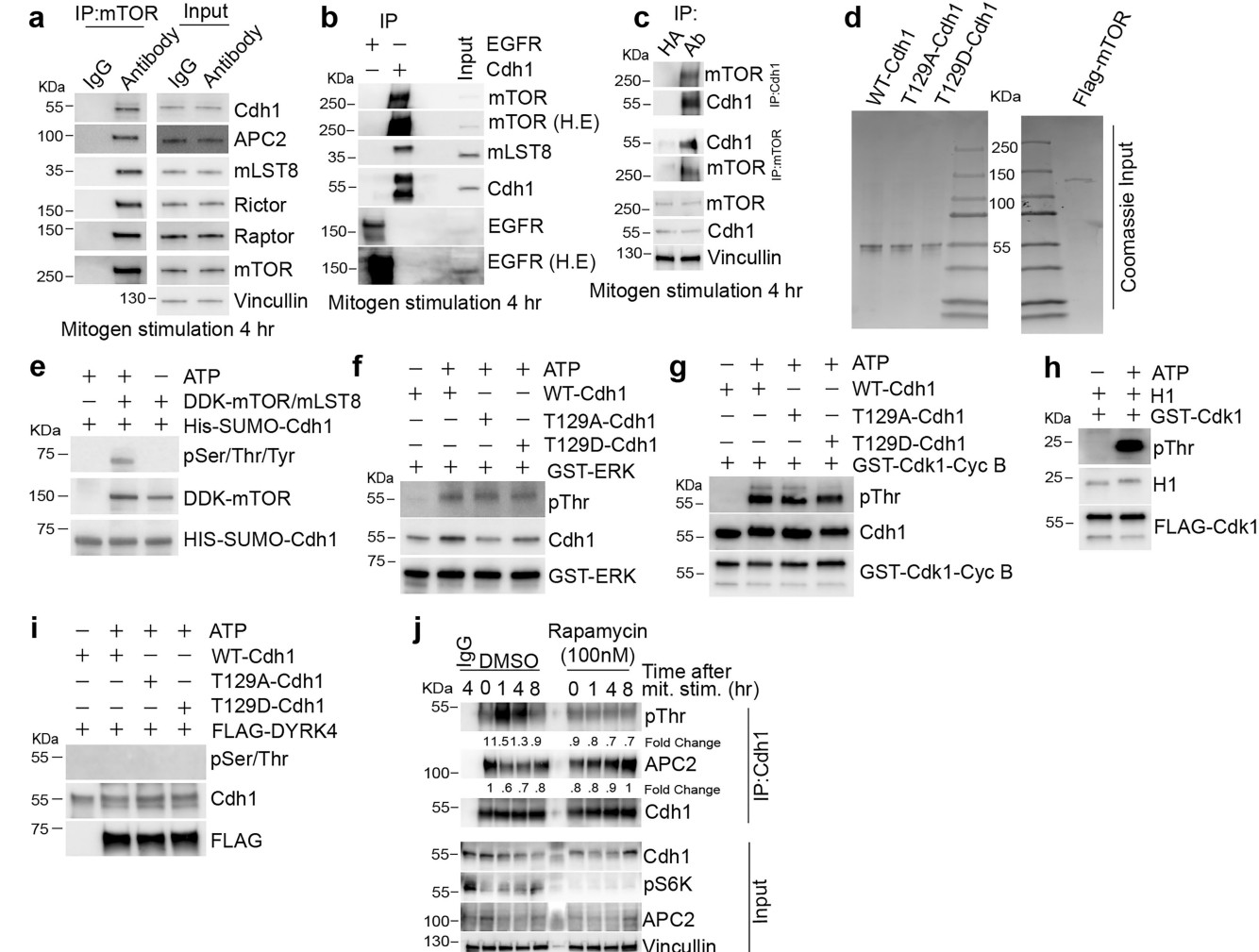

**Extended Data Fig. 4 | mTORC1 phosphorylates Cdh1. a**, MCF-10A cells were starved for 48 h to induce quiescence followed by mitogen stimulation for 4 h. Cells were then collected and lysed. Whole cell extracts were immunoprecipitated with an mTOR antibody and probed for the indicated proteins. Representative blot from n = 2 independent experiments. **b**, MCF-10A cells were starved for 48 h to induce quiescence followed by mitogen stimulation for 4 h. Cells were then collected and lysed. Whole cell extracts were immunoprecipitated with either Cdh1 or EGFR antibody and probed for the indicated proteins. Representative blot from n = 2 independent experiments. **c**, MCF-10A cells were starved for 48 h to induce quiescence followed by mitogen stimulation for 4 h. Cells were then collected and lysed. Whole cell extracts were immunoprecipitated with HA as a negative control or with antibodies to either Cdh1 (top) or mTOR (middle) and probed for the indicated proteins. Whole cell lysates are shown at the bottom. Representative blot from n = 2 independent experiments. **d**, Coomassie stained SDS-PAGE gels of 400 ng of the indicated purified proteins. **e**, In vitro kinase assay with purified His-Sumo-Cdh1 from bacteria and recombinant mTOR-mLST8. Kinase reactions were resolved on SDS-PAGE and probed with a pan phospho Serine/Threonine/Tyrosine antibody. Representative blot from n = 3 independent experiments. **f**, In vitro kinase assay of untagged Cdh1-WT, Cdh1-T129A, and Cdh1-T129D with recombinant GST-ERK. Kinase reactions were resolved on SDS-PAGE and probed with a pan phospho-threonine antibody. Representative blot from n = 2 independent experiments. **g**, In vitro kinase assay with untagged Cdh1-WT, Cdh1-T129A, and Cdh1-T129D with recombinant GST-Cdk1-Cyclin B. Kinase reactions were resolved on SDS-PAGE and probed with a pan phospho-threonine antibody. Representative blot from n = 2 independent experiments. **h**, In vitro kinase assay with untagged H1 and recombinant GST-Cdk1-cyclin B. Kinase reactions were resolved on SDS-PAGE and probed with a pan phospho-serine/threonine/tyrosine antibody. Representative blot from n = 2 independent experiments. **i**, In vitro kinase assay with untagged Cdh1-WT, Cdh1-T129A, and Cdh1-T129D with recombinant FLAG-DYRK4. Kinase reactions were resolved on SDS-PAGE and probed with a pan phospho-threonine antibody. Representative blot from n = 2 independent experiments. **j**, Cells were starved for 48 h to induce quiescence, followed by mitogen stimulation with or without rapamycin. Whole cell lysates were immunoprecipitated with anti-Cdh1 antibody and probed for indicated proteins. Representative blot from n = 2 independent experiments.

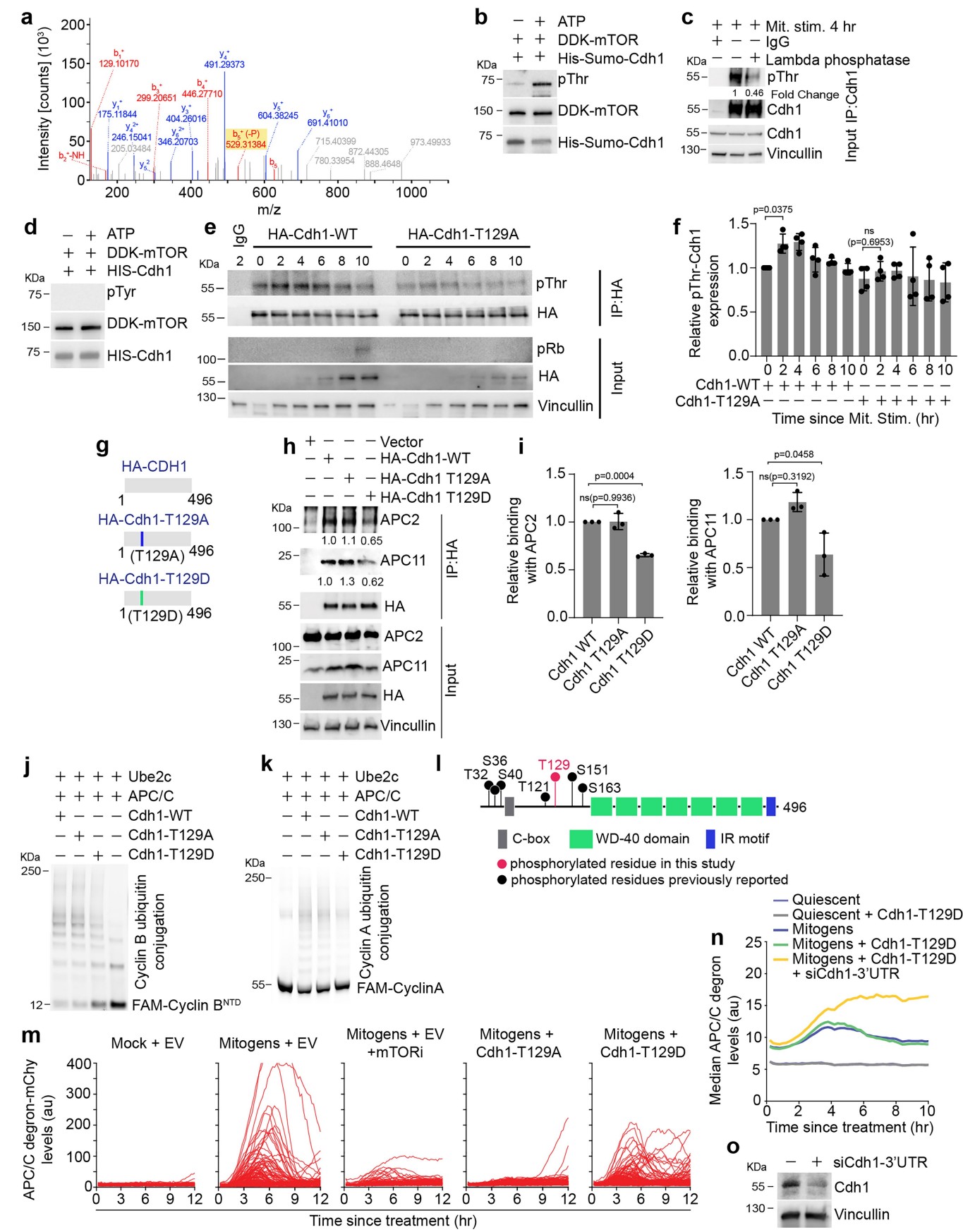

**Extended Data Fig. 5** | See next page for caption.

**Extended Data Fig. 5 | mTORC1 phosphorylates Cdh1 on T129. a**, Purified Cdh1 phosphorylation spectra analysed by mass spectrometry from Extended Data Fig. 4e. The red and blue peaks represent matched b+ and y+ ions, respectively, confirming the peptide sequence and phosphorylation site at T129. **b**, In vitro kinase assay with mTOR and Cdh1. Kinase reactions were resolved on SDS-PAGE and probed with a pan-phospho Threonine antibody. **c**, MCF-10A cells were starved for 48 h to induce quiescence. Cells were mitogen stimulated for 4 h, collected, and lysed. Whole cell lysates were immunoprecipitated with a Cdh1 antibody and treated with or without Lambda phosphatase. Immunoprecipitates were resolved on SDS-PAGE and probed for indicated proteins. Relative band intensities are shown below each band. Representative blot from n = 2 independent experiments. **d**, In vitro kinase assay with mTOR and Cdh1. Kinase reactions were resolved on SDS-PAGE and probed with a pan-phospho Tyrosine antibody. **e**, MCF-10A cells expressing either pTet$_{on}$-HA-tagged wild-type Cdh1 or Cdh1-T129A mutant were starved for 48 h to induce quiescence. Cells were treated with doxycycline for 12 h followed by mitogen stimulation for the indicated time points. Cells were then collected, lysed, and whole cell lysates were immunoprecipitated with an HA antibody and probed for the indicated proteins. n = 4 independent experiments. **f**, Quantification of the relative phospho-threonine-Cdh1 levels from (e). Data are mean ± SD from n = 4 independent experiments. P-values were calculated using a one-way ANOVA. ns, Not significant. **g**, Schematic diagram of wild-type and threonine 129 mutated to either alanine (T129A) or aspartic acid (T129D). **h**, HEK-293T cells were transfected with indicated plasmids. Post 48 h of transfection, cells were collected, and lysed. Whole cell lysates were immunoprecipitated with an HA antibody. Immunoprecipitates were resolved on SDS-PAGE and probed for indicated proteins. Representative blot from n = 3 independent experiments. **i**, Quantification from (h). Data are mean ± SD from n = 3 independent experiments. P-values were calculated using a one-way ANOVA. ns, Not significant. **j**, Ubiquitylation reactions using recombinant APC/C and UBE2C combined with wild-type Cdh1, Cdh1-T129A, or Cdh1-T129D. Ubiquitinated cyclin B1$^{NTD}$ was detected by fluorescence scanning. Representative gel from n = 3 independent experiments. **k**, Ubiquitylation reactions using recombinant APC/C and UBE2C combined with wild-type Cdh1, Cdh1-T129A, or Cdh1-T129D. Ubiquitinated cyclin A2 was detected by fluorescence scanning. Representative gel from n = 3 independent experiments. **l**, Schematic diagram of the domain architecture of Cdh1 (alias: Fzr1). Previously identified phosphorylation sites are noted with black dots and the T129 phosphorylation site identified in this study is noted with a pink dot. **m**, MCF-10A cells were starved for 48 h to induce quiescence, followed by mitogen stimulation to promote cell cycle entry. Single-cell traces of APC/C biosensor levels in cells treated with mitogens with and without Rapamycin or transient ectopic expression of Cdh1-T129A or Cdh1-T129D. N = 200 cells for each condition. **n**, Median APC/C degron levels from the indicated condition. MCF-10A cells expressing pTet$_{on}$-Cdh1-T129D were starved for 48 h to induce quiescence. Cells were treated with doxycycline for 12 h. Where indicated, cells were transfected with siRNA targeting the 3'UTR of Cdh1. Cells were then mitogen stimulated to induce cell cycle entry and time-lapse imaging was conducted to measure APC/C degron levels. **o**, Immunoblot validation of Cdh1 knockdown using 3' UTR specific siRNA.

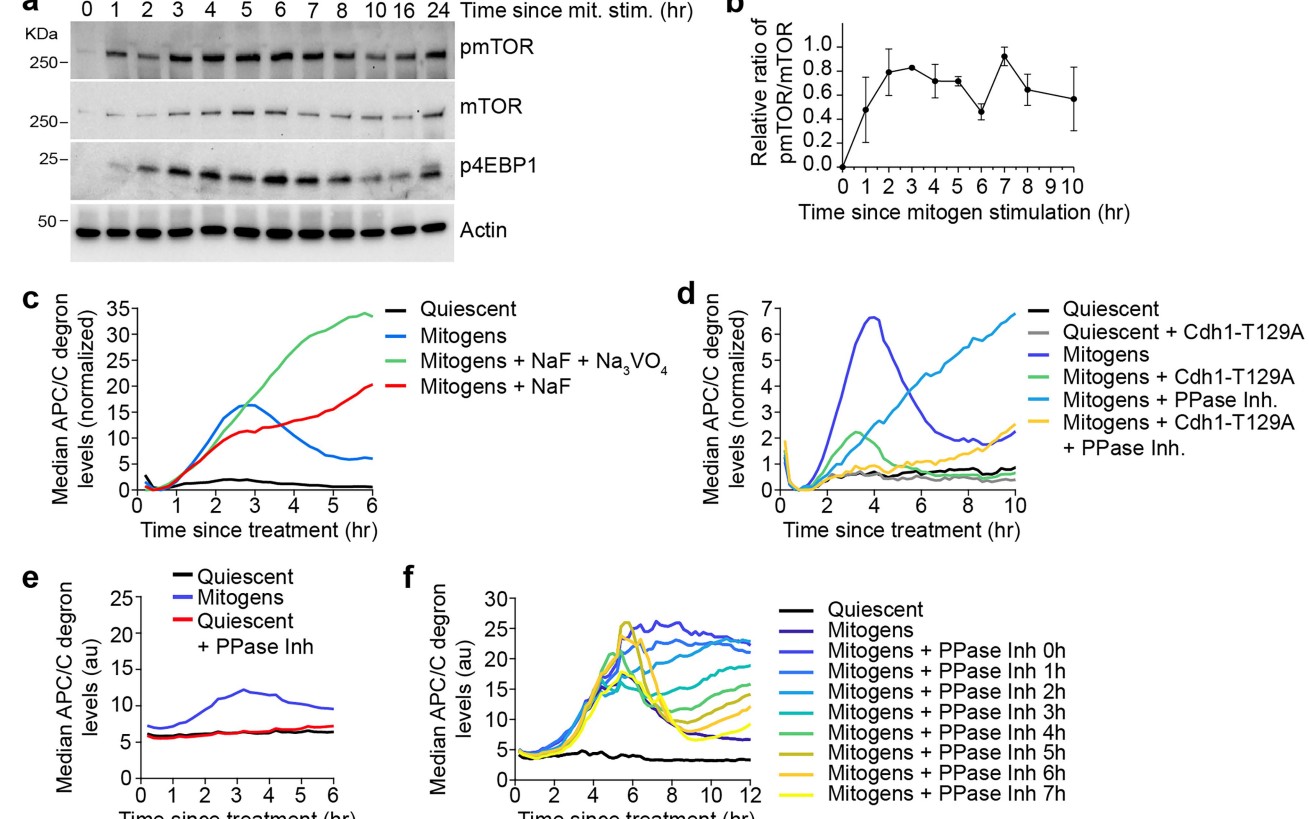

**Extended Data Fig. 6 | Protein phosphatase activity leads to APC/C reactivation. a**, MCF-10A cells were starved for 48 h to induce quiescence followed by mitogen stimulation to promote cell cycle entry. Cells were collected at the indicated time points, lysed, and whole cell lysates were immunoblotted for the indicated proteins. Representative blot of n = 3 independent experiments. **b**, Quantification of pmTOR/mTOR ratio from (a). n = 3 independent experiments. Error bars represent mean ± SEM. **c**, MCF-10A cells were starved for 48 h to induce quiescence. Cells were stimulated with either DMSO or mitogens in the presence or absence of sodium fluoride (NaF) plus sodium orthovanadate (Na₃VO₄) or just NaF. Data are median APC/C degron levels. **d**, MCF-10A cells expressing pTet_on-Cdh1-T129A were starved for 48 h to induce quiescence.

Cells were treated with doxycycline for 12 h, followed by stimulation with either DMSO or mitogens and in the presence or absence of protein phosphatase inhibitor at time 0 h. Data are median APC/C degron levels. N = 100 cells in each condition. **e**, MCF-10A cells were starved for 48 h to induce quiescence. Cells were stimulated with either DMSO (quiescent) or mitogens and in the presence or absence of protein phosphatase inhibitor at time 0 h. Data are median APC/C degron levels. N = 100 cells in each condition. **f**, MCF-10A cells were starved for 48 h to induce quiescence. Cells were stimulated with either DMSO (quiescent) or mitogens at time 0 h and then treated with protein phosphatase inhibitor at time the indicated time. Data are median APC/C degron levels. N = 100 cells in each condition.

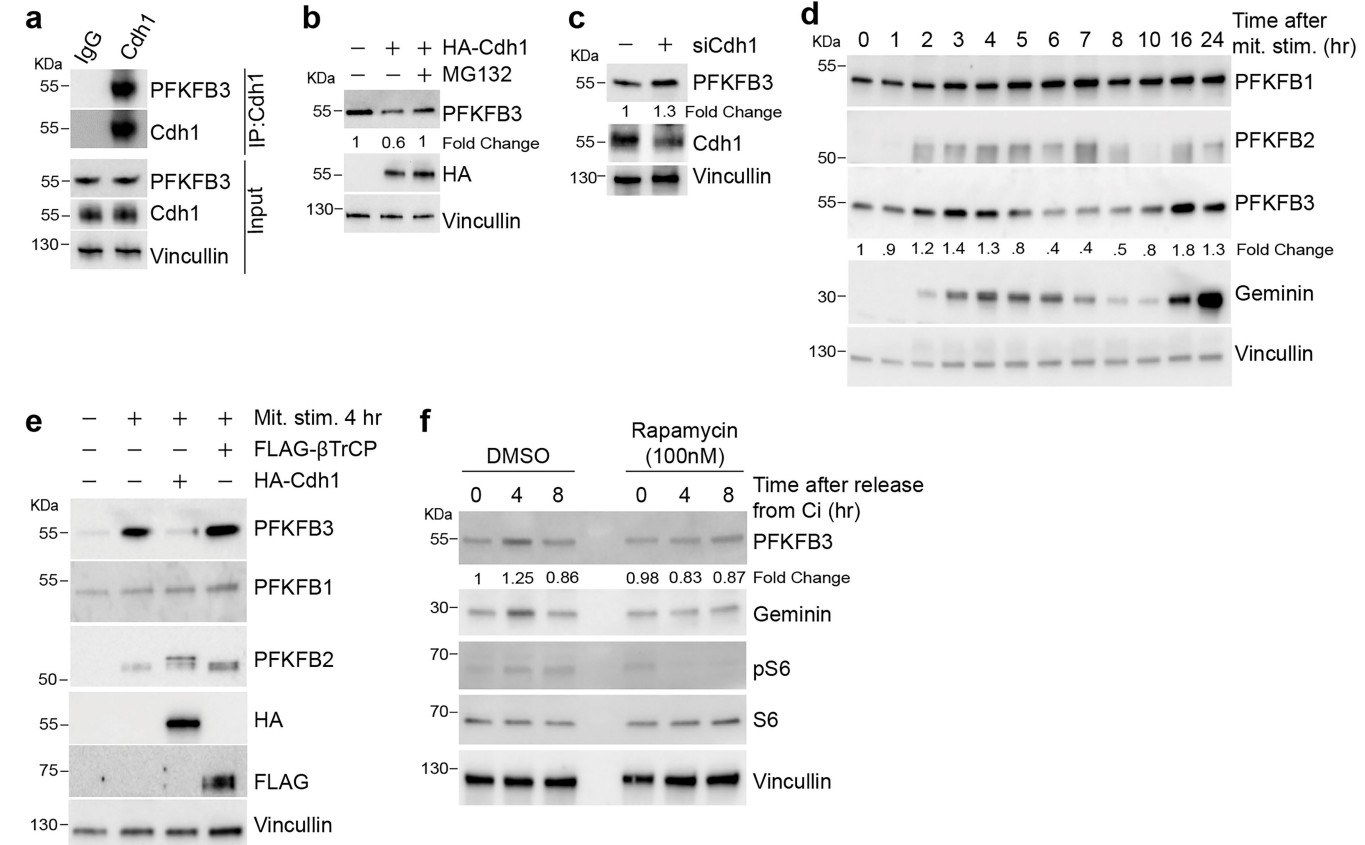

**Extended Data Fig. 7 | Transient accumulation of PFKFB3 is dependent on mTOR. a**, MCF-10A cells were treated with 5 µM of MG132 for 6 h and collected. Whole cell lysates were immunoprecipitated with a Cdh1 antibody and the immunoprecipitates were resolved on SDS-PAGE and immunoblotted for the indicated proteins. Representative blot from n = 3 independent experiments. **b**, MCF-10A cells were transfected with either vector control or HA-Cdh1 for 48 h. Cells were treated with 5 µM of MG132 for the last 6 h before collection. Whole cell lysates were resolved on SDS-PAGE and immunoblotted for the indicated proteins. Representative blot from n = 3 independent experiments. **c**, MCF-10A cells were transfected with either control or Cdh1 siRNA. Cells were collected after 48 h, and whole cell lysates were immunoblotted with the indicated proteins. Representative blot from n = 3 independent experiments. **d**, MCF-10A cells were starved for 48 h to induce quiescence followed by mitogen stimulation to promote cell cycle entry, and cells were collected at the indicated time points. Whole cell lysates were resolved on SDS-PAGE and probed for the indicated proteins. Representative blot from n = 3 independent experiments. **e**, MCF-10A cells were starved for 48 h to induce quiescence followed by transfection with the indicated plasmids. Cells were mitogen stimulated and collected at the indicated time point. Whole cell lysates were resolved on SDS-PAGE and probed for the indicated proteins. Representative blot from n = 2 independent experiments. **f**, MCF-10A cells were contact inhibited for 48 h to induce quiescence followed by release with or without rapamycin treatment. Cells were collected at the indicated time points and whole cell lysates were resolved on SDS-PAGE and probed for indicated proteins. Representative blot of n = 2 independent experiments.

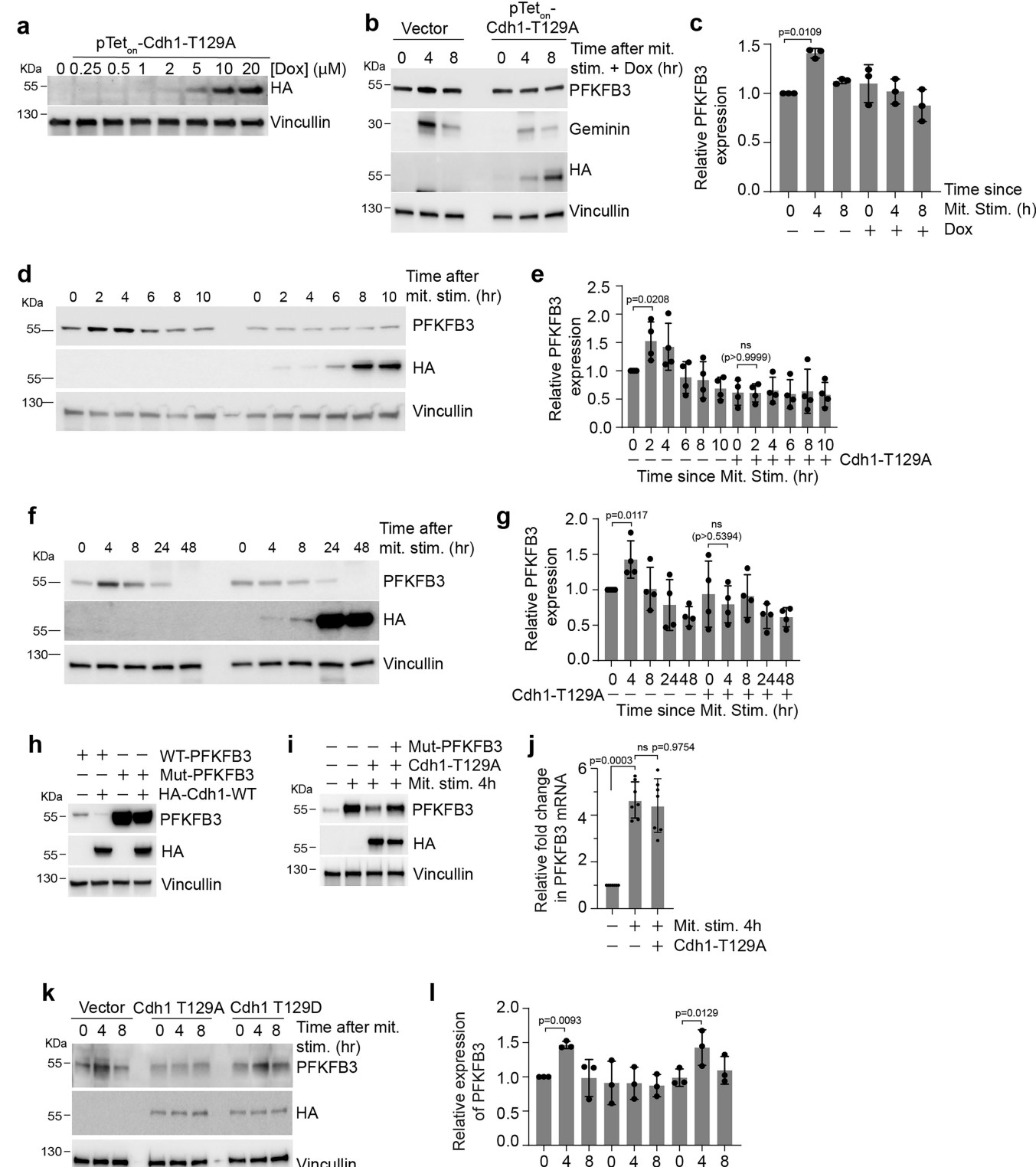

**Extended Data Fig. 8 |** See next page for caption.

**Extended Data Fig. 8 | Transient accumulation of PFKFB3 is dependent on Cdh1-T129 phosphorylation by mTOR. a**, MCF-10A cells infected with pTet$_{on}$-Cdh1-T129A were mitogen starved for 48 h to induce quiescence. Cells were then treated with increasing concentration of doxycycline for 8 h to induce the expression of HA-Cdh1-T129A. Representative blot of n = 3 independent experiments. **b**, MCF-10A cells infected with either empty vector or pTet$_{on}$-Cdh1-T129A were starved for 48 h to induce quiescence followed by mitogen stimulation with 2.5 μM Doxycycline treatment. Cells were collected at the indicated time points. Whole cell lysates were resolved on SDS-PAGE and probed for the indicated proteins. Representative blot of n = 3 independent experiments. **c**, Quantification of PFKFB3 relative expression in presence of doxycycline inducible Cdh1-T129A, from (b). Data are mean ± SD from n = 3 independent experiments. P-values were calculated using a one-way ANOVA. **d**, MCF-10A cells infected with either empty vector or pTet$_{on}$-Cdh1-T129A were starved for 48 h to induce quiescence followed by mitogen stimulation with doxycycline treatment. Cells were collected at the indicated time points. Whole cell lysates were resolved on SDS-PAGE and probed for the indicated proteins. Representative blot of n = 4 independent experiments. **e**, Quantification of relative PFKFB3 expression from (d). Data is from n = 4 independent experiments. P-values were calculated using a one-way ANOVA. **f**, MCF-10A cells infected with either empty vector or pTet$_{on}$-Cdh1-T129A were starved for 48 h to induce quiescence followed by mitogen stimulation with doxycycline treatment. Cells were collected at the indicated time points. Whole cell lysates were resolved on SDS-PAGE and probed for the indicated proteins. Representative blot of n = 4 independent experiments. **g**, Quantification of relative PFKFB3 expression from (f). Data are mean ± SD from n = 4 independent experiments. P-values were calculated using a one-way ANOVA with Sidak's multiple comparison test. ns, Not significant. **h**, MCF-10A cells transfected with vectors to express the indicated protein. WT-PFKFB3, wild-type PFKFB3; PFKFB3$^{KEN mut}$, KEN box mutant PFKFB3, with and without WT-Cdh1. Whole cell lysates were resolved on SDS-PAGE and probed for the indicated proteins. **i**, Quiescent MCF-10A cells were transfected with either Cdh1-T129A, PFKFB3$^{KEN mut}$, or both and stimulated with mitogens for 4 h. Whole cell lysates were resolved on SDS-PAGE and probed for the indicated proteins. **j**, Quiescent MCF-10A cells transfected with either empty vector or Cdh1-T129A were treated with either DMSO or mitogens for 4 h. Data is mean fold change ± SD of PFKFB3 mRNA from n = 3 independent experiments. P-values were calculated using a one-way ANOVA. ns, Not significant. **k**, Quiescent MCF-10A cells transfected with either empty vector, Cdh1-T129A, or Cdh1-T129D were treated with either DMSO or mitogens for the indicated time points. Whole cell lysates were probed for the indicated proteins. Representative blot from n = 3 independent experiments. **l**, Quantification of (i). Data are mean ± SD from n = 3 independent experiments. P-values were calculated using a one-way ANOVA. ns, Not significant.

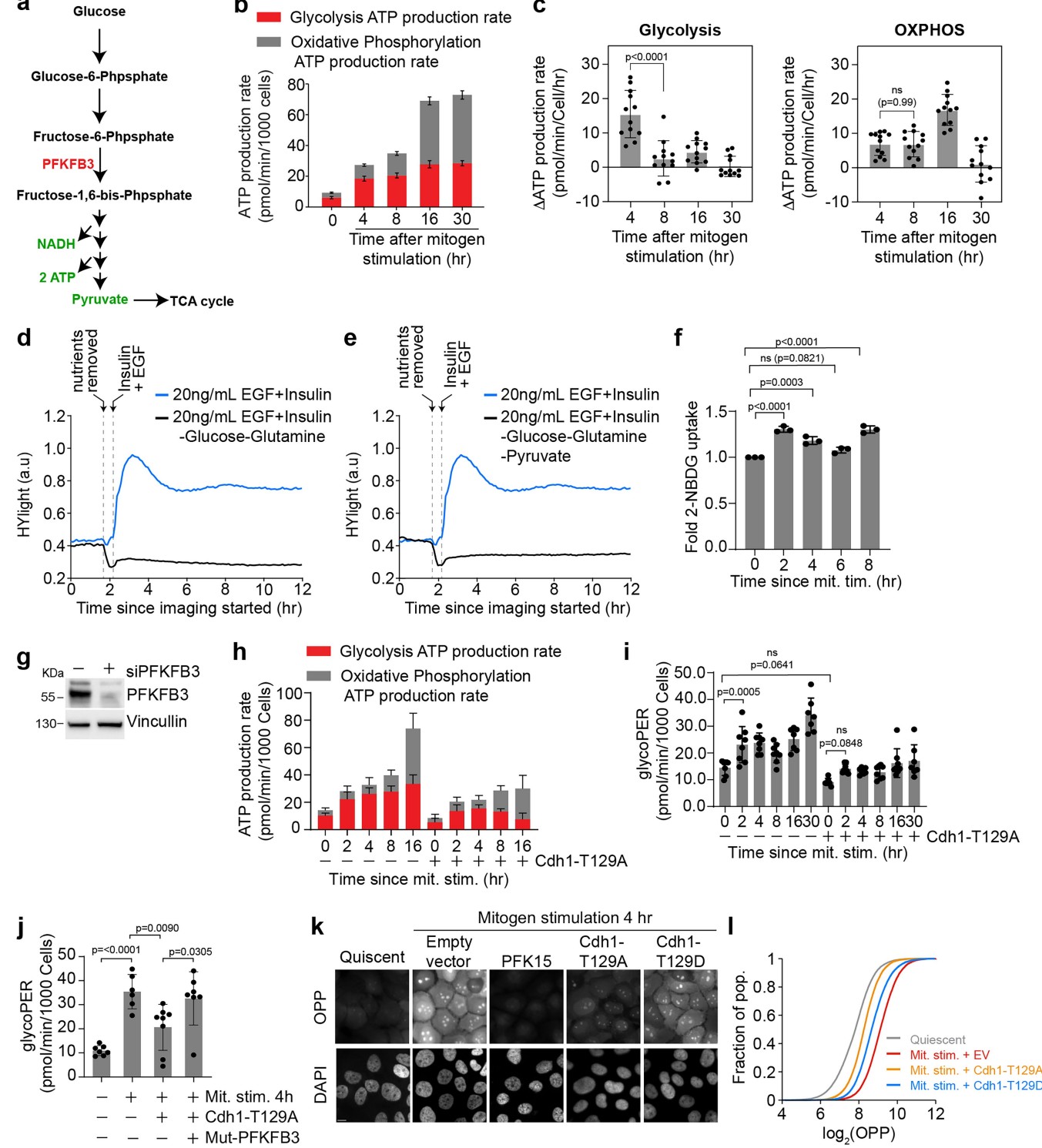

**Extended Data Fig. 9** | See next page for caption.

**Extended Data Fig. 9 | Transient APC/C inactivation promotes glycolysis and protein translation during cell cycle entry. a**, A flowchart depicting the glycolysis pathway. **b**, ATP produced either from glycolysis (red bar) or the mitochondrial OXPHOS pathway (grey bar) in MCF-10A cells entering the cell cycle. Samples were taken at the indicated timepoints. Data are mean ± SD from n = 3 independent experiments. **c**, Change in the rate of ATP production by either glycolysis (left) or oxidative phosphorylation (left; OXPHOS) between time points from (b). Data are mean ± SD from n = 3 independent experiments. P-values were calculated using a one-way ANOVA. ns, Not significant. **d**, Plots depict mean HYlight signals. MCF-10A were cells cultured without growth factors in medium containing glucose, pyruvate, and glutamine for 50 h. Cells were then deprived glucose and glutamine for 35 min, followed by treatment with 20 ng/mL EGF + 1 ug/ml insulin. HYlight signals were calculated as the ratio of fluorescence recorded using two filter sets, GFP and Sapphire, with measurements taken every 6 min. Means represent a minimum of 2,500 cells. **e**, Plots depict mean HYlight signals. MCF-10A cells were cultured without growth factors in medium containing glucose, pyruvate and glutamine for 50 h. Cells were then deprived glucose, glutamine, and pyruvate for 35 min, followed by treatment with 20 ng/mL EGF + 1 ug/ml insulin. HYlight signals were calculated as the ratio of fluorescence recorded using two filter sets, GFP and Sapphire, with measurements taken every 6 min. Means represent a minimum of 2,500 cells. **f**, Quantification of relative 2-NBDG uptake by MCF-10A cells at indicated timepoints. MCF-10A cells are starved for 48 h to induce quiescence. Last 2 h of starvation, media was deprived of glucose and then stimulated with 25 µg/ml of 2-NBDG. Data are mean ± SD from n = 3 independent experiments. P-values were calculated using a one-way ANOVA. ns, Not significant. **g**, Immunoblot validation of PFKFB3 knockdown using siRNA. **h**, Quantification of the relative rate of ATP generation from either glycolysis or OXPHOS in MCF-10A cells either in quiescence or mitogen stimulated for indicated time points in presence or absence of Cdh1-T129A. **i**, Quantification of the relative glycolytic rate in MCF-10A cells either in quiescence or mitogen stimulated for indicated time points in presence or absence of Cdh1-T129A. Data are mean ± SD from n = 3 independent experiments. P-values were calculated using a one-way ANOVA. ns, Not significant. glycoPER, glycolysis proton efflux rate. **j**, Quantification of the relative glycolytic rate in MCF-10A cells either in quiescence or mitogen stimulated for 4 h in presence or absence of Cdh1-T129A and PFKFB3[KENmut]. Data are mean ± SD from minimum of N = 6 repeats across n = 2 independent experiments. P-values were calculated using a one-way ANOVA. ns, Not significant. glycoPER, glycolysis proton efflux rate. **k**, MCF-10A cells were starved for 48 h to induce quiescence. Cells were then stimulated with either DMSO or mitogens for 4 h, fixed, and then stained using an OPP assay kit. Cells were transfected with either empty vector, Cdh1 T129A, Cdh1 T129D, or treated with the PFKFB3 inhibitor PFK15. Representative images from n = 3 independent experiments. Scale bar is 10 µM. **l**, Cumulative distribution function of single-cell OPP fluorescence levels from cells treated as indicated, similar to (k).

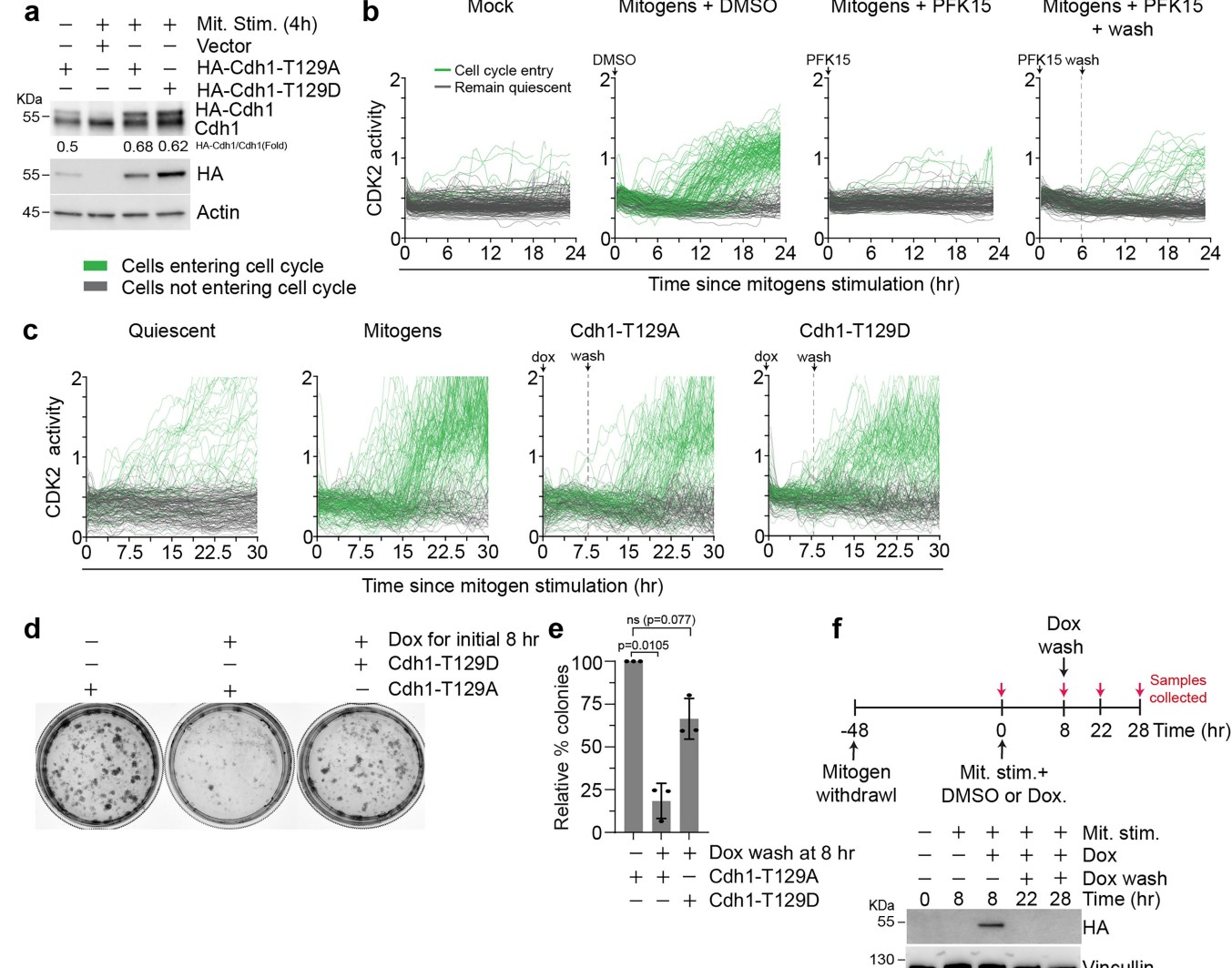

**Extended Data Fig. 10 | Transient APC/C inactivation ensures robust cell cycle entry. a**, MCF-10A cells were starved for 48 h to induce quiescence followed by ectopic over expression of either vector, HA-Cdh1-T129A or HA-Cdh1-T129D. Cells were either kept quiescent or stimulated with mitogen for 4 h, collected, and lysed. Whole cell extracts were immunoblotted for the indicated proteins. Representative blot from n = 2 independent experiments. **b**, MCF-10A cells were starved for 72 h to induce quiescence and then mitogen stimulated to promote cell cycle entry. CDK2 traces of single MCF-10A cells treated as indicated. Vertical dashed line represents when treatment was washed off and replaced with full growth medium. Green traces are cells activating CDK2 and entering the cell cycle and grey traces are cells that did not activate CDK2. N = 200 cells in each condition. **c**, MCF-10A cells were starved for 72 h to induce quiescence and then mitogen stimulated to promote cell cycle entry. CDK2 traces of single MCF-10A cells expressing either pTet_{on}-Cdh1-T129A or pTet_{on}-Cdh1-T129D. Cells were stimulated with either DMSO or mitogens with and without doxycycline at time 0 h. Vertical dashed line

indicates when doxycycline was washed off and replaced with full growth medium. Green traces are cells activating CDK2 and entering the cell cycle and grey traces are cells that did not activate CDK2. N = 200 cells in each condition. **d**, Colony formation assay of MCF-10A cells expressing either pTet_{on}-Cdh1-T129A or pTet_{on}-Cdh1-T129D. Cells were initially starved for 72 h to induce quiescence and then mitogen stimulated to promote cell cycle entry. Where indicated, doxycycline was treated for the first 8 h at time of mitogen stimulation and then washed off. Cells were then allowed to grow for 7 days and then were fixed and stained with crystal violet. Representative image from n = 3 independent experiments. **e**, Quantification of percent colonies as in (d). Data are mean ± SD from n = 3 independent experiments. P-values were calculated using a one-way ANOVA. ns, Not significant. **f**, MCF-10A cells infected with pTet_{on}-Cdh1-T129A were mitogen starved for 72 h to induce quiescence. Cells were then stimulated with mitogens with doxycycline for 8 h to induce the expression of HA-Cdh1-T129A. Doxycycline was washed off after 8 h. Samples were collected at 0, 8, 22, and 28 h. Representative blot of n = 2 independent experiments.

# Reporting Summary

## Statistics

For all statistical analyses, confirm that the following items are present in the figure legend, table legend, main text, or Methods section.

| n/a | Confirmed | |
|---|---|---|
| ☐ | ☒ | The exact sample size (*n*) for each experimental group/condition, given as a discrete number and unit of measurement |
| ☐ | ☒ | A statement on whether measurements were taken from distinct samples or whether the same sample was measured repeatedly |
| ☐ | ☒ | The statistical test(s) used AND whether they are one- or two-sided<br>*Only common tests should be described solely by name; describe more complex techniques in the Methods section.* |
| ☒ | ☐ | A description of all covariates tested |
| ☒ | ☐ | A description of any assumptions or corrections, such as tests of normality and adjustment for multiple comparisons |
| ☐ | ☒ | A full description of the statistical parameters including central tendency (e.g. means) or other basic estimates (e.g. regression coefficient) AND variation (e.g. standard deviation) or associated estimates of uncertainty (e.g. confidence intervals) |
| ☐ | ☒ | For null hypothesis testing, the test statistic (e.g. *F*, *t*, *r*) with confidence intervals, effect sizes, degrees of freedom and *P* value noted<br>*Give P values as exact values whenever suitable.* |
| ☒ | ☐ | For Bayesian analysis, information on the choice of priors and Markov chain Monte Carlo settings |
| ☒ | ☐ | For hierarchical and complex designs, identification of the appropriate level for tests and full reporting of outcomes |
| ☒ | ☐ | Estimates of effect sizes (e.g. Cohen's *d*, Pearson's *r*), indicating how they were calculated |

*Our web collection on statistics for biologists contains articles on many of the points above.*

## Software and code

Policy information about availability of computer code

| Data collection | NIS Elements (v5.11.00) |
|---|---|
| Data analysis | MATLAB (vR2020b). Automated image analysis was performed using custom MATLAB scripts as described in Cappell, S.D. et al Cell 166, 167-180 (2016) (https://github.com/scappell/Cell_tracking). Graphpad Prism 9 (v9.2.0) was used for statistical analysis. RStudio (v1.3.1093) was used for mathematical modeling along with the deSolve package. Proteome Discoverer Software (v2.4) to analyze mass spec data. Xcalibur Quan Browser (v4.1) was used to analyze glucose tracing data. |

For manuscripts utilizing custom algorithms or software that are central to the research but not yet described in published literature, software must be made available to editors and reviewers. We strongly encourage code deposition in a community repository (e.g. GitHub). See the Nature Portfolio guidelines for submitting code & software for further information.

## Data

Policy information about availability of data

All manuscripts must include a data availability statement. This statement should provide the following information, where applicable:

- Accession codes, unique identifiers, or web links for publicly available datasets
- A description of any restrictions on data availability
- For clinical datasets or third party data, please ensure that the statement adheres to our policy

All data is available in the Source Data file. The datasets generated during and/or analyzed during the current study are also available from the corresponding author on reasonable request. All data supporting the findings of this study are available from the corresponding author on reasonable request. Data from the

glucose tracing experiment is available on MassIVE (https://massive.ucsd.edu/ProteoSAFe/static/massive.jsp?redirect=auth) with unique identifier MSV000098175.

## Research involving human participants, their data, or biological material

Policy information about studies with human participants or human data. See also policy information about sex, gender (identity/presentation), and sexual orientation and race, ethnicity and racism.

| | |
|---|---|
| Reporting on sex and gender | n/a |
| Reporting on race, ethnicity, or other socially relevant groupings | n/a |
| Population characteristics | n/a |
| Recruitment | n/a |
| Ethics oversight | n/a |

Note that full information on the approval of the study protocol must also be provided in the manuscript.

# Field-specific reporting

Please select the one below that is the best fit for your research. If you are not sure, read the appropriate sections before making your selection.

[✗] Life sciences          [ ] Behavioural & social sciences          [ ] Ecological, evolutionary & environmental sciences

For a reference copy of the document with all sections, see nature.com/documents/nr-reporting-summary-flat.pdf

# Life sciences study design

All studies must disclose on these points even when the disclosure is negative.

| | |
|---|---|
| Sample size | All sample sizes were chosen based on conventional standards in our fields, considering previously published results. See Chung et al., 2019 (https://doi.org/10.1016/j.molcel.2019.08.020), Cornwell et al., 2023 (https://doi.org/10.1038/s41586-023-06274-3), Franks et al., 2020 (https://doi.org/10.1371/journal.pbio.3000975), Kosaisawe et al., 20221 (https://doi.org/10.1016/j.cmet.2021.01.014). |
| Data exclusions | No data was excluded form the experiments. |
| Replication | All experiments in which p-values are present have been carried out with at least 3 replicates. All experiments were independently reproduced at least twice. |
| Randomization | Samples were allocated randomly for imaging and analysis. Representative single-cell traces where chosen at random from the population for visualization. |
| Blinding | Blinding was not relevant to this study. Image acquisition and analysis was conducted using automated scripts which are not subject to experimental bias. For western blots, blinding is not possible because samples need to be loaded in a particular order. |

# Reporting for specific materials, systems and methods

We require information from authors about some types of materials, experimental systems and methods used in many studies. Here, indicate whether each material, system or method listed is relevant to your study. If you are not sure if a list item applies to your research, read the appropriate section before selecting a response.

## Materials & experimental systems

| n/a | Involved in the study |
|---|---|
| [ ] | [✗] Antibodies |
| [ ] | [✗] Eukaryotic cell lines |
| [✗] | [ ] Palaeontology and archaeology |
| [✗] | [ ] Animals and other organisms |
| [✗] | [ ] Clinical data |
| [✗] | [ ] Dual use research of concern |
| [✗] | [ ] Plants |

## Methods

| n/a | Involved in the study |
|---|---|
| [✗] | [ ] ChIP-seq |
| [✗] | [ ] Flow cytometry |
| [✗] | [ ] MRI-based neuroimaging |

# Antibodies

| | |
|---|---|
| Antibodies used | Cdh1(FZR1) antibody (Santa Cruz; sc56312; 1:800)<br>PFKFB3 (Abcam, AB181861-1001, 1:4000)<br>Cdh1(FZR1) antibody (Abcam; ab217038; 1:1000)<br>mTOR antibody (Cell Signaling Technologies; #2972, 1:1000)<br>phospho mTOR (Cell Signaling Technologies; #291, 1:1000)<br>Ubiquitin (Santa Cruz; SC-8017; 1:800)<br>mouse anti-goat IgG-HRP (Santa Cruz; sc-2354; 1:10000)<br>Anti-rabbit IgG, HRP-linked Antibody (Cell Signaling Technologies; #7074; 1:10000)<br>Anti-mouse IgG, HRP-linked Antibody (Cell Signaling Technologies; #7076; 1:10000)<br>normal rabbit IgG (Cell Signaling Technologies; #2729, 2μg per IP)<br>normal mouse IgG (Santa Cruz; Sc2025, 2μg per IP).<br>mCherry (Abcam; ab167453; 1:1000)<br>Geminin (Cell Signaling Technologies; #5165; 1:1000)<br>Cyclin D1 (Thermo Scientific; MA5-14512; 1:750)<br>phospho Rb (Cell Signaling Technologies; #8516; 1:1000)<br>Rb (Cell Signaling Technologies; #9309; 1:2000)<br>Vinculin (Sigma; V9131; 1:10000)<br>Ki67(Abcam; ab8191;1:1000)<br>p21 (BD biosciences; 556430; 1:1000)<br>p27 (Cell Signaling Technologies; #3686; 1:1000)<br>His tag (Santa Cruz; Sc8036; 1:800)<br>anti-DDK/FLAG (Sigma; F3165; 1:1000)<br>phospho S6 Kinase (Cell Signaling Technologies; #9205; 1:1000)<br>APC2 (Cell Signaling Technologies; #12301; 1:1000)<br>APC6 (Cell Signaling Technologies; #9499; 1:1000)<br>APC11 (Cell Signaling Technologies; 14090; 1:1000)<br>S6 kinase (Cell Signaling Technologies; #9202; 1:1000)<br>phospho 4EBP1 (Cell Signaling Technologies; #2855; 1:1000)<br>Cyclin A2 (Santa Cruz; sc-271682; 1:500)<br>Emi1 (Santa Cruz; Sc-365212; 1:500)<br>Cyclin F (Santa Cruz; Sc-515207; 1:500)<br>βTrCP (Cell Signaling Technologies; #4394; 1:1000)<br>pSer/Thr/Tyr (Fisher; 61-8300; 1:1000)<br>pThr (Abcam; ab9337; 1:500)<br>pTyr (Abcam; ab10321; 1:1000)<br>HA (Santa Cruz; Sc-7392; 1:800)<br>HA (Cell Signaling Technologies; #3724; 1:1000)<br>GST (Santa Cruz; sc-138; 1:750)<br>PFKFB3 (MBS; 9604769; 1:750)<br>PFKFB2 (Cell Signaling Technologies; #13029; 1:1000)<br>PFKFB1 (Abcam; ab155564; 1:1000)<br>βActin (Abcam; ab6276; 1:2000)<br>GAPDH (Abcam; ab128915; 1:2000)<br>Raptor (Cell Signaling Technologies; #2280; 1:1000)<br>Rictor (Cell Signaling Technologies; #2114; 1:1000)<br>MSLT8 (Cell Signaling Technologies; #3274; 1:1000)<br><br>TSC1 (Cell Signaling Technologies ,#6935, 1:1000)<br>NPRL2 (Cell Signaling Technologies, #37344, 1:1000)<br>Histone H1 (Abcam, 11079, 1:1000)<br>EGFR (Santa Cruz, sc-373746, 1:800)<br>AKT (Cell Signaling technologies,#9272, 1:1000)<br>phospho AKT (Cell Signaling technologies ,#4060, 1:1000)<br>pan anti-pS/T antibody (Phospho Solutions, Cat. #PP2551; 1:2000)<br>CDH1 (Sigma, Cat. # CC43-100UG; 1:2000)<br>mouse anti-Rabbit secondary-HRP (SantaCruz Biotechnology, Cat. #sc-2357; 1:2500)<br>or recombinant anti-mouse (SantaCruz Biotechnology, Cat. #sc-516102; 1:2500) |
| Validation | All the antibodies used in this study are commercially available and extensively validated by the company, us, or others. Validation data is available in each of these company's website. In addition, we have confirmed the specificity of the following antibodies using siRNA-mediated knockdown and western blotting: Cyclin A2(ED Fig. 3b), Emi1 (ED Fig. 3c), Cyclin F (ED Fig. 3d), beta-TrCP (ED Fig. 3e), Cdh1 (alias FZR1, Fig. 4c, ED Fig. 5o), PFKFB3 (ED Fig. 9g), Raptor and Rictor (ED Fig. 3m). All other antibodies were not directly validated by us but were validated by the manufacturer for the same species and application as they were used in this study.<br><br>The validation for all the antibodies used are as follows- Cdh1 (FZR1) antibody (Abcam, Ab217038, 1:1000, IP: 2 μg) validated in this study and PMCID: PMC7505520, and (Santa Cruz, sc-56312, IB: 1:800) validated in PMID: 29160310 and PMID: 33523889, (Sigma, |

Cat. # CC43-100UG; 1:2000) validated in PMID: 32345958 , mTOR antibody (CST, #2972, 1:1000) is validated by the manufacturer and PMID: 38886756, phospho mTOR (CST, #2971, 1:1000) is validated by the manufacturer and PMID: 38886756, mCherry (Abcam, ab167453, 1:1000) is validated by the manufacturer and PMCID: PMC10699776, Geminin (CST, #5165, 1:1000) in  PMCID: PMC6390124, Cyclin D1 (Thermo Scientific, MA5 14512, 1:750) is validated by the manufacturer, phospho Rb (Ser807/811) (CST, #8516, 1:1000) by manufacturer and PMCID: PMC11208143, Rb (CST, #9309, 1:2000) by the manufacturer, Vinculin (Sigma, V9131, 1:10000) by manufacturer and , Ki67 (Abcam, ab8191, 1:1000) by PMID: 31707342, p21 (BD biosciences, 556430, 1:1000) by PMID: 38811535, p27 (CST, #3686, 1:1000) by the manufacturer, His tag (Santa Cruz, sc-8036, 1:800) by the manufacturer and PMID: 38848692, anti-DDK/FLAG (Sigma, F3165, 1:1000) by PMID:39705142, phospho S6 Kinase (CST, #9205, 1:1000) by manufacturer and PMID:39028622, PMID: 38886756, APC2 (CST, #12301, 1:1000) by manufacturer and PMID: 29987118, APC6 (CST, #9499, 1:1000) by manufacturer and PMID: 34626566, APC11 (CST, #14090, 1:1000) by manufacturer and PMID: 29987118, S6 kinase (CST, #9202, 1:1000) by manufacturer and PMID: 37083230, phospho 4EBP1 (CST, #2855, 1:1000) by PMID: 38124228, Cyclin A2 (Santa Cruz, sc-271682, 1:500) by PMID: 31380287, Emi1 (Santa Cruz, sc-365212, 1:500) by PMID: 28604711 and PMID: 28604711, Cyclin F (Santa Cruz, sc-515207, 1:500) by PMID: 36951214, βTrCP (CST, #4394, 1:1000) by PMID: 36973255, pSer/Thr/Tyr (Fisher, 61-8300, 1:1000) by manufacturer and PMID: 37874675, pThr (Abcam, ab9337, 1:500) in this study, pTyr (Abcam, ab10321, 1:1000) by manufacturer and PMID: 34697378, pan anti-pS/T antibody (Phospho Solutions, Cat. #PP2551; 1:2000) by PMID: 27841876, HA (Santa Cruz, sc-7392, 1:800, IP: 2 μg and CST, #3724, 1:1000) by manufacturer and PMID: 39627198, GST (Santa Cruz, sc-138, 1:750) by manufacturer and PMID: 40287465, PFKFB3 (MBS, 9604769, IB: 1:750, IP: 2 μg) by PMID: 36289220, PFKFB3 (Abcam, AB181861-1001, 1:4000) by manufacturer, PFKFB2 (CST, #13029, 1:1000) by manufacturer and PMID: 32718270, PFKFB1 (Abcam, ab155564, 1:1000) by PMID: 34679684, βActin (Abcam, ab6276, 1:2000) by manufacturer, GAPDH (Abcam, ab128915, 1:2000) by manufacturer and PMID: 38123554, Raptor (CST, #2280, 1:1000) by PMID: 35869262, Rictor (CST, #2114, 1:1000) by PMID: 35869262, MSLT8 (CST, #3274, 1:1000) by PMID: 20169205, Ubiquitin (Santa Cruz, sc-8017, 1:800) by PMID: 36746962, TSC1 (CST,#6935, 1:1000) by manufacturer and PMID: 36396656, NPRL2 (CST, #37344, 1:1000) by PMID: 36044864, Histone H1 (Abcam, 11079, 1:1000) by manufacturer, EGFR (Santa Cruz, sc-373746, 1:800 ) by PMID: 36795511, AKT (CST,#9272, 1:1000) by PMID: 40155685, phospho AKT (CST,#4060, 1:1000) by manufacturer and PMID: 40191596, mouse anti-goat IgG-HRP (Santa Cruz, sc-2354, 1:10000) by PMID: 40033150, Anti-rabbit IgG, HRP-linked Antibody (CST, #7074, 1:10000) by PMID: 39972131, mouse anti-Rabbit secondary-HRP (SantaCruz Biotechnology, Cat. #sc-2357; 1:2500) by PMID: 33397958 or recombinant anti-mouse (SantaCruz Biotechnology, Cat. #sc-516102; 1:2500) by PMID: 32341344, Anti-mouse IgG, HRP-linked Antibody (CST, #7076, 1:10000) by PMID: 40069201, normal rabbit IgG (CST, #2729, IP: 2 μg) by PMID: 28813410, normal mouse IgG (Santa Cruz, sc-2025, IP: 2 μg) by PMID: 26880551.

# Eukaryotic cell lines

Policy information about cell lines and Sex and Gender in Research

| Cell line source(s) | MCF10A (ATCC: CRL-10317)<br>RPE-1 (ATCC: CRL-4000)<br>HLF (ATCC: PCS-201-013)<br>HEK293T (gift from Dr. Tobias Meyer's Laboratory at Weil Cornell Medical School, ATCC: CRL3216)<br>Mouse MEF (gift from Dr. Joanna Vidigal's Laboratory at the National Cancer Institute) |
|---|---|
| Authentication | Cell lines purchased from ATCC were not further authenticated. HEK293T are not authenticated. |
| Mycoplasma contamination | Cells used in all experiments were routinely tested for mycoplasma contimination and only mycoplasma-negative cells were used in experiments |
| Commonly misidentified lines<br>(See ICLAC register) | No commonly misidentified cell lines were used in the study |

