## [Peer Review File · Nature]

Transient APC/C inactivation by mTOR boosts glycolysis during cell cycle entry

Corresponding Author: Dr Steven Cappell

Version 0:

Reviewer comments:

Referee #1

(Remarks to the Author)

In the present study, Paul et al. attempt to answer a fundamental question concerning the intricacies of cellular metabolism and cell cycle regulation - specifically, how cells coordinate the glycolytic switch and cell cycle entry from a state of cellular quiescence.

The authors began by addressing existing discrepancies in the current understanding of cellular mechanisms, focusing on three key observations: 1) the necessity of glycolysis for cell cycle entry during the G1 phase, 2) the inhibition of glycolysis through PFKFB3 degradation mediated by the Anaphase Promoting Complex/Cyclosome (APC/C), and 3) the continuous high activity of APC/C throughout the G1 phase until its sharp inactivation upon transitioning from the G1 to the S phase.

Upon reassessment, the authors found that the third observation, which suggested that APC/C remains active throughout the G1 phase, is not entirely accurate. Instead, they discovered that the APC/C undergoes a period of transient inactivation during the G1 phase prior to its full inactivation upon the G1/S transition. This key insight corrects a previous oversight by one of the current authors in a previous study (Cappel et al., 2016, Cell), exemplifying commendable scientific integrity through this diligent and transparent revisitation of prior work.

Expanding upon this initial observation, the authors provide substantial evidence for a model in which APC/C is transiently inactivated to promote the activation of glycolysis, essential for the initiation of the S phase. They reveal that this transient inactivation of APC/C is regulated by the mammalian target of rapamycin (mTOR), a well-known regulator of cell growth and metabolism. This regulation is facilitated through direct phosphorylation of CDH1 by mTOR itself, which is subsequently reversed by phosphatase activity during the reactivation of APC/C. The authors also identify PFKFB3, an essential enzyme in the glycolytic pathway previously shown to be an APC/C substrate, as a key mediator through which transient APC/C inactivation triggers a metabolic shift from oxidative phosphorylation to glycolysis, essential for cell cycle re-entry. Lastly, the authors clarify how this APC/C regulation, by coordinating glycolysis, ATP production, and other biosynthetic processes with G1 phase progression, effectively facilitates cell cycle entry and colony formation. The study underscores the critical role of transient APC/C inactivation, downstream of mTOR, in orchestrating the metabolic changes during the cell cycle transition from a quiescent state back into the cell cycle.

Overall, the manuscript presents groundbreaking research that contributes significantly to the understanding of the interplay between cell metabolism and cell cycle progression. The implications of these findings are significant, given the pivotal roles both the APC/C and mTOR signalling pathway play in cell growth, proliferation, and cancer. By bridging the gap between the understanding of the cell cycle and metabolism, their findings are likely to appeal to a large general audience and stimulate further investigations into the regulatory nexus between the cell cycle, metabolic control, and disease states. The narrative is well-constructed, the study is thorough and robust, and the data provided are largely high quality and convincing. Therefore, the scope and impact of the manuscript fit well with the criteria for publication in Nature.

However, while this research has substantial merit, there are several concerns that currently prevent a full endorsement of the manuscript for publication. These concerns primarily relate to the experimental design, interpretation of results, as well as a lack of clarity on the consistency across experimental replicates, which is partly due to the way the data is presented. I will elaborate on these issues in the subsequent sections. I believe that addressing these issues will significantly strengthen the manuscript, making it more deserving of publication in such a prestigious journal.

Major Concerns:

1. Many of the results heavily rely on Immunoblot data. Although the experiments are stated to be performed multiple times ($n = 2$ or 3), in most cases, only one representative blot is presented. The differences in band intensities are fairly subtle in some data sets, leading to potential questions regarding variability and reproducibility. For these data to be validated, band intensities should be quantified across all replicates and mean values with variabilities, or each value, should be shown.
2. Although glycolysis is associated with cell proliferation, its general requirement for cell cycle entry hasn't been fully established. While this was demonstrated in MCF-7 cells (extended Fig.1c), is this requirement applicable to other human cells? To ascertain the importance of the present findings, testing should be done across several other cell lines, including untransformed cells. Further, what would occur if glycolysis isn't turned off and remains active, for example, by overexpressing PFKFB3 or TOR hyperactivation?
3. The role of phosphatases (PPases) in the reactivation of APC/C after its transient inactivation and the significance of this reactivation for cell cycle progression is not entirely convincing. Observations made in this paper using PPase inhibitors could be due to changes in the global protein phosphorylation status, potentially irrelevant to physiological conditions. Further studies, such as using PPase inhibitors at different time points (it might induce CDH1 phosphorylation even in G0) or measuring PPase activity, could help address this.
4. Can transient PPase inhibitor treatment or overproduction of CDH1T129D cause any defects in subsequent cell cycle phases, such as DNA replication or mitosis?
5. The evidence for the mechanisms and role of APC/CDDH1 inhibition by mTOR is not wholly convincing. Further studies using wild-type CDH1 and examining the effect of mTOR addition on CDH1's association with APC/C and/or APC/C's E3 activity could strengthen the presented evidence.
6. To establish that mTORC1, not mTORC2, is responsible for the transient APC/C inactivation, experiments examining the effects of mTORC1-specific component depletion on APC/C activity and cell cycle entry should be performed. How does mTOR generally recognize its substrate?

Minor Points:

1. To assist non-expert readers, the authors should specify early on that their subject is APC/CDDH1, not APC/CCDC20, given that these have distinct functions and regulations.
2. Line 67: Could you clarify the meaning of "Proper" (also in the title)? Please consider using a more descriptive term to articulate the phenotype.
3. Line 75: A number appears to be missing before "hours." Being more specific would enhance clarity.
4. Line 82: Revise "fig1k-n" to "fig. 1k-o".
5. Line 91-92: The calculation of the 33% value needs clarification, along with its implications. If not necessary, consider removing it.
6. Line 216-217: The term "post-transcriptional" might be more appropriate than "post-translational", particularly because mTOR is known to regulate translation.
7. Extended Fig. 1i/j: Considering one of the major concerns, following release from MEK inhibition or contact inhibition, the "transient" nature of APC/C inactivation seems less pronounced. Many cells seem to exhibit a two-step or gradual inactivation (slow and f of APC/C, instead of transient inactivation followed by reactivation). Please provide clarification.
8. In the legend for Fig.1d, please specify what the top and bottom panels represent.
9. Fig.2d: Using prey as a control in the colP experiment may not be ideal. If the protein is sticky, it would attach to the beads. Use -bait (CDH1) as control.
10. Ex.3c, g: The specificity of the phospho-specific antibodies has not been validated. pSer/Thr (ex.3c) displays relatively weak signals on CDH1 at 0hr (quiescence), but pThr already exhibits relatively high signals (ex.3g). pThr also seems to recognise T129A. Ideally, a T129 phospho-specific antibody should be generated. Alternatively, the specificity against phosphorylation should be validated by treating the samples with PPases to check for loss of detection.
11. L130-131: For a similar reason as above, without a negative control with pTyr, it cannot be definitively concluded that there is no Typ phosphorylation. Please moderate the statement.
12. Ex.3c, L125-6: The reduction of Apc2 was very subtle and seemed to recover by 4h, even though CDH1 remained phosphorylated. If the hypothesis is that Tor-dependent phosphorylation causes APC/C to dissociate from CDH1, this does not align with the observed data. It would be beneficial to test in vitro if Tor-dependent phosphorylation can indeed dissociate CDH1 from the APC/C complex and reduce its E3 activity (also mentioned in the general comment).
13. Fig. 2H: Is T129 located near the C-box, which has been shown to be important for APC/C binding and/or activation of its activity? Including the domain structure of CDH1 in the figure, complete with WD40 repeats and C-box and IR motifs, would be beneficial.
14. Fig.3g: The lines are difficult to distinguish. Please alter the line styles and colours for easier interpretation.
15. Ex. 6d: Depletion of PFKFB3 by siRNA does not seem efficient, yet it still shows inhibition of ATP synthesis comparable to that of PFK15. This finding requires explanation.

Referee #2

(Remarks to the Author)

The molecular processes regulating the transition between the quiescent and proliferative cell state, i.e. cell cycle entry, are still not fully understood. Through an impressive amount of work, here the authors suggest the existence of a mechanism that links mTOR as a key signaling pathway to the activity of the anaphase-promoting complex/cyclosome (APC/C) as a major cell cycle regulator, and to the core metabolism, i.e. glycolysis. This link, if convincingly demonstrated, would be one of the first established molecular links between the core cell cycle machinery and the core metabolic machinery in vertebrates, and hence an important scientific study with broad implications for both the cell cycle field as well as the

metabolism field. However, several conclusions of the manuscript are currently not well supported by the data in the manuscript. The authors would need to address the raised concerns satisfyingly before publication is warranted.

General remarks on how the manuscript is written:

In order to sell their story in an attractive manner, the authors sketch a “paradoxical situation”. The paradoxical situation emerges if one considers things in a “black-and-white” manner, e.g. “glycolysis is necessary for cell cycle start”. However, “glycolysis” is likely not a binary variable (i.e. on or off) but is rather a gradual one with glycolysis having a certain flux/activity. Also, the statement of “APC/C inhibits glycolysis” is likely not a “black-and-white” question either. Everybody in biology knows that hardly anything is so black and white and that it is rather a question of activities, rates, or fluxes, i.e. more graded. But by putting it so black and white at the beginning (in words in the introduction) and in Fig. 1b and c, readers might be annoyed. In fact, as the authors also show even quiescent cells retain some glycolytic activity and then if the authors’ black-and-white statement of “glycolysis is necessary for cell cycle start” would be true, then also quiescent cells could exit quiescence.

Are conclusions valid/solid?

1. In vivo Cdh1 phosphorylation dynamics: In vitro, the authors convincingly show that Cdh1 and mTOR physically interact, that mTOR phosphorylates Cdh1 at T129 and that this phosphorylation results in a mild decrease in APC/C-Cdh1 activity. However, when it comes to Cdh1 phosphorylation dynamics in cells exiting quiescence in vivo, the overall phosphorylation dynamics are less clear – this is in parts because the authors are not using a phospho-specific antibody against the T129 site but rely on Cdh1 immunoprecipitation in combination with a pan-phospho-threonine/serine or pan-phospho-threonine antibodies. The only direct evidence that T129 is the only regulated phospho-site on Cdh1 during the transient inactivation is provided in Figure 2i, where they probe Cdh1 and Cdh1 T129A immunoprecipitated from cell extracts made 4 hours after release from quiescence and show that the wt but not the T129 mutant is recognized by an anti-phospho-threonine antibody suggesting that T129 indeed is the only threonine site phosphorylated in vivo at 4 h time point. However, the time courses in Extended Figures 3C and 3g seem more puzzling. Extended Figure 3C shows an increase in phospho-Cdh1. However, the experiment is cut short at 4 hrs, and it is not possible to assess whether Cdh1 phosphorylation remains high or decreases at later times. Moreover, the antibody used (pan anti pSer/Thr) does not allow one to conclude that it is specifically the T129 phosphorylation (could be other mediated by additional phospho-serines). The experiment in Extended Figure 3g shows a longer time course (up to 8 h) with the appropriate pan-anti pThr antibody, however, the changes in phosphorylation state are much less pronounced in this experiment with a suspiciously high phosphorylation signal already at 0 h and a very mild increase at 4 and mild decrease at 8h, respectively. To convincingly demonstrate the phosphorylation dynamics of Cdh1 by mTOR the authors should at the very least repeat the experiment shown in Extended Figure 3c with the pan-anti pThr antibody including at least another 2-3 timepoints between 4 and 10 hours and ideally including the T129A mutant as an additional control. The immunoblot repeats should be quantified. Mass spectrometry analysis of Cdh1 isolated from 4 hours after release from serum starvation demonstrating that indeed T129 is the only prominent phospho-site would further strengthen the authors argument.

2. Sustained mTOR activity following mitogen stimulation: This is an important claim and is based on the data of Extended Fig. 4a. However, it seems that mTOR abundance is not constant over time in the shown immunoblot, and the phosphorylation of mTOR and p4EBP1 also fluctuates. It is difficult to assert the mTOR activity dynamics from this blot. Some type of quantification would be necessary here (e.g. phospho over mTOR bands). The other two replicates should be also quantified to ensure reproducibility of the result.

3. Excluding other regulators of the APC/C: The authors observe that Cdh1 T129D cells still shows transient APC/C inactivation probably due to presence of endogenous Cdh1 in the background, but this could also suggest that either additional regulators inactivate the APC/C, or that mTOR controls additional phosphosites on Cdh1. The link between transient Cdh1 T129 phosphorylation and APC/C inactivation is an important part of the mechanism described here, and it is important that it is clearly established (e.g. using Cdh1 siRNA to eliminate the endogenous Cdh1).

4. Time-delayed activation of a counteracting phosphatase mediates re-activation of the APC/C: The authors use sodium fluoride and sodium ortho-vanadate to inhibit phosphatase activity and observe a sustained inactivation of the APC/C in most cells. They conclude that an unidentified phosphatase acts on T129 in a time-delayed manner resulting in the re-activation of APC/C after the initial phosphorylation and inactivation by mTOR. However, the used phosphatase inhibitors constitute a massive, global perturbation impacting the phosphorylation state of numerous proteins and regulators and hence does not allow for this specific conclusion. The authors would need to demonstrate that the observed sustained inactivation is indeed solely mediated via Cdh1-T129 phosphorylation (e.g. by performing a ‘rescue’ experiment with the Cdh1-T129A mutant or mTOR inhibition). Furthermore, ortho-vanadate is thought to be a tyrosine phosphatase inhibitor – considering that the authors excluded tyrosine phosphorylation as a regulatory mechanism, the inhibitor seems a suboptimal choice.

5. Metabolism: A key claim of their model is that RATE through glycolysis transiently increases. To support this claim they argue that the abundance of a rate-limiting enzyme of glycolysis increases and provide Seahorse and ECAR measurements. Beyond, the authors also measured metabolite concentrations (ATP, NADH and pyruvate). There are several concerns about the metabolic aspect of the model:

a. The measurements of the metabolite concentrations are irrelevant for the model as the model is about a metabolic RATE and not about metabolite levels. Note, from metabolite levels one cannot draw any conclusions on metabolic rates: For instance, if ATP levels are high one cannot conclude that rate through glycolysis is high. The concentration of a metabolite emerges from both the rates of production of the metabolite and the rates of consumption of the metabolite. Furthermore, also the way the authors measure the metabolite concentrations is questionable: (i) As metabolic turnover rates are high, directly

after sampling of cells for metabolite analysis, metabolic activity needs to be immediately quenched to prevent abiotic changes in the metabolite levels. Nothing on quenching is mentioned in the method section (lines 470ff), raising doubts on the overall validity on the reported metabolite concentrations. (ii) Across the different measured metabolite concentrations, the authors report very different units (c.f. Fig. 4f, j, k). The one metabolite is reported in absolute grams, the other in absolute moles, the third is reported in moles/cell. Data in absolute grams or absolute moles do not make sense. To check the validity of these measurements, it is key that metabolite data are reported in units per cell or per cell volume.

b. As mentioned above, the key question is whether there is indeed an increase in the RATE of glycolysis around the time of the proposed APC/C inactivation. If the authors are right with their story, then it really needs to be an increase in the glycolytic rate (and not a black-and-white point; see comment above). This is because in fact quiescent cells still have glycolytic activity (as the authors also show). Thus, is the data really showing an increase in the glycolytic rate? To this end, the authors use Seahorse measurements (Fig. 4g and h) from which they infer both the contribution of glycolysis and OXPHOS towards ATP generation and perform measurements of ECAR (extracellular acidification rate, correlating with the rate of glycolysis, Fig. 4i). There are several concerns with these measurements: (i) Both are not direct measurements of the rate of glycolysis but indirect ones. Can the Seahorse assay really without doubt reveal the proportion between glycolytic and OXPHOS-based ATP generation? The differences reported in the proportions are not huge. Thus, assumptions might be critical. As a more direct measurement of the rate of glycolysis, the authors could measure the depletion of glucose from the medium and from this data infer the rate of glycolysis. (ii) As the authors claim that an increase in the glycolytic rate occurs, the authors need to show the absolute rates (of glycolytic and OXPHOS ATP generates) instead of the proportions. Why not plotting the change in glycolytic ATP production rate (i.e. the difference between 2 measurement time points) as a function of time? (iii) In Fig. 4h and i, the authors only show data from a single time point. Time courses of ATP production rate and ECAR in the control and in key mutants/perturbations would be needed.

c. The final "metabolism-related" question is whether an increase in the RATE of glycolysis is indeed caused by reduced degradation of PFKFB3. The authors argue that there is a transient accumulation of this enzyme. Also here, more thorough time course measurements of this protein in the control and in key mutants with more fine-grained sampling and over a longer period of time would be needed. Also, Fig. 4a shows only a "representative" blot. In addition, the bands should be quantified and the data from all independent experiment should be plotted together.

Minor comments:

1. In the mammalian field, the term "cell cycle entry" is reserved for the transition from quiescence to the proliferating state. As the term is very similar to "cell cycle start" and as for instance the yeast field uses this term for the moment when cells commit to the next cell cycle, there is the danger that the term "cell cycle entry" could be considered the same as "cell cycle start". As the authors mention in the discussion (line 282), their finding is only true for the G0/G1 transition (i.e. for "entry" and not for "start"), it would be good if the title and abstract would avoid this potential confusion with readers from other fields, by for instance changing the title from "...ensures proper cell cycle entry" to "... ensures proper exit from quiescence".
2. Figure 2C: typo in y-axis label
3. Fig 2F&G: The source of the GST-Cdh1 is not described in the methods. Furthermore, the His-Cdh1 (Fig 2F) and the GST-Cdh1 (Fig 2G) run almost at the same molecular weight although the GST tag is significantly larger.
4. Figure 4A - what happened to Cdh1 at time point 0h? Cdh1 is present at that time point in other blots (e.g. Extended Figure 3C).
5. Discussion: Licensing argument: they do not see any evidence for transient kinase activity.
6. Line 71: the argument seems to be "Cdh1 overexpression -> APC overactivation -> reduced glycolysis". How can we exclude that reduced glycolysis is not a consequence of cell cycle misregulation by an overactive APC/C? How strong is the overexpression of Cdh1?
7. Line 90-92: measurements for other time points before and after 5 hrs are not presented, so it is difficult to tell how APC/C activity varies over time. Granted, microscopy observations provide strong evidence that APC/C is transiently inactivated, but the MG132 assay is more quantitative and is done at only one time point.
8. Line 135: the band in Fig.2g is very faint.
9. 134-135: what does the binding of mTOR and Cdh1 demonstrate? If mTOR still binds the T129A mutant, doesn't this imply that it may phosphorylate another residue (or residues) as well? What about the T129D mutant? (is it also bound less by mTOR?)
10. Line 146: band in blot shown in Fig.2j are too faint
11. Line 193-194: the phrasing is a bit sloppy here, but is revealing on the way they think: fast inhibition followed by slow activation does not generate an incoherent feedforward (IFF), but can be generated by an incoherent feedforward. In other words, an IFF can produce what they observe, but what they observe is not necessarily produced by an IFF.
12. Line 196-198: the model is too phenomenological to be of any use. No reaction mechanisms are modelled, just a bunch of sigmoidal activation/inhibition terms. The meaning of several states is unclear (what is mTOR? APC? PP?). The authors don't describe how they came up with the parameters. On a more general level, it is hard to understand what the model contributes to the paper. Saying "we made a model and it reproduces what we observed" does not mean much. The model is supposed to reveal some aspect of the system that was not obvious from the data, or make a prediction that is validated by further experiments.
13. Line 431: The authors refer to previously described methods without referring to any papers where a description of these methods can be found.
14. Why is Fig. 1k normalized to data in Ext. Fig 3k?
15. Line 844: The "coordinating" aspect is not shown in this paper, and this is thus speculation.
16. Fig. 5e: Typo in glycolysis

Referee #3

(Remarks to the Author)

In this manuscript, Paul et al tackle an apparent paradox in cell cycle regulation, whereby the ubiquitin ligase APC/C, which is necessary for cell cycle progression, also inhibits glycolysis, an obligate metabolic process for cells to exit quiescence and resume proliferation.

To reconcile these opposing actions, the authors report that cells briefly deactivate APC/C upon entering the cell cycle, allowing a shift to glycolysis. This occurs due to mTOR-triggered phosphorylation of the APC/C adapter protein Cdh1, leading to increased levels of the glycolytic enzyme PFKFB3. Later, phosphatase activity restores APC/C function, transitioning cells back to oxidative phosphorylation. This process coordinates cell cycle and metabolism through a temporary inhibition of APC/C, facilitating a burst of glycolysis.

Overall, this is a conceptually novel and interesting work that connects selective APC regulation during cell cycle re-entry to metabolism. The authors provide convincing evidence that PFKFB3 phosphorylation occurs in a Cdh1 phosphorylation-dependent manner. Moreover, they show that PFKFB3 stabilization due to mTORC1-dependent Cdh1 phosphorylation and APC inhibition results in increased intracellular ATP levels, possibly as a way to sustain resumption of cell cycle.

However, there are notable weaknesses that need to be addressed mechanistically. In particular, the data provided in support of mTORC1-dependent Cdh1 phosphorylation are weak and unconvincing. Also, whether phosphatase-mediated Cdh1 dephosphorylation represents a specific regulatory step, as opposed to baseline activity, is not clarified. Specific points follow.

1- 2-DG is used to support the need for glycolysis in cell cycle re-entry. However, this compound has pleiotropic effects and can induce cell death. Orthogonal methods to test the requirement for heightened glycolysis should be tested, such as reducing glucose levels, switching cells to galactose, knocking down various enzymes along the glycolytic cascade and/or the main glucose transporters (i.e. GLUT1).

2- The direct interaction of Cdh1 with mTOR (Fig. 2d) is unconvincing. mTORC1 substrates do not stably interact with the kinase, so as to favor rapid catalysis. Also, empty IgG as the negative control is not appropriate. An unrelated should be used in the negative control lane. Same issue in Fig. 2e

3- The kinase assay in fig. 2f is problematic. mTOR requires accessory subunits, typically Raptor, to recruit substrates to the kinase active site, both in cells and in vitro. This assay was carried out with the mTOR kinase domain alone, which is not evidence of specific phosphorylation, but merely a sign that, with enough kinase and protein mixed together, some phosphate transfer can occur.

4- Do other kinases (e.g. CDK1, ERK) fail to in vitro phosphorylate Cdh1 when used under identical conditions as in 2f?

5- 2h: the decreased interaction of Cdh1 S129A mutant with mTOR is also puzzling: the recruitment motif and phosphorylation motif on virtually all mTORC1 substrates are separate, in other words, substrates are not recruited to mTORC1 via the phospho-site but via the TOS motif (PMID: 12747827). Does Cdh1 harbor a putative TOS-like motif, and would its mutation decrease interaction with mTORC1?

6- The authors provide some (relatively weak) data suggesting that mTOR-dependent phosphorylation of S129 decreases Cdh1 interaction with the APC core (e.g. ED 3i). Given that the APC-Cdh1 interaction has been dissected using both biochemical and structural approaches, it is important that the authors provide some mechanistic rationale as to how S129 phosphorylation (which occurs in a predicted unstructured region of Cdh1) would interfere with binding of the ordered WD40 domain to the APC core.

7- If mTORC1 is the kinase that directly phosphorylates Cdh1, this process should be enhanced in cells deleted for negative mTORC1 regulators such as Tuberous Sclerosis Complex 1/2 or GATOR1. Does deletion of either of these complexes lead to constitutive Cdh1 phosphorylation, APC inactivation and more efficient cell exit from quiescence? Do these deletions phenocopy expression of the phospho-mimetic S129D mutant of Cdh1?

8- The significance of the protein phosphatase data is unclear. The authors imply that a specific protein phosphatase becomes activated at 4h post-induction, but such phosphatase is not identified. It also remains unclear whether a specific activation mechanism is involved, versus 'baseline' phosphatase activity eventually dephosphorylating Cdh1 in a non-regulated manner.

Version 1:

Reviewer comments:

Referee #1

(Remarks to the Author)

The revised manuscript by Paul et al. has been significantly strengthened through extensive revisions and the addition of new data, effectively addressing previous major concerns. The authors have provided substantial evidence supporting their model of mTOR-mediated transient APC/C inactivation as necessary for cell cycle entry. This research offers crucial insights into the coordination of cellular metabolism and cell cycle regulation. I am convinced that the manuscript is now poised for publication, pending the minor but crucial adjustments described below, to ensure clarity and precision in the presentation

and interpretation of the data. I anticipate that it will be well-received by the general readership of Nature.

1. While it is commendable that quantification and statistical analyses are now provided for most data, separating the representative data and the corresponding quantifications into main and supplementary data is not ideal. Presenting them adjacent to each other in the same figures would facilitate better understanding for readers and is therefore recommended.
2. In lines 20-22 in the summary, "Simultaneously, cells activate APC/C." is confusing, as this can be interpreted as APC/C being inactive in quiescent (G0) cells and becoming activated upon cell cycle re-entry, prior to the G1/S transition. However, as the authors wrote in the main text (lines 46-52), APC/C-CDH1 is already active in G0 cells and remains active until the G1/S transition. The authors should revise these sentences to avoid confusion. Similarly, the meaning of the sentence in lines 55-56, "Similarly, if the APC/C is inactivated..." is unclear and should be clarified.
3. In Figures 1a, 1c, 1j, 3a, and 3h, the placement of "Mitogens" in the diagrams does not seem appropriate. It would be more consistent with the final model in Figure 5e to position it above Glycolysis and APC/C-CDH1 or next to G0 before Glycolysis.
4. Regarding minor point 5, the calculation of "33%" is still not entirely clear from the explanation in the text (lines 104-107) and Figure 1h, i. The value on the x-axis in Figure 1h, labeled as "slope of APC/C degron rise (au/hr)" spanning from 0 to 100, is unclear and requires further explanation. The x-axis in Figure 1i should perhaps span from 0 to 100.
5. In Extended Data Fig. 5a, more description is needed to interpret the data in the figure and in the legend. For instance, which peak corresponds to which modified or unmodified peptides?
6. In Extended Data Fig. 5h, n=3 should be n=4.
7. The sentence in lines 180-185 is excessively long and should be revised for clarity.
8. In L196-197 and Extended Data Fig. 6a, b, while the pmTOR/mTOR ratio may not be, the phosphorylated mTOR level (therefore, active mTOR), as well as p4EBP1 level, is clearly transiently increased around 3-7 hours after mitogen stimulation. The potential regulation of mTOR activity, in cooperation with subsequent PPase activation in the control of APC/C-CDH1 activity, cannot be excluded. The text should be amended accordingly.
9. In Figure 3e, mTOR inhibition after mitogen stimulation appears to inhibit APC/C inactivation in a graded manner, according to the timing of rapamycin addition, rather than a flipped switch manner (The partial APC/C inactivation observed in the 1-hour mTOR inhibitor addition condition suggests a more gradual effect). As Reviewer #2 mentioned, biological phenomena are hardly black and white. The authors should remove "completely" in line 213.
10. Fig. 5a and Extended Data Fig. 10c appear to be equivalent. Consider removing the redundancy.

Referee #3

(Remarks to the Author)

re. Reviewer 3's comments: the authors have addressed the concerns raised in the initial submission. In particular, the connection between mTOR-dependent Cdh1 phosphorylation and APC inhibition has been strengthened. I have no further comments.

re. Reviewer 2's comments:

Points 1-4 have been addressed satisfactorily.

Point 5, glycolytic rate, has not been answered properly, as the methodologies employed (whether ATP concentration measurements or fluorescent biosensors) do not represent the gold standard in the field. Metabolic rate is most accurately measured using mass spectrometry (metabolomics) with isotopically labeled tracers, as described here: <https://www.sciencedirect.com/science/article/pii/S0092867418303878> and in many other publications.

Many metabolic experiments were carried out in previously quiescent cells that were simply stimulated with mitogens (i.e. fig.4g-i). These experiments tell very little, as it is a well-known and expected fact that mitogens will increase glycolytic rate via a myriad of mechanisms, such as AKT-stimulated glucose uptake. The key experiment here is to show that changing Cdh1 phosphorylation status has noticeable effects on these parameters, as it seems to do for ATP levels and glycoPER (extended data Fig 9h-i).

To summarize, glycolytic metabolite flux analysis with appropriate isotopically labeled tracers should be carried out in cells in which the phosphorylation status of Cdh1-T129 has been manipulated through mutagenesis.

A second point is to more strongly link mTOR-dependent APC regulation to PFKFB3 degradation and to downstream alteration in glycolytic rate. A key hypothesis is that transient stabilization of PFKFB3 is what drives the spike in glycolysis. If so, expressing a PFKFB3 protein mutated in the APC degron should blunt this effect, as the basal (non-mitogen stimulated) glycolytic rate should be higher than in control cells (and, perhaps, these cells may be less quiescent).

This Reviewer is aware that these requests impose significant extra work on the already large body of experiments provided by the authors. However, given the provocative and potentially highly impactful model they put forward, which in the authors' mind would supersede the myriad other ways through which mitogens can link cell proliferation and metabolism, it is essential that the key statements are thoroughly tested.

Version 2:

Reviewer comments:

Referee #3

(Remarks to the Author)

The authors have satisfactorily addressed my remaining concerns. I recommend acceptance of this interesting and innovative manuscript.

General response to all reviewers:

We thank all the reviewers for providing such thoughtful feedback on our manuscript. We aimed to address each point raised by the reviewers with appropriate experiments as well as a thorough explanation. In this revised manuscript, we have included **58 new or revised figure panels (+3 rebuttal figures)**. By incorporating your suggestions, we believe the manuscript is now much strengthened. Below you will find a point-by-point response to each of the suggestions that were raised. Your comments are shown in *italics* and our responses are shown in **blue**.

Referee #1 (Remarks to the Author):

In the present study, Paul et al. attempt to answer a fundamental question concerning the intricacies of cellular metabolism and cell cycle regulation - specifically, how cells coordinate the glycolytic switch and cell cycle entry from a state of cellular quiescence.

The authors began by addressing existing discrepancies in the current understanding of cellular mechanisms, focusing on three key observations: 1) the necessity of glycolysis for cell cycle entry during the G1 phase, 2) the inhibition of glycolysis through PFKFB3 degradation mediated by the Anaphase Promoting Complex/Cyclosome (APC/C), and 3) the continuous high activity of APC/C throughout the G1 phase until its sharp inactivation upon transitioning from the G1 to the S phase.

Upon reassessment, the authors found that the third observation, which suggested that APC/C remains active throughout the G1 phase, is not entirely accurate. Instead, they discovered that the APC/C undergoes a period of transient inactivation during the G1 phase prior to its full inactivation upon the G1/S transition. This key insight corrects a previous oversight by one of the current authors in a previous study (Cappell et al., 2016, Cell), exemplifying commendable scientific integrity through this diligent and transparent revisitation of prior work.

Expanding upon this initial observation, the authors provide substantial evidence for a model in which APC/C is transiently inactivated to promote the activation of glycolysis, essential for the initiation of the S phase. They reveal that this transient inactivation of APC/C is regulated by the mammalian target of rapamycin (mTOR), a well-known regulator of cell growth and metabolism. This regulation is facilitated through direct phosphorylation of CDH1 by mTOR itself, which is subsequently reversed by phosphatase activity during the reactivation of APC/C. The authors also identify PFKFB3, an essential enzyme in the glycolytic pathway previously shown to be an APC/C substrate, as a key mediator through which transient APC/C inactivation triggers a metabolic shift from oxidative phosphorylation to glycolysis, essential for cell cycle re-entry. Lastly, the authors clarify how this APC/C regulation, by coordinating glycolysis, ATP production, and other biosynthetic processes with G1 phase progression, effectively facilitates cell cycle entry and colony formation. The study underscores the critical role of transient APC/C inactivation, downstream of mTOR, in orchestrating the metabolic changes during the cell cycle transition from a quiescent state back into the cell cycle.

Overall, the manuscript presents groundbreaking research that contributes significantly to the understanding of the interplay between cell metabolism and cell cycle progression. The implications of these findings are significant, given the pivotal roles both the APC/C and mTOR

signalling pathway play in cell growth, proliferation, and cancer. By bridging the gap between the understanding of the cell cycle and metabolism, their findings are likely to appeal to a large general audience and stimulate further investigations into the regulatory nexus between the cell cycle, metabolic control, and disease states. The narrative is well-constructed, the study is thorough and robust, and the data provided are largely high quality and convincing. Therefore, the scope and impact of the manuscript fit well with the criteria for publication in Nature.

However, while this research has substantial merit, there are several concerns that currently prevent a full endorsement of the manuscript for publication. These concerns primarily relate to the experimental design, interpretation of results, as well as a lack of clarity on the consistency across experimental replicates, which is partly due to the way the data is presented. I will elaborate on these issues in the subsequent sections. I believe that addressing these issues will significantly strengthen the manuscript, making it more deserving of publication in such a prestigious journal.

We appreciate the reviewer's valuable inputs. We have incorporated additional experiments and quantification of replicates to ensure the significance and reproducibility of the data. We have also performed some key additional experiments that further strengthen our initial observations and support our model.

Major Concerns:

1. Many of the results heavily rely on Immunoblot data. Although the experiments are stated to be performed multiple times ($n = 2$ or 3), in most cases, only one representative blot is presented. The differences in band intensities are fairly subtle in some data sets, leading to potential questions regarding variability and reproducibility. For these data to be validated, band intensities should be quantified across all replicates and mean values with variabilities, or each value, should be shown.

We thank the reviewer for making this suggestion. We have now provided quantification and statistics for many of the immunoblot data presented in the manuscript. We have focused primarily on those figure panels that show relatively subtle differences as pointed out by the reviewer. We think the inclusion of these quantifications demonstrates the reproducibility and reliability of our results. Given the transient nature of this phenomenon, bulk measurements via western blotting are expected to yield small but significant differences, and we agree that quantification combined with statistical analysis is important to include.

2. Although glycolysis is associated with cell proliferation, its general requirement for cell cycle entry hasn't been fully established. While this was demonstrated in MCF-7 cells (extended Fig.1c), is this requirement applicable to other human cells? To ascertain the importance of the present findings, testing should be done across several other cell lines, including untransformed cells. Further, what would occur if glycolysis isn't turned off and remains active, for example, by overexpressing PFKFB3 or TOR hyperactivation?

We agree that demonstrating the requirement for glycolysis in cell cycle entry in multiple cell lines is critically important. We have now included data to this effect for three different non-transformed cell lines: MCF-10A, RPE1, and MEFs. Given that Reviewer 3 (Major comment 1) also pointed out 2DG can have pleiotropic effects and induce cell death, we decided to test these cell lines using orthogonal approaches to test the requirement for glycolysis, specifically by

depleting glucose from the media or providing galactose as the primary carbon source (New Extended Data Fig. 1e-i). We found that depleting glucose or providing only galactose as the carbon source blocked mitogen-stimulated cell cycle entry in MCF-10A, RPE1, and MEFs (New Extended Data Fig. 1g-i). Interestingly, we also blocked OXPHOS in MCF-10A cells as an additional control and found that cells could still activate CDK2 and enter the cell cycle, indicating that cell cycle entry is majorly dependent on glycolysis (New Extended Data Fig. 1d).

As suggested by the reviewer, we also tested the effect of mTOR hyperactivation on cell cycle entry. We used siRNA to knockdown two negative regulators of mTOR: NPRL2 or TSC1. We found no significant difference in the percent of cells activating CDK2 in siNPRL2 or siTSC1 treated cells compared to siControl treated cells, indicating that mTOR might be maximally activated upon mitogen stimulation, and that depletion of negative regulators does not further increase the ability of cells to enter the cell cycle (Rebuttal Fig. R1).

3. The role of phosphatases (PPases) in the reactivation of APC/C after its transient inactivation and the significance of this reactivation for cell cycle progression is not entirely convincing. Observations made in this paper using PPase inhibitors could be due to changes in the global protein phosphorylation status, potentially irrelevant to physiological conditions. Further studies, such as using PPase inhibitors at different time points (it might induce CDH1 phosphorylation even in G0) or measuring PPase activity, could help address this.

We agree with the concerns raised by the Reviewer that our observations after treatment with the PPase inhibitors could be simply due to large global changes in protein phosphorylation status and not necessarily specifically due to Cdh1 de-phosphorylation. To address this concern as well as a similar comment made by Reviewer 2 (main point #4), we have performed several new experiments.

First, we have assessed the effect of PPase inhibitors on APC/C activity at different time points as suggested by the Reviewer. Despite adding the PPase inhibitor either simultaneously or as early as 1 hour after the addition of mitogens, the slope of the APC/C degron level accumulation does not deviate from the untreated control until 4 hours after mitogen addition, indicating that the phosphatase is likely only active until approximately 4 hours after mitogen addition (new Extended data Fig. 6f). Furthermore, treatment with the PPase inhibitors at time points later than 4 hours led to a rapid accumulation of the APC/C degron levels.

Second, to assess whether the phosphatase may be functioning during G₀, we treated serum starved cells with the phosphatase inhibitor and performed live-cell imaging of the APC/C degron biosensor. Notably, we did not observe accumulation of the APC/C degron biosensor, indicating the phosphatase is likely not regulating APC/C activity during G₀ (new Extended data Fig. 6e, see red line).

Third, to address the issue that the phenotypes we observed may be due to global phosphorylation changes, as well as the comment made by Reviewer 2 (main point #4), we have now performed a rescue experiment with the Cdh1-T129A mutant. The rationale behind this experiment is that if the increased accumulation of the APC/C degron sensor is due to large changes in global protein phosphorylation status, then expressing Cdh1-T129A should not be able to dampen this effect. We found that ectopic expression of Cdh1-T129A can rescue the protein phosphatase mediated sustained accumulation of the APC/C degron (new Extended data Fig. 6d), suggesting that the increased accumulation of the APC/C degron is not due to a change in global protein phosphorylation, rather dephosphorylation of APC/C Cdh1.

4. Can transient PPase inhibitor treatment or overproduction of CDH1T129D cause any defects in subsequent cell cycle phases

To address this question, we have now performed two additional experiments. First, we measured EDU incorporation to assess DNA replication in cells expressing Cdh1-T129D (Rebuttal Fig. R2a). We found that expression of Cdh1-T129D does not alter EDU incorporation compared to empty vector, indicating there are no defects in S phase. Second, we did a long-term time-lapse experiment to look at the rate of mitosis for the first cell cycle after mitogen stimulation. We found only a slight delay in the overall time it took cells to reach mitosis, indicating only minor effects on subsequent cell cycle phases (Rebuttal Fig. R2b,c).

Finally, to investigate the potential long-term consequences of transient Cdh1-T129D overexpression on subsequent cell cycles we did a colony forming assay using an inducible Cdh1-T129D vector (see Extended Data Fig. 10d). We generated MCF10A cells expressing doxycycline-inducible Cdh1-T129D or Cdh1-T129A. We treated these cells with doxycycline (Dox) following mitogen stimulation and then washed off the doxycycline 8 hours later. Cells were allowed to grow for 7 days before staining with crystal violet. Compared to control cells, cells that transiently expressed Cdh1-T129D were only slightly growth delayed, whereas cells that transiently expressed Cdh1-T129A were significantly delayed.

5. The evidence for the mechanisms and role of APC/CCDH1 inhibition by mTOR is not wholly convincing. Further studies using wild-type CDH1 and examining the effect of mTOR addition on CDH1's association with APC/C and/or APC/C's E3 activity could strengthen the presented evidence.

We thank the reviewer for this suggestion. We have now performed additional in vitro ubiquitination assays examining the effect of mTOR addition on APC/C-Cdh1's E3 ligase activity (New Fig. 2k, l). We performed APC/C-UBE2C-dependent ubiquitination assays in the presence of Cdh1-WT, Cdh1-T129A, or Cdh1-T129D. We observed that the Cdh1-T129D mutant was less effective in promoting ubiquitination of substrates when compared to either Cdh1-WT or Cdh1-T129A (see lane 4). When we added recombinant mTOR to Cdh1, the effectiveness of Cdh1-WT in promoting ubiquitination of substrates was reduced to similar levels as Cdh1-T129D (compare lane 4 and 5). We included quantification of the three

independent experiments in new Fig. 2l. Notably, Cdh1-T129D or Cdh1 mixed with recombinant mTOR did not completely reduce APC/C activity towards its substrate, consistent with our model that the APC/C partially, rather than fully, inactivates during exit from quiescence.

6. To establish that mTORC1, not mTORC2, is responsible for the transient APC/C inactivation, experiments examining the effects of mTORC1-specific component depletion on APC/C activity and cell cycle entry should be performed. How does mTOR generally recognize its substrate?

To better establish mTORC1, not mTORC2, as being responsible for transient APC/C inactivation, we took two approaches (See new Extended Data Fig. 3j-m). First, we used siRNA to knockdown the mTORC1 core component Raptor or the mTORC2 core component Rictor. We observed reduced transient APC/C inactivation as measured by the APC/C degron reporter in cells treated with siRaptor (mTORC1) but not siRictor (mTORC2). Second, we treated cells with the mTORC1-selective inhibitor RMC6272 and also observed a reduction in transient APC/C inactivation. Both of these results indicate that mTORC1 is responsible for the transient APC/C inactivation we observe in cells exiting quiescence.

mTOR usually recognizes its substrates via TOS/RIAP/NPXY motifs¹. However, mTOR is also reported to phosphorylate substrates that do not contain any of these consensus motifs¹⁻⁴. In fact, many mTOR substrates do not harbor these interaction motifs and their interaction with mTOR is TOS-independent. Cdh1 does not have a TOS motif, indicating its interaction with mTOR is also TOS-independent.

Minor Points:

1. To assist non-expert readers, the authors should specify early on that their subject is APC/CCDH1, not APC/CCDC20, given that these have distinct functions and regulations.

We thank the reviewer for this suggestion. We have edited the text in both the Abstract as well as the opening introductory paragraph to clearly specify this study refers to APC/C-Cdh1 and not APC/C-CDC20.

2. Line 67: Could you clarify the meaning of "Proper" (also in the title)? Please consider using a more descriptive term to articulate the phenotype.

We edited the text to remove the term "Proper". We agree it is too vague.

3. Line 75: A number appears to be missing before "hours." Being more specific would enhance clarity.

This error has been fixed. The modified lines are "Strikingly, we observed that the APC/C partially and transiently inactivates during the G0/G1 transition (Fig. 1d-f, Extended Data Fig. 2b, and Supplementary Video 1), approx. 6-8 hrs before the G1/S transition, when the APC/C fully inactivates" (Line 84).

4. Line 82: Revise "fig1k-n" to "fig. 1k-o".

This error has been fixed.

5. Line 91-92: The calculation of the 33% value needs clarification, along with its implications. If not necessary, consider removing it.

Thank you for bringing this confusion to our attention. We have added additional text (new line 99-106) to better clarify this point. We think this is a minor, but still important, point to make since it corroborates some of the biochemical assays presented later in the manuscript.

6. Line 216-217: *The term "post-transcriptional" might be more appropriate than "post-translational", particularly because mTOR is known to regulate translation.*

We appreciate the reviewer's suggestion and have changed the text accordingly. (new line 253)

7. *Extended Fig. 1i/j: Considering one of the major concerns, following release from MEK inhibition or contact inhibition, the "transient" nature of APC/C inactivation seems less pronounced. Many cells seem to exhibit a two-step or gradual inactivation (slow and f of APC/C, instead of transient inactivation followed by reactivation). Please provide clarification.*

We thank the reviewer for bringing this to our attention. The "two-step" gradual APC/C inactivation depicted in the original figure is simply an outlier experiment and due to experimental variability. We now include more representative examples for these figures that demonstrate a similar "one-step" transient inactivation followed by reactivation (new Extended data Fig. 2f,g).

8. *In the legend for Fig. 1d, please specify what the top and bottom panels represent.*

Our apologies. This error has been fixed.

9. *Fig. 2d: Using prey as a control in the coIP experiment may not be ideal. If the protein is sticky, it would attach to the beads. Use -bait (CDH1) as control.*

We addressed this issue two different ways. First, we performed the reciprocal IP where we pulled down on mTOR and probed for Cdh1 (Extended Data Fig. 4a) in order to help control for mTOR simply being a sticky protein. Additionally, as suggested by Reviewer 3 (comment #2), we performed IP experiments using two different unrelated proteins. We pulled down with an EGFR antibody (new Extended Data Fig. 4b) and as well as an HA antibody (new Extended Data Fig. 4c). In both cases, we did not observe mTOR or Cdh1 binding non-specifically to the negative control antibody-coated beads.

10. *Ex. 3c, g: The specificity of the phospho-specific antibodies has not been validated. pSer/Thr (ex. 3c) displays relatively weak signals on CDH1 at 0hr (quiescence), but pThr already exhibits relatively high signals (ex. 3g). pThr also seems to recognise T129A. Ideally, a T129 phospho-specific antibody should be generated. Alternatively, the specificity against phosphorylation should be validated by treating the samples with PPases to check for loss of detection.*

We thank the reviewer for these helpful suggestions for further validating the phospho antibodies. We attempted to generate a phospho-specific antibody for T129; however that region of the protein is predicted to have very low antigenicity and several companies we contacted to generate the antibody gave the project a very low chance of success. Notably, despite many other previous papers describing Cdh1 phosphorylation in the same N-terminal unstructured region by other kinases (eg CDK2, ERK1), no one has succeeded in making a phospho-specific antibody in this region of the protein, highlighting the probable technical challenges associated with generating a phospho-specific antibody for this particular site. Given the high cost for generating the antibody and low chance of success, we took the alternative approach of using IPs and validating pThr antibodies.

As per the Reviewer's suggestion, we have validated the pThr antibody using lambda phosphatase treatment (New Extended Data Fig. 5c). We observed a significant decrease of phosphorylated Cdh1 upon treatment with lambda phosphatase, suggesting that the pThr antibody used in the study specifically recognizes phosphorylated Cdh1. However, the band did not go away entirely, suggesting a small degree of background/non-specific staining or incomplete de-phosphorylation by lambda phosphatase.

11. L130-131: For a similar reason as above, without a negative control with pTyr, it cannot be definitively concluded that there is no Tyrosine phosphorylation. Please moderate the statement.

We agree with the Reviewer and have moderated the statement so as to not make a definitive statement about Tyrosine phosphorylation (new line 147).

12. Ex.3c, L125-6: The reduction of Apc2 was very subtle and seemed to recover by 4h, even though CDH1 remained phosphorylated. If the hypothesis is that Tor-dependent phosphorylation causes APC/C to dissociate from CDH1, this does not align with the observed data. It would be beneficial to test in vitro if Tor-dependent phosphorylation can indeed dissociate CDH1 from the APC/C complex and reduce its E3 activity (also mentioned in the general comment).

We thank the reviewer for this suggestion and agree that transient and subtle changes are better observed in vitro. We have now performed additional in vitro ubiquitination assays examining the effect of mTOR addition on APC/C's E3 ligase activity (New Fig. 2k, l). We performed APC/C-UBE2C-dependent ubiquitination assays in the presence of Cdh1-WT, Cdh1-T129A, or Cdh1-T129D. We observed that the Cdh1-T129D mutant was less effective in promoting ubiquitination of substrates when compared to either Cdh1-WT or Cdh1-T129A. When we added recombinant mTOR to phosphorylate Cdh1, the effectiveness of Cdh1-WT in promoting ubiquitination of substrates was reduced to similar levels as Cdh1-T129D. We included quantification of the three independent replicated blots in new Fig. 2l. Notably, Cdh1-T129D or Cdh1 mixed with recombinant mTOR did not completely reduce APC/C activity towards its substrate, consistent with our model that the APC/C partially inactivates during exit from quiescence.

13. Fig. 2H: Is T129 located near the C-box, which has been shown to be important for APC/C binding and/or activation of its activity? Including the domain structure of CDH1 in the figure, complete with WD40 repeats and C-box and IR motifs, would be beneficial.

Thank you for this suggestion. We now included a diagram of CDH1's domain structure and also point out the location of phosphorylation sites that were previously identified by other kinases that are also known to disrupt the binding of CDH1 with the APC/C (see New Extended Data Fig. 5m). T129 is indeed located within the cluster of these other phosphorylation sites.

14. Fig. 3g: The lines are difficult to distinguish. Please alter the line styles and colors for easier interpretation.

We have now adjusted the line colors and pattern to make them more distinguishable.

15. Ex. 6d: Depletion of PFKFB3 by siRNA does not seem efficient, yet it still shows inhibition of ATP synthesis comparable to that of PFK15. This finding requires explanation.

We thank the reviewer for pointing out this discrepancy. For the ATP assays we used 40 nM of siPFKFB3 and now have updated Extended Data Fig. 9g with a western blot of cell lysates

treated with 40 nM of siPFKFB3 to better match this data. The knockdown shown in Extended Data Fig. 9g is now more efficient.

Referee #2 (Remarks to the Author):

The molecular processes regulating the transition between the quiescent and proliferative cell state, i.e. cell cycle entry, are still not fully understood. Through an impressive amount of work, here the authors suggest the existence of a mechanism that links mTOR as a key signaling pathway to the activity of the anaphase-promoting complex/cyclosome (APC/C) as a major cell cycle regulator, and to the core metabolism, i.e. glycolysis. This link, if convincingly demonstrated, would be one of the first established molecular links between the core cell cycle machinery and the core metabolic machinery in vertebrates, and hence an important scientific study with broad implications for both the cell cycle field as well as the metabolism field. However, several conclusions of the manuscript are currently not well supported by the data in the manuscript. The authors would need to address the raised concerns satisfyingly before publication is warranted.

General remarks on how the manuscript is written:

In order to sell their story in an attractive manner, the authors sketch a “paradoxical situation”. The paradoxical situation emerges if one considers things in a “black-and-white” manner, e.g. “glycolysis is necessary for cell cycle start”. However, “glycolysis” is likely not a binary variable (i.e. on or off) but is rather a gradual one with glycolysis having a certain flux/activity. Also, the statement of “APC/C inhibits glycolysis” is likely not a “black-and-white” question either. Everybody in biology knows that hardly anything is so black and white and that it is rather a question of activities, rates, or fluxes, i.e. more graded. But by putting it so black and white at the beginning (in words in the introduction) and in Fig. 1b and c, readers might be annoyed. In fact, as the authors also show even quiescent cells retain some glycolytic activity and then if the authors’ black-and-white statement of “glycolysis is necessary for cell cycle start” would be true, then also quiescent cells could exit quiescence.

We agree with the Reviewer and apologize for not using more accurate language. We certainly did not mean to convey that cells rely entirely on glycolysis or entirely on OXPHOS at various points of the cell cycle. Rather, we meant to convey there is a shift in the relative rates of glycolysis and OXPHOS during the exit from quiescence and that “APC/C partially inhibits glycolysis”. We have edited the text in the introduction to acknowledge the more nuanced and accurate point made by the Reviewer.

Are conclusions valid/solid?

1. In vivo Cdh1 phosphorylation dynamics: In vitro, the authors convincingly show that Cdh1 and mTOR physically interact, that mTOR phosphorylates Cdh1 at T129 and that this phosphorylation results in a mild decrease in APC/C-Cdh1 activity. However, when it comes to Cdh1 phosphorylation dynamics in cells exiting quiescence in vivo, the overall phosphorylation dynamics are less clear – this is in parts because the authors are not using a phospho-specific antibody against the T129 site but rely on Cdh1 immunoprecipitation in combination with a pan-phospho-threonine/serine or pan-phospho-threonine antibodies. The only direct evidence that T129 is the only regulated phospho-site on Cdh1 during the transient inactivation is provided in Figure 2i, where they probe Cdh1 and Cdh1 T129A immunoprecipitated from cell extracts made

4 hours after release from quiescence and show that the wt but not the T129 mutant is recognized by an anti-phospho-threonine antibody suggesting that T129 indeed is the only threonine site phosphorylated in vivo at 4 h time point. However, the time courses in Extended Figures 3C and 3g seem more puzzling. Extended Figure 3C shows an increase in phospho-Cdh1. However, the experiment is cut short at 4 hrs, and it is not possible to assess whether Cdh1 phosphorylation remains high or decreases at later times. Moreover, the antibody used (pan anti pSer/Thr) does not allow one to conclude that it is specifically the T129 phosphorylation (could be other mediated by additional phospho-serines). The experiment in Extended Figure 3g shows a longer time course (up to 8 h) with the appropriate pan-anti pThr antibody, however, the changes in phosphorylation state are much less pronounced in this experiment with a suspiciously high phosphorylation signal already at 0 h and a very mild increase at 4 and mild decrease at 8h, respectively. To convincingly demonstrate the phosphorylation dynamics of Cdh1 by mTOR the authors should at the very least repeat the experiment shown in Extended Figure 3c with the pan-anti pThr antibody including at least another 2-3 timepoints between 4 and 10 hours and ideally including the T129A mutant as an additional control. The immunoblot repeats should be quantified. Mass spectrometry analysis of Cdh1 isolated from 4 hours after release from serum starvation demonstrating that indeed T129 is the only prominent phospho-site would further strengthen the authors argument.

We thank the reviewer for their suggestion. We attempted to generate a phospho-specific antibody for T129, however that region of the protein is predicted to have low immunogenicity (one company even told us the region had “terrible immunogenicity”) and several companies we contacted to generate the antibody gave the project a very low chance of success. Notably, despite many other previous papers describing Cdh1 phosphorylation in the same N-terminal unstructured region by other kinases (eg CDK2, ERK), no one has succeeded in making a phospho-specific antibody in this region of the protein, highlighting the technical challenges associated with generating a phospho-specific antibody for this particular site. Given the high cost for generating the antibody and low chance of success, we took the alternative approach of using IPs and pan-phospho antibodies.

As per the Reviewer’s suggestion we have performed immunoprecipitation experiments with both Cdh1-WT and Cdh1-T129A from 0-10 hr at 2 hr intervals and assessed Cdh1 phosphorylation using the pan-phospho-Threonine antibody (New Extended Data Fig. 5e,f). We have found an increase in Cdh1 threonine phosphorylation at 2 and 4 hr after mitogen stimulation followed by a decrease in the levels of Cdh1 threonine phosphorylation at 6 and 8 hr. Furthermore, upon mutation of T129A, the phosphorylation dynamics are lost, with only non-specific staining remaining.

2. Sustained mTOR activity following mitogen stimulation: This is an important claim and is based on the data of Extended Fig. 4a. However, it seems that mTOR abundance is not constant over time in the shown immunoblot, and the phosphorylation of mTOR and p4EBP1 also fluctuates. It is difficult to assert the mTOR activity dynamics from this blot. Some type of quantification would be necessary here (e.g. phospho over mTOR bands). The other two replicates should be also quantified to ensure reproducibility of the result.

We acknowledge the reviewer's suggestion and have quantified the phospho-mTOR/mTOR ratio accordingly (New Extended Data Fig. 6b). We find no significant change of phospho-mTOR/mTOR ratio during the 3-5 hr timepoint, coinciding with APC/C reactivation. However, our findings reveal fluctuations in mTOR activity post 4-5 hr, with a notable decrease observed

at 6 hr. Therefore, while mTOR activity fluctuates, it is unlikely to impact APC/C reactivation. In addition, a recent study has investigated mTOR activity dynamics in cells exiting quiescence after mitogen stimulation using a live-cell mTOR biosensor⁵. The authors observed sustained mTOR activity after stimulation with EGF for up to 6 hr, which is past when we observe APC/C re-activating (See Figure 1F from Sparta et al, 2023) in corroboration of our results.

Consequently, we have revised our statement to reflect this, stating “We assessed the ratio of phospho-mTOR/mTOR to quantify the mTOR activity and found no significant change in phospho-mTOR/mTOR ratio during the time when APC/C transiently reactivates (3-5 hr after mitogen stimulation) consistent with previously published live-cell imaging experiments⁵” (Line 194-197).

3. Excluding other regulators of the APC/C: The authors observe that Cdh1 T129D cells still shows transient APC/C inactivation probably due to presence of endogenous Cdh1 in the background, but this could also suggest that either additional regulators inactivate the APC/C, or that mTOR controls additional phosphosites on Cdh1. The link between transient Cdh1 T129 phosphorylation and APC/C inactivation is an important part of the mechanism described here, and it is important that it is clearly established (e.g. using Cdh1 siRNA to eliminate the endogenous Cdh1).

We agree with the reviewer that the Cdh1 T129D cells shows transient APC/C inactivation due to the presence of endogenous Cdh1. To better address this point, we generated a cell line expressing a doxycycline-inducible Cdh1-T129D and designed siRNA targeting the 3' UTR of Cdh1. When we expressed Cdh1-T129D and knockdown the endogenous Cdh1 we observed a sustained accumulation of the APC/C degron, establishing a clearer link between Cdh1-T129 phosphorylation and transient APC/C inactivation (new Extended Data Fig. 5o).

4. Time-delayed activation of a counteracting phosphatase mediates re-activation of the APC/C: The authors use sodium fluoride and sodium ortho-vanadate to inhibit phosphatase activity and observe a sustained inactivation of the APC/C in most cells. They conclude that an unidentified phosphatase acts on T129 in a time-delayed manner resulting in the re-activation of APC/C after the initial phosphorylation and inactivation by mTOR. However, the used phosphatase inhibitors constitute a massive, global perturbation impacting the phosphorylation state of numerous proteins and regulators and hence does not allow for this specific conclusion. The authors would need to demonstrate that the observed sustained inactivation is indeed solely mediated via Cdh1-T129 phosphorylation (e.g. by performing a 'rescue' experiment with the Cdh1-T129A mutant or mTOR inhibition). Furthermore, ortho-vanadate is thought to be a tyrosine phosphatase inhibitor – considering that the authors excluded tyrosine phosphorylation as a regulatory mechanism, the inhibitor seems a suboptimal choice.

We agree with the concerns raised by the Reviewer that our observations after treatment with the PPase inhibitors could be simply due to large global changes in protein phosphorylation status and not necessarily specifically due to Cdh1 de-phosphorylation. To address this concern as well as a similar comment made by Reviewer 1 (main point #3), we have performed several new experiments.

To address the issue that the phenotypes we observed may be due to global phosphorylation changes, as well as the comment made by Reviewer 1 (main point #3), we have now performed a

rescue experiment with the Cdh1-T129A mutant as suggested. The rationale behind this experiment is that if the increased accumulation of the APC/C degron sensor is due to large changes in global protein phosphorylation status, then expressing Cdh1-T129A should not be able to dampen this affect. We found that ectopic expression of Cdh1-T129A can rescue the protein phosphatase mediated sustained accumulation of the APC/C degron (new Extended data Fig. 6d), suggesting that the increased accumulation of the APC/C degron is not due to a change in global protein phosphorylation, rather dephosphorylation of APC/C Cdh1.

We have assessed the effect of PPase inhibitors on APC/C activity at different time points as suggested by the Reviewer 1. Despite adding the PPase inhibitor either simultaneously or as early as 1 hour after the addition of mitogens, the slope of the APC/C degron level accumulation does not deviate from the untreated control until 4 hours after mitogen addition, indicating that the phosphatase is likely only active until approximately 4 hours after mitogen addition (new Extended data Fig. 6f). Furthermore, treatment with the PPase inhibitors at time points later than 4 hours led to a rapid accumulation of the APC/C degron levels.

We agree to the reviewer's comment that in general sodium ortho-vanadate is considered as a tyrosine protein phosphatase inhibitor. However recent articles have shown, both in vitro and in cells, that vanadium complex inhibitors act more as general protein phosphatase inhibitors, with activity towards serine and threonine along with tyrosine^{6,7}. Additionally, the combination of NAF+Na₃VO₄ is widely used as pan protein phosphatase inhibitor, therefore we choose to use the combination, as these experiments were conducted before we had identified the phosphorylation site on Cdh1.

5. Metabolism: A key claim of their model is that RATE through glycolysis transiently increases. To support this claim they argue that the abundance of a rate-limiting enzyme of glycolysis increases and provide Seahorse and ECAR measurements. Beyond, the authors also measured metabolite concentrations (ATP, NADH and pyruvate). There are several concerns about the metabolic aspect of the model:

a. The measurements of the metabolite concentrations are irrelevant for the model as the model is about a metabolic RATE and not about metabolite levels. Note, from metabolite levels one cannot draw any conclusions on metabolic rates: For instance, if ATP levels are high one cannot conclude that rate through glycolysis is high. The concentration of a metabolite emerges from both the rates of production of the metabolite and the rates of consumption of the metabolite.

Furthermore, also the way the authors measure the metabolite concentrations is questionable:

(i) As metabolic turnover rates are high, directly after sampling of cells for metabolite analysis, metabolic activity needs to be immediately quenched to prevent abiotic changes in the metabolite levels. Nothing on quenching is mentioned in the method section (lines 470ff), raising doubts on the overall validity on the reported metabolite concentrations.

(ii) Across the different measured metabolite concentrations, the authors report very different units (c.f. Fig. 4f, j, k). The one metabolite is reported in absolute grams, the other in absolute moles, the third is reported in moles/cell. Data in absolute grams or absolute moles do not make sense. To check the validity of these measurements, it is key that metabolite data are reported in units per cell or per cell volume.

We thank the reviewer for bringing up this point. We agree that concentrations of specific metabolites do not convey any information about metabolic rates for the reasons the Reviewer rightly points out. We originally included some measurements of metabolite concentration (NADH and pyruvate) to try and corroborate our rate measurements, but we agree that it introduces a point of confusion. Therefore, we have removed the data measuring NADH and Pyruvate levels, since this data neither supports or refutes our model. We are still including one measurement of total cellular ATP in cells exiting quiescence (Fig. 4f) as it serves as an initial observation that led us to do more metabolic rate analysis. However, we now plot the data as the change between two time points to report the rate change in ATP levels over time as suggested (new Fig. 4g).

b. As mentioned above, the key question is whether there is indeed an increase in the RATE of glycolysis around the time of the proposed APC/C inactivation. If the authors are right with their story, then it really needs to be an increase in the glycolytic rate (and not a black-and-white point; see comment above). This is because in fact quiescent cells still have glycolytic activity (as the authors also show). Thus, is the data really showing an increase in the glycolytic rate? To this end, the authors use Seahorse measurements (Fig. 4g and h) from which they infer both the contribution of glycolysis and OXPHOS towards ATP generation and perform measurements of ECAR (extracellular acidification rate, correlating with the rate of glycolysis, Fig. 4i). There are several concerns with these measurements:

(i) Both are not direct measurements of the rate of glycolysis but indirect ones. Can the Seahorse assay really without doubt reveal the proportion between glycolytic and OXPHOS-based ATP generation? The differences reported in the proportions are not huge. Thus, assumptions might be critical. As a more direct measurement of the rate of glycolysis, the authors could measure the depletion of glucose from the medium and from this data infer the rate of glycolysis.

We thank the reviewer for this suggestion. Though Seahorse measurements are well accepted in the community as a proxy to measure metabolic rates, we understand the reviewer's concern and performed several new experiments to measure the rate of glycolysis via complementary approaches.

First, following the Reviewer's advice, we evaluated cellular glucose uptake using 2-NBDG. We observed elevated glucose uptake during the transient APC/C inactivation, consistent with an increased rate of glycolysis at this stage of exit from quiescence (new Extended Data Fig. 9f). This data aligns with a previous study by Hung et al, which reported increased labeled glucose uptake in MCF10A cells two hours post mitogen stimulation⁸.

Second, we have measured a proxy for the rate of glycolysis using a live-cell biosensor called HYlight⁹. Briefly, this biosensor is a fusion of a circular permutized GFP tag onto the CggR (central glycolytic gene repressor) protein from *Bacillus subtilis*. The sensor was designed such that binding of fructose 1,6-bisphosphate (FBP) to the HYlight biosensor disrupts fluorescence of the cpGFP tag and allows for the monitoring of real-time changes in intracellular FBP levels in live cells, which the authors have validated as correlating with glycolytic flux. Using this approach, we could monitor changes in the HYlight sensor in single-cells with high time resolution (every 6 minutes) during exit from quiescence. Similar to our Seahorse assays, we observed a transient increase in the HYlight sensor levels after

mitogen stimulation that was occurring at the time of transient APC/C inactivation and was dependent on the presence of glucose (new Fig. 4i and new Extended Data Fig. 9d,e).

Third, we have measured the glycolytic rate at multiple timepoints during the exit from quiescence using the Seahorse assay. We again observed a transient increase in the glycolytic rate at 2 and 4 hr post mitogen stimulation that was lower by 8 hr post mitogen stimulation (new Extended Data Fig. 9h). We did not observe the same dynamics in cells expressing Cdh1-T129A.

Taken together, these complementary experiments support our model that during exit from quiescence, cells transiently increase the rate of glycolysis, and it is regulated by the APC/C.

(ii) As the authors claim that an increase in the glycolytic rate occurs, the authors need to show the absolute rates (of glycolytic and OXPHOS ATP generates) instead of the proportions. Why not plotting the change in glycolytic ATP production rate (i.e. the difference between 2 measurement time points) as a function of time?

We have addressed this comment in a number of different ways.

First, we plotted the change in the total ATP levels between two time points from Fig. 4f and present the data in Fig. 4g.

Second, we show the absolute rate of ATP production from both glycolysis and OXPHOS in Extended Data Fig. 9b.

Third, we have plotted the change in the glycolytic ATP production rate as the difference between these time measurements in new Extended Data Fig. 9c.

Fourth, we have now reported a more direct measurement of glycolytic rate using the Seahorse assay for glycolytic Proton Efflux Rate (glycoPER) in new Extended Data Fig. 9i. We think these were excellent suggestions that help to strengthen our model and more accurately reflect these rate changes over time as cells exit quiescence.

(iii) In Fig. 4h and i, the authors only show data from a single time point. Time courses of ATP production rate and ECAR in the control and in key mutants/perturbations would be needed.

We appreciate the Reviewer's suggestion. We have conducted glycolytic rate and ATP production rate measurements over an extended period in the presence of the Cdh1-T129A mutant. Our results demonstrate that ectopic expression of Cdh1-T129A leads to a reduction in both glycolytic rate and ATP production rate, with the degree of suppression intensifying over time (Extended Data Fig. 9h,i). This underscores the role of transient APC/C-Cdh1 inactivation in regulating the increase in glycolysis upon cell cycle entry.

c. The final “metabolism-related” question is whether an increase in the RATE of glycolysis is indeed caused by reduced degradation of PFKFB3. The authors argue that there is a transient accumulation of this enzyme. Also here, more thorough time course measurements of this protein in the control and in key mutants with more fine-grained sampling and over a longer period of time would be needed. Also, Fig. 4a shows only a “representative” blot. In addition, the bands should be quantified and the data from all independent experiment should be plotted together.

We appreciate the reviewer's suggestion to do longer time courses and to quantify the independent experiments. First, we have quantified the experiments that were in the original manuscript and provide bar graphs with statistical analysis (new Extended Data Fig. 7f and 8a,d). Second, we have now performed long-term time-course experiments to assess the levels of PFKFB3 in absence and presence of Cdh1-T129A during both the first 10 hr after mitogen stimulation (new Extended Data Fig. 8e,f) and during the first 48 hr after mitogen stimulation (new Extended Data Fig. 8g,h). All of these experiments corroborate our results that PFKFB3 accumulates transiently from as soon as 2 hr after mitogen stimulation and starts to decline by 6 hr after mitogen stimulation.

Minor comments:

1. In the mammalian field, the term "cell cycle entry" is reserved for the transition from quiescence to the proliferating state. As the term is very similar to "cell cycle start" and as for instance the yeast field uses this term for the moment when cells commit to the next cell cycle, there is the danger that the term "cell cycle entry" could be considered the same as "cell cycle start". As the authors mention in the discussion (line 282), their finding is only true for the G0/G1 transition (i.e. for "entry" and not for "start"), it would be good if the title and abstract would avoid this potential confusion with readers from other fields, by for instance changing the title from "...ensures proper cell cycle entry" to "... ensures proper exit from quiescence".

We thank the reviewer for bringing this potential point of confusion to our attention. In order to avoid confusion with the yeast field, we have added the word "mammalian" to the title. Since we provide no data to support this finding extending to yeast, we think its appropriate to state our model only applies to mammalian systems at this time.

2. Figure 2C: typo in y-axis label

Thank you for catching this typo. It has been fixed.

3. Fig 2F&G: The source of the GST-Cdh1 is not described in the methods. Furthermore, the His-Cdh1 (Fig 2F) and the GST-Cdh1 (Fig 2G) run almost at the same molecular weight although the GST tag is significantly larger.

We apologize for this confusion. The panel in Fig. 2f was mislabeled. His-Cdh1 is actually tagged both with 6xHis as well as Sumo, accounting for the elevated molecular weight of the protein. The figure labeling has been corrected (now moved to Extended Data Fig. 4e).

4. Figure 4A - what happened to Cdh1 at time point 0h? Cdh1 is present at that time point in other blots (e.g. Extended Figure 3C).

We have now replaced the concerned figure with a more representative blot that shows Cdh1 present at time 0 hr.

5. Discussion: Licensing argument: they do not see any evidence for transient kinase activity.

The Reviewer is correct. We do not provide any data to support a role in licensing so we have removed this point from the Discussion.

6. Line 71: the argument seems to be "Cdh1 overexpression -> APC overactivation -> reduced glycolysis". How can we exclude that reduced glycolysis is not a consequence of cell cycle misregulation by an overactive APC/C? How strong is the overexpression of Cdh1?

One way we can control for APC/C having indirect effects on glycolysis through mis-regulation of the cell cycle is to inhibit a specific APC/C substrate while keeping APC/C activity intact. We provide this data in Fig. 4k (compare second and third bar) where we treated cells with the PFKBF3 inhibitor PFK15. PFKBF3 is the APC/C substrate directly linked to glycolysis. In Fig. 4k we show that by inhibiting PFKBF3 while keeping APC/C activity intact, we still see a reduction in glycolysis. This provides evidence that PFKBF3 is the APC/C substrate responsible for the reduction in glycolysis and not a general misregulation of the cell cycle.

We overexpressed Cdh1 to 1.5 fold compared to the endogenous Cdh1 (Extended Data Fig. 10a).

7. Line 90-92: measurements for other time points before and after 5 hrs are not presented, so it is difficult to tell how APC/C activity varies over time. Granted, microscopy observations provide strong evidence that APC/C is transiently inactivated, but the MG132 assay is more quantitative and is done at only one time point.

Thank you for the suggestion to look at additional timepoints. We agree that the MG132 assay provides an extra layer of support of our other data. We repeated this experiment by treating cells with MG132 at 1, 2, 3, 4 or 5 hours after mitogen stimulation and measured the slope of the increase in the APC/C degron levels (New Extended Data Fig. 2m). Because the proteasome is inhibited after treatment with MG132, the resulting rise in the APC/C degron levels is essentially a measure of the combined transcription and translation rates of the sensor. By measuring this transcription/translation rate at multiple timepoints, we can rule out the possibility that the transient accumulation of the APC/C degron sensor is due to changes in transcription/translation rate. Furthermore, we compared the underlying transcription/translation rate in the MG132 treated cells with the rate of accumulation of the APC/C degron sensor in the mitogen stimulated condition (Mit. Stim. + DMSO) (New Extended Data Fig. 2n). We found that the rate of transcription/translation of the sensor remains relatively constant between 1 to 5 hr after mitogen stimulation, which means that the change in the APC/C degron levels we observed strongly suggest changes in proteasomal-mediated degradation of the APC/C degron sensor and likely not due to changes in transcription and/or translation.

8. Line 135: the band in Fig.2g is very faint.

We have now replaced Fig. 2g with a new figure panel with darker bands.

9. 134-135: what does the binding of mTOR and Cdh1 demonstrate? If mTOR still binds the T129A mutant, doesn't this imply that it may phosphorylate another residue (or residues) as well? What about the T129D mutant? (is it also bound less by mTOR?)

To address both of these points at the same time, we have repeated the experiment shown in Fig. 2g using newly purified Cdh1-WT, Cdh1-T129A, and Cdh1-T129D expressed from baculovirus in insect cells. The Cdh1 expressed in insect cells is more stable than the Cdh1 expressed previously in bacteria and yields more consistent results. As shown in New Fig. 2g, mTOR binds Cdh1-WT and Cdh1-T129A, but binds less efficiently to Cdh1-T129D.

10. Line 146: band in blot shown in Fig.2j are too faint

We have repeated the experiment in Fig. 2j to also include the addition of mTOR and to also provide darker bands. The new blot is now Fig. 2k. We also added in quantification of the repeated experiments (Fig. 2l).

11. Line 193-194: the phrasing is a bit sloppy here, but is revealing on the way they think: fast inhibition followed by slow activation does not generate an incoherent feedforward (IFF), but can be generated by an incoherent feedforward. In other words, an IFF can produce what they observe, but what they observe is not necessarily produced by an IFF.

We apologize for the confusion and seek to clarify both here and in the text the use of the term incoherent feedforward system. The system is incoherent because the upstream regulators have both positive and negative effects on the APC/C. In other words, mTOR inhibits the APC/C and phosphatases activate the APC/C. The system contains feedforward regulation because upstream mitogen signaling affects downstream APC/C activity through more than one pathway¹⁰. So, we believe it is correct to describe this signaling architecture as an incoherent feedforward loop. As the reviewer rightly points out, such incoherent feedforward systems can produce pulsatile dynamics in the downstream protein being regulated, in our case the APC/C, which is precisely what we observe^{11,12}. It is possible these same dynamics could be produced by alternative signaling architectures. However, our biochemical, genetic, and drug-addition experiments have revealed an incoherent feedforward architecture linking mitogens to APC/C activity. We included the mathematical model in Fig. 3g, to demonstrate how the incoherent feedforward loop could generate the pulsatile APC/C activity that we observe. Taken all together, we believe the most likely explanation for the pulsatile dynamics of APC/C activity is due to the incoherent feedforward loop.

12. Line 196-198: the model is too phenomenological to be of any use. No reaction mechanisms are modelled, just a bunch of sigmoidal activation/inhibition terms. The meaning of several states is unclear (what is mTOR? APC? PP?). The authors don't describe how they came up with the parameters. On a more general level, it is hard to understand what the model contributes to the paper. Saying "we made a model and it reproduces what we observed" does not mean much. The model is supposed to reveal some aspect of the system that was not obvious from the data, or make a prediction that is validated by further experiments.

We agree with the reviewer that the model does not necessarily add anything new. We included it simply to accommodate a more general audience that may not understand how an incoherent feedforward loop could lead to a transient pulse of APC/C inactivation. The modeling simply provides a visual representation of how inhibition of the APC/C by mTOR followed by activation of the APC/C by a phosphatase can reproduce our experimentally measured transient inactivation of the APC/C. The critical parameters used in this model, the timing of when mTOR and the phosphatase are activated, were experimentally measured (Fig. 3e,f and Extended Data Fig. 6f).

We would certainly be okay with removing the model from the manuscript, but since we do not claim the model proves our data, we have left it in at this time.

13. Line 431: The authors refer to previously described methods without referring to any papers where a description of these methods can be found.

We apologize for this oversight. We have now added the required citation.

14. Why is Fig. 2k normalized to data in Ext. Fig 3k?

We apologize for this confusion, as our description in the figure legend was a bit vague. The data shown in what is now Fig. 2m is a quantification of single-cell data, which is shown in its raw

form in Extended Data Fig. 5n (note this was previously Ext. Fig 3k in the original submission). They are essentially the same experiment, just shown in two different forms (eg single-cell data, and bar graph form to show averages and statistics). We have edited the figure legend to be more clear.

15. Line 844: The “coordinating” aspect is not shown in this paper, and this is thus speculation.

We removed the text.

16. Fig. 5e: Typo in glycolysis

The typo has been fixed.

Referee #3 (Remarks to the Author):

In this manuscript, Paul et al tackle an apparent paradox in cell cycle regulation, whereby the ubiquitin ligase APC/C, which is necessary for cell cycle progression, also inhibits glycolysis, an obligate metabolic process for cells to exit quiescence and resume proliferation.

To reconcile these opposing actions, the authors report that cells briefly deactivate APC/C upon entering the cell cycle, allowing a shift to glycolysis. This occurs due to mTOR-triggered phosphorylation of the APC/C adapter protein Cdh1, leading to increased levels of the glycolytic enzyme PFKFB3. Later, phosphatase activity restores APC/C function, transitioning cells back to oxidative phosphorylation. This process coordinates cell cycle and metabolism through a temporary inhibition of APC/C, facilitating a burst of glycolysis.

Overall, this is a conceptually novel and interesting work that connects selective APC regulation during cell cycle re-entry to metabolism. The authors provide convincing evidence that PFKFB3 phosphorylation occurs in a Cdh1 phosphorylation-dependent manner. Moreover, they show that PFKFB3 stabilization due to mTORC1-dependent Cdh1 phosphorylation and APC inhibition results in increased intracellular ATP levels, possibly as a way to sustain resumption of cell cycle.

However, there are notable weaknesses that need to be addressed mechanistically. In particular, the data provided in support of mTORC1-dependent Cdh1 phosphorylation are weak and unconvincing. Also, whether phosphatase-mediated Cdh1 dephosphorylation represents a specific regulatory step, as opposed to baseline activity, is not clarified. Specific points follow.

We appreciate the reviewer's thorough review and valuable input. In the revised manuscript, we have incorporated additional experiments to support the discovery of mTOR-Cdh1 interaction both *in vivo* and *in vitro*. We acknowledge the potential limitations of using a pan phosphatase inhibitor and have performed several control and rescue experiments to bolster our conclusions and mitigate any concerns regarding off-target effects.

1- 2-DG is used to support the need for glycolysis in cell cycle re-entry. However, this compound has pleiotropic effects and can induce cell death. Orthogonal methods to test the requirement for heightened glycolysis should be tested, such as reducing glucose levels, switching cells to galactose, knocking down various enzymes along the glycolytic cascade and/or the main glucose transporters (I.e. GLUT1).

Yes, we agree with the Reviewer's point that 2DG can have pleiotropic effects. Although, 2DG is widely used as a glycolysis inhibitor, the suggestions made by the reviewer for orthogonal tests are well received. We have now used two orthogonal approaches to test the requirement for glycolysis, specifically by depleting glucose from the media and providing galactose as the primary carbon source (New Extended Data Fig. 1d-i). We found that depleting glucose or providing only galactose as the carbon source blocked mitogen-stimulated cell cycle entry in MCF-10A cells (New Extended Data Fig. 1e-g). Interestingly, we also blocked OXPHOS in MCF-10A cells as an additional control and found that cells could still activate CDK2 and enter the cell cycle, indicating that although cells undergo both glycolysis and OXPHOS during quiescence, cells are more reliant on glycolysis than OXPHOS for cell cycle entry (New Extended Data Fig. 1d). Further, inhibition of GLUT transporters also blocked cell cycle entry from quiescence (Extended data fig. 1f), indicating that glucose uptake and its downstream processing is necessary for cell cycle entry in mammalian cells. Finally, given that Reviewer 1 (Major Comment 2) also asked us to test for the requirement for glycolysis for cell cycle entry in additional cell lines we repeated these experiments with RPE1 and MEFs (New Extended Data Fig. 1h,i). Similar to MCF10A cells, we observed that depleting glucose or providing only galactose as the carbon source blocked mitogen-stimulated cell cycle entry as measured by Rb phosphorylation.

2- The direct interaction of Cdh1 with mTOR (Fig. 2d) is unconvincing. mTORC1 substrates do not stably interact with the kinase, so as to favor rapid catalysis. Also, empty IgG as the negative control is not appropriate. An unrelated [protein] should be used in the negative control lane. Same issue in Fig. 2e

To address the issue of using empty IgG as a negative control (also mentioned by Reviewer 1, minor point #9), we performed IP experiments using two different unrelated proteins. First, we pulled down with an EGFR antibody (new Extended Data Fig. 4b) and second, we pulled down with an HA antibody (new Extended Data Fig. 4c). In both cases, we did not observe mTOR or Cdh1 binding non-specifically to the negative control antibody-coated beads.

Additionally, we have used FLAG-DYRK4 as a control for an *in vitro* interaction (new Fig. 2e). We found that mTOR binds more specifically to Cdh1 than DYRK4. However, there is background binding present, which might be because of the sticky nature of the purified Cdh1 protein. Finally, we note that mTOR has been shown to immunoprecipitate with many of its substrates²⁻⁴. Therefore, it is possible that Cdh1 belongs to those cohort of substrates which immunoprecipitates with mTOR.

3- The kinase assay in fig. 2f is problematic. mTOR requires accessory subunits, typically Raptor, to recruit substrates to the kinase active site, both in cells and in vitro. This assay was carried out with the mTOR kinase domain alone, which is not evidence of specific phosphorylation, but merely a sign that, with enough kinase and protein mixed together, some phosphate transfer can occur.

We apologize for this confusion. In the kinase assay shown in Fig. 2f (now Extended Data Fig. 4e), there is both mTOR and mLST8 present. We have amended the labeling of this figure panel to more clearly point out the inclusion of mLST8 in the reaction. The mTOR-mLST8 complex has been previously shown to be active and is able to phosphorylate bona fide substrates¹³. However, to better directly address this important point raised by the reviewer, we have also repeated the *in vitro* kinase assay including Raptor into the reaction. Consistent with our

previous results, we observed mTOR-dependent phosphorylation of Cdh1 (new Fig. 2f). Further, we have also tested another kinase DYRK4 and found no threonine phosphorylation of Cdh1. This suggests that mixing enough random kinase with Cdh1 does not lead to its phosphorylation (new Extended Data Fig. 4g,i).

4- Do other kinases (e.g. CDK1, ERK) fail to in vitro phosphorylate Cdh1 when used under identical conditions as in 2f?

We thank the reviewer for this suggestion as it is a good control for the specificity of Cdh1 phosphorylation. ERK and CDK1 have actually been previously shown to phosphorylate Cdh1^{14,15}. When we performed in vitro kinase assays with either ERK or CDK1 and Cdh1, we found that they could phosphorylate wild-type Cdh1 as well as Cdh1-T129A and Cdh1-T129D mutants, consistent with ERK phosphorylating residues other than T129 on Cdh1 (new Extended Data Fig. 4f) and presumably CDK1 phosphorylating the known CDK2 sites. (new Extended Data Fig. 4g,h). Thus, the two known kinases are able to phosphorylate WT and well as T129A or T129D Cdh1, indicating that they do not overlap with mTOR targeted sites.

5- Fig. 2h: the decreased interaction of Cdh1 S129A mutant with mTOR is also puzzling: the recruitment motif and phosphorylation motif on virtually all mTORC1 substrates are separate, in other words, substrates are not recruited to mTORC1 via the phospho-site but via the TOS motif (PMID: 12747827). Does Cdh1 harbor a putative TOS-like motif, and would its mutation decrease interaction with mTORC1?

We share the concern of the reviewer. The in-silico analysis of Cdh1 sequence does not show a TOS-like motif (FxxEDz; where x is any a.a and z is a hydrophobic one). We have also looked for other motifs, like RAIP and NPXY. Considering the existence of multiple mTOR substrates lacking TOS or similar motifs, we propose that Cdh1 may be recognized by mTOR in a TOS-independent manner. Previous research has demonstrated mTOR's ability to interact with and phosphorylate its substrates in a TOS-independent manner, suggesting that Cdh1 could be one such substrate. Here, we cite some well recognized articles showing TOS independent interaction of mTOR and its substrates^{4,16-18}.

Additionally, we have repeated the experiment shown in Fig. 2g using newly purified Cdh1-WT, Cdh1-T129A, and Cdh1-T129D expressed from baculovirus-infected insect cells. The Cdh1 expressed in insect cells is more stable than the Cdh1 expressed previously in bacteria and yields more consistent results. As shown in new Fig. 2g, mTOR binds Cdh1-WT and Cdh1-T129A, but binds less efficiently to Cdh1-T129D, indicating that the new data aligns with the Reviewer's expectation.

6- The authors provide some (relatively weak) data suggesting that mTOR-dependent phosphorylation of S129 decreases Cdh1 interaction with the APC core (e.g. ED 3i). Given that the APC-Cdh1 interaction has been dissected using both biochemical and structural approaches, it is important that the authors provide some mechanistic rationale as to how S129 phosphorylation (which occurs in a predicted unstructured region of Cdh1) would interfere with binding of the ordered WD40 domain to the APC core.

We now provide a clearer rationale for why phosphorylation of T129 can lead to partial disruption of the Cdh1-APC/C interaction. As shown in new Extended Data Fig. 5m, the unstructured region of Cdh1 is phosphorylated by a number of other kinases, most notably

CDK2. Many previous studies have shown that phosphorylation of this unstructured region by CDK2 disrupts the Cdh1-APC/C interaction, which is one of the mechanisms by which CDK2 inactivates the APC/C at the G1/S transition¹⁹⁻²². These previous findings are also the main reason we investigated this mechanism in the first place for mTOR-mediated phosphorylation of Cdh1, given the close proximity of T129 with these other CDK2 phosphorylation sites (S40, T121, S151, and S163). Structural studies have provided the mechanistic rationale for why phosphorylation of this N-terminal unstructured region would interfere with the overall binding of Cdh1 with the rest of the APC/C. Briefly, segments of the N-terminal domain of Cdh1 interact with the APC/C subunits Apc8 and Apc1. Phosphorylation of these residues would destabilize the interaction between APC/C and Cdh1 through electrostatic repulsion and steric hinderance. Notably, in the case of CDK2 phosphorylation of Cdh1, Chang et al found that phosphorylation of individual residues each caused a partial disruption of the Cdh1-APC/C interaction whereas phosphorylation of all 4 residues caused a complete disruption²³. This is consistent with our observations that phosphorylation of T129 caused a partial disruption of the Cdh1-APC/C interaction.

7- If mTORC1 is the kinase that directly phosphorylates Cdh1, this process should be enhanced in cells deleted for negative mTORC1 regulators such as Tuberous Sclerosis Complex 1/2 or GATOR1. Does deletion of either of these complexes lead to constitutive Cdh1 phosphorylation, APC inactivation and more efficient cell exit from quiescence? Do these deletions phenocopy expression of the phospho-mimetic S129D mutant of Cdh1?

We conducted knockdown experiments targeting the mTOR negative regulators TSC1 or NPRL2 and evaluated the phosphorylation status of Cdh1. Our results showed an increase in pan phospho-threonine phosphorylation of Cdh1 in quiescent cells following knockdown of these regulators (Rebuttal Fig. R3a,b). However, in the presence of mitogens, we observed no significant change in Cdh1 phosphorylation, suggesting that mTOR might be fully activated upon mitogen stimulation and knocking down TSC or GATOR complex proteins does not further enhance its activity significantly.

Furthermore, to test if the deletion of the negative regulators leads to a phenotype similar to Cdh1-T129D, we tracked the APC/C degnon in live cells as they enter cell cycle (Rebuttal Fig. R1a-c). We found that deletion of TSC1 or NPRL2 leads to the transient accumulation of the APC/C degnon sensor and it phenocopies the ectopic expression of Cdh1-T129D phenotype. We do not observe a sustained accumulation of the APC/C degnon because of the presence of endogenous Cdh1.

Rebuttal Fig. R1

a, MCF-10A cells were starved for 48 hr to induce quiescence, followed by transfection with the indicated siRNAs. Mitogens were added at 0 hr to promote cell cycle entry. Single-cell traces of APC/C biosensor levels were measured. N=200 cells for each condition. The large increase in APC/C biosensor levels are an indication of S-phase entry. **b**, Immunoblot validation of NPRL2 knockdown using siRNA. **c**, Immunoblot validation of TSC1 knockdown using siRNA.

8- The significance of the protein phosphatase data is unclear. The authors imply that a specific protein phosphatase becomes activated at 4h post-induction, but such phosphatase is not identified. It also remains unclear whether a specific activation mechanism is involved, versus 'baseline' phosphatase activity eventually dephosphorylating Cdh1 in a non-regulated manner.

Based on this comment as well as comments made by Reviewers 1 and 2, we have attempted to further investigate the role of the phosphatase in regulating transient APC/C inactivation.

First, to address the issue that the phenotypes we observed may be due to global phosphorylation changes, as well as the comment made by Reviewer 1 (main point #3) and Reviewer 2 (main point #4), we have now performed a rescue experiment with the Cdh1-T129A mutant. The rationale behind this experiment is that if the increased accumulation of the APC/C degnon sensor is due to large changes in global protein phosphorylation status, then expressing Cdh1-T129A should not be able to dampen this effect. We found that ectopic expression of Cdh1-T129A can rescue the protein phosphatase mediated sustained accumulation of the APC/C degnon (new Extended data Fig. 6d), suggesting that the increased accumulation of the APC/C degnon is not due to a change in global protein phosphorylation, rather dephosphorylation of APC/C Cdh1.

Second, to assess whether the phosphatase may be functioning at baseline levels and is unregulated we have assessed the effect of PPase inhibitors on APC/C activity at different time points (new Extended data Fig. 6f). Despite adding the PPase inhibitor either simultaneously or as early as 1 hour after the addition of mitogens, the slope of the APC/C degron level accumulation does not deviate from the untreated control until 4 hr after mitogen addition. Conversely, adding the PPase inhibitors at time points later than 4 hr led to a rapid accumulation of the APC/C degron levels, indicating that the phosphatase is likely not active during the first 4 hours after mitogen addition and then is activated after 4 hours. We attempted to identify the phosphatase involved in the hopes of further investigating the mechanism of this delay by doing a large-scale siRNA screen using siRNA pools targeting all known human phosphatases. However, this screen did not identify any one individual phosphatase. We speculate this is likely due to the redundant nature of many phosphatases and more in depth studies that go beyond the scope of this study will be required to identify the phosphatases involved.

Third, to assess whether the phosphatase may be functioning during G0, we treated serum starved cells with the phosphatase inhibitor and performed live-cell imaging of the APC/C degron biosensor. Notably, we did not observe accumulation of the APC/C degron biosensor, indicating the phosphatase is likely not regulating APC/C activity during G0 (new Extended data Fig. 6e).

We believe the addition of these new experiments provide important clarity to the role of phosphatases in regulating APC/C re-activation during exit from quiescence. However, we acknowledge that additional follow-up studies are needed to further investigate the precise mechanism underlying this regulation.

Literature Cited

- 1 Battaglioni, S., Benjamin, D., Walchli, M., Maier, T. & Hall, M. N. mTOR substrate phosphorylation in growth control. *Cell* **185**, 1814-1836 (2022). <https://doi.org:10.1016/j.cell.2022.04.013>
- 2 Hoxhaj, G. *et al.* The E3 ubiquitin ligase ZNRF2 is a substrate of mTORC1 and regulates its activation by amino acids. *eLife* **5** (2016). <https://doi.org:10.7554/eLife.12278>
- 3 Thedieck, K. *et al.* PRAS40 and PRR5-like protein are new mTOR interactors that regulate apoptosis. *PloS one* **2**, e1217 (2007). <https://doi.org:10.1371/journal.pone.0001217>
- 4 Yu, Y. *et al.* Phosphoproteomic analysis identifies Grb10 as an mTORC1 substrate that negatively regulates insulin signaling. *Science* **332**, 1322-1326 (2011). <https://doi.org:10.1126/science.1199484>
- 5 Sparta, B., Kosaisawe, N., Pargett, M., Patankar, M., DeCuzzi, N. & Albeck, J. G. Continuous sensing of nutrients and growth factors by the mTORC1-TFEB axis. *eLife* **12** (2023). <https://doi.org:10.7554/eLife.74903>
- 6 Kim, H. G., Jeong, S. G., Kim, J. H. & Cho, J. Y. Phosphatase inhibition by sodium orthovanadate displays anti-inflammatory action by suppressing AKT-IKKbeta signaling in RAW264.7 cells. *Toxicol Rep* **9**, 1883-1893 (2022). <https://doi.org:10.1016/j.toxrep.2022.09.012>

- 7 Morioka, M. *et al.* Serine/threonine phosphatase activity of calcineurin is inhibited by sodium orthovanadate and dithiothreitol reverses the inhibitory effect. *Biochem Biophys Res Commun* **253**, 342-345 (1998). <https://doi.org:10.1006/bbrc.1998.9783>
- 8 Hung, Y. P. *et al.* Akt regulation of glycolysis mediates bioenergetic stability in epithelial cells. *eLife* **6** (2017). <https://doi.org:10.7554/eLife.27293>
- 9 Koberstein, J. N. *et al.* Monitoring glycolytic dynamics in single cells using a fluorescent biosensor for fructose 1,6-bisphosphate. *Proc Natl Acad Sci U S A* **119**, e2204407119 (2022). <https://doi.org:10.1073/pnas.2204407119>
- 10 Ferrell, J. E. *Systems Biology of Cell Signaling: Recurring Themes and Quantitative Models.* (CRC Press, 2021).
- 11 Goentoro, L., Shoval, O., Kirschner, M. W. & Alon, U. The incoherent feedforward loop can provide fold-change detection in gene regulation. *Mol Cell* **36**, 894-899 (2009). <https://doi.org:10.1016/j.molcel.2009.11.018>
- 12 Ferrell, J. E., Jr. Perfect and Near-Perfect Adaptation in Cell Signaling. *Cell Syst* **2**, 62-67 (2016). <https://doi.org:10.1016/j.cels.2016.02.006>
- 13 Kang, S. A. *et al.* mTORC1 phosphorylation sites encode their sensitivity to starvation and rapamycin. *Science* **341**, 1236566 (2013). <https://doi.org:10.1126/science.1236566>
- 14 Wan, L. *et al.* The APC/C E3 Ligase Complex Activator FZR1 Restricts BRAF Oncogenic Function. *Cancer Discov* **7**, 424-441 (2017). <https://doi.org:10.1158/2159-8290.CD-16-0647>
- 15 Listovsky, T., Zor, A., Laronne, A. & Brandeis, M. Cdk1 is essential for mammalian cyclosome/APC regulation. *Exp Cell Res* **255**, 184-191 (2000). <https://doi.org:10.1006/excr.1999.4788>
- 16 Duan, S. *et al.* mTOR generates an auto-amplification loop by triggering the betaTrCP- and CK1alpha-dependent degradation of DEPTOR. *Mol Cell* **44**, 317-324 (2011). <https://doi.org:10.1016/j.molcel.2011.09.005>
- 17 Hsu, P. P. *et al.* The mTOR-regulated phosphoproteome reveals a mechanism of mTORC1-mediated inhibition of growth factor signaling. *Science* **332**, 1317-1322 (2011). <https://doi.org:10.1126/science.1199498>
- 18 Nazio, F. *et al.* mTOR inhibits autophagy by controlling ULK1 ubiquitylation, self-association and function through AMBRA1 and TRAF6. *Nat Cell Biol* **15**, 406-416 (2013). <https://doi.org:10.1038/ncb2708>
- 19 Jaspersen, S. L., Charles, J. F. & Morgan, D. O. Inhibitory phosphorylation of the APC regulator Hct1 is controlled by the kinase Cdc28 and the phosphatase Cdc14. *Curr Biol* **9**, 227-236 (1999). [https://doi.org:10.1016/s0960-9822\(99\)80111-0](https://doi.org:10.1016/s0960-9822(99)80111-0)
- 20 Kramer, E. R., Scheuringer, N., Podtelejnikov, A. V., Mann, M. & Peters, J. M. Mitotic regulation of the APC activator proteins CDC20 and CDH1. *Mol Biol Cell* **11**, 1555-1569 (2000). <https://doi.org:10.1091/mbc.11.5.1555>
- 21 Zachariae, W., Schwab, M., Nasmyth, K. & Seufert, W. Control of cyclin ubiquitination by CDK-regulated binding of Hct1 to the anaphase promoting complex. *Science* **282**, 1721-1724 (1998). <https://doi.org:10.1126/science.282.5394.1721>
- 22 Lukas, C. *et al.* Accumulation of cyclin B1 requires E2F and cyclin-A-dependent rearrangement of the anaphase-promoting complex. *Nature* **401**, 815-818 (1999). <https://doi.org:10.1038/44611>

- 23 Chang, L., Zhang, Z., Yang, J., McLaughlin, S. H. & Barford, D. Atomic structure of the APC/C and its mechanism of protein ubiquitination. *Nature* **522**, 450-454 (2015).
<https://doi.org:10.1038/nature14471>

General response to all reviewers:

We thank all the reviewers for taking the time to review our manuscript. By incorporating your suggestions, we believe the manuscript is now much strengthened. Below you will find a point-by-point response to each of the suggestions that were raised. Your comments are shown in *italics* and our responses are shown in blue.

Referee #1

The revised manuscript by Paul et al. has been significantly strengthened through extensive revisions and the addition of new data, effectively addressing previous major concerns. The authors have provided substantial evidence supporting their model of mTOR-mediated transient APC/C inactivation as necessary for cell cycle entry. This research offers crucial insights into the coordination of cellular metabolism and cell cycle regulation. I am convinced that the manuscript is now poised for publication, pending the minor but crucial adjustments described below, to ensure clarity and precision in the presentation and interpretation of the data. I anticipate that it will be well-received by the general readership of Nature.

We would like to thank the reviewer for such encouraging comments and for taking the time to thoroughly and thoughtfully review our manuscript. Addressing these comments has helped us strengthen the manuscript.

1. While it is commendable that quantification and statistical analyses are now provided for most data, separating the representative data and the corresponding quantifications into main and supplementary data is not ideal. Presenting them adjacent to each other in the same figures would facilitate better understanding for readers and is therefore recommended.

We thank the reviewer for making this point. We have now edited the figures to include the quantification next to the relevant blot. We agree that flipping back and forth between the main figures and the supplement is onerous for the reader and providing them side-by-side will certainly facilitate better understanding.

2. In lines 20-22 in the summary, “Simultaneously, cells activate APC/C.” is confusing, as this can be interpreted as APC/C being inactive in quiescent (G0) cells and becoming activated upon cell cycle re-entry, prior to the G1/S transition. However, as the authors wrote in the main text (lines 46-52), APC/C-CDH1 is already active in G0 cells and remains active until the G1/S transition. The authors should revise these sentences to avoid confusion. Similarly, the meaning of the sentence in lines 55-56, “Similarly, if the APC/C is inactivated...” is unclear and should be clarified.

We thank you for pointing out this confusion. Indeed, as written the sentence does incorrectly imply that the APC/C is inactive during G0 and initially is activated. We have now edited this sentence to more accurately reflect the correct biology:

Lines 20-22: “Simultaneously, the ubiquitin ligase Anaphase-Promoting Complex/Cyclosome (APC/C)-Cdh1 remains active to allow origin licensing and block premature DNA replication.”

Lines 55-56: “Similarly, if the APC/C is prematurely inactivated upon exit from quiescence, then how can cells license origins of replication ahead of S phase?”

3. In Figures 1a, 1c, 1j, 3a, and 3h, the placement of “Mitogens” in the diagrams does not seem appropriate. It would be more consistent with the final model in Figure 5e to position it above Glycolysis and APC/C-CDH1 or next to G0 before Glycolysis.

We have addressed this by moving the “Mitogens” label and the arrow more to the left so that it is before Glycolysis. Thank you for catching this error.

4. Regarding minor point 5, the calculation of “33%” is still not entirely clear from the explanation in the text (lines 104-107) and Figure 1h, i. The value on the x-axis in Figure 1h, labeled as “slope of APC/C degron rise (au/hr)” spanning from 0 to 100, is unclear and requires further explanation. The x-axis in Figure 1i should perhaps span from 0 to 100.

We have now edited the text in lines 98-101 to hopefully better clarify the 33% calculations. The reviewer is correct that the x-axis in Figure 1i should span from 0 to 100. We originally plotted the data as “fraction” but labeled it as “%” in error. We certainly understand how this error contributed to the confusion. In addition to correcting the x-axis label, we have included a simple mathematical equation that shows how we went from the numbers in Fig. 1h to the numbers in Fig. 1i. We hope the reviewer finds these changes help clarify the issue.

5. In Extended Data Fig. 5a, more description is needed to interpret the data in the figure and in the legend. For instance, which peak corresponds to which modified or unmodified peptides?

A comprehensive description of the spectra interpretation has now been included in the Materials and Methods section. Also, additional lines have been added to the figure legend to explain the color code and peak modifications.

6. In Extended Data Fig. 5h, $n=3$ should be $n=4$.

We have changed the figure legend to reflect the $n=4$. Thank you for catching this error.

7. The sentence in lines 180-185 is excessively long and should be revised for clarity.

We have now edited this sentence to shorten it and make it more readable.

8. In lines 196-197 and Extended Data Fig. 6a, b, while the $pmTOR/mTOR$ ratio may not be, the phosphorylated $mTOR$ level (therefore, active $mTOR$), as well as $p4EBP1$ level, is clearly transiently increased around 3-7 hours after mitogen stimulation. The potential regulation of $mTOR$ activity, in cooperation with subsequent PPase activation in the control of APC/C-CDH1 activity, cannot be excluded. The text should be amended accordingly.

The reviewer raises a good point. It is likely that a combination of PPase activity and $mTOR$ regulation underlies the APC/C-CDH1 dynamics we observed. While we observed a strong contribution of PPase activity involved in these dynamics, we cannot rule out $mTOR$ regulation is also contributing given the presented data.

To address this comment, we have edited the text in lines 198-202:

“We assessed the ratio of phospho-mTOR/mTOR to quantify the mTOR activity and found a slight increase in phospho-mTOR/mTOR ratio between 3 and 7 hours after mitogen stimulation during the time when APC/C transiently reactivates (3-5 hr after mitogen stimulation) (Extended Data Fig. 6a,b), suggesting that regulation of mTOR could in part underlie the regulation of transient APC/C inactivation.”

9. In Figure 3e, mTOR inhibition after mitogen stimulation appears to inhibit APC/C inactivation in a graded manner, according to the timing of rapamycin addition, rather than a flipped switch manner (The partial APC/C inactivation observed in the 1-hour mTOR inhibitor addition condition suggests a more gradual effect). As Reviewer #2 mentioned, biological phenomena are hardly black and white. The authors should remove “completely” in line 213.

We agree with the reviewer that biological phenomenon is hardly black and white, and therefore we have removed “completely” from the sentence in line 219.

10. Fig. 5a and Extended Data Fig. 10c appear to be equivalent. Consider removing the redundancy.

Fig. 5a and Extended Data Fig. 10c are slightly different. Fig. 5a is ectopic over expression while Extended Data Fig. 10c is with a doxycycline-induced expression followed by a wash. The purpose of the dox-inducible experiment is to determine whether the cell cycle defects we have observed are due to expression of Cdh1 mutants during the G0/G1 transition or whether they could be due to the Cdh1 mutants having effects at later stages of the cell cycle. By washing off the doxycycline and halting the expression of the Cdh1 mutants, we can support the first scenario, that the Cdh1 mutants are primarily affecting the G0/G1 transition.

Referee #2

Points 1-4 have been addressed satisfactorily.

Point 5, glycolytic rate, has not been answered properly, as the methodologies employed (whether ATP concentration measurements or fluorescent biosensors) do not represent the gold standard in the field.

Metabolic rate is most accurately measured using mass spectrometry (metabolomics) with isotopically labeled tracers, as described here: <https://www.sciencedirect.com/science/article/pii/S0092867418303878> and in many other publications.

Many metabolic experiments were carried out in previously quiescent cells that were simply stimulated with mitogens (i.e. fig.4g-i). These experiments tell very little, as it is a well-known and expected fact that mitogens will increase glycolytic rate via a myriad of mechanisms, such as AKT-stimulated glucose uptake. The key experiment here is to show that changing Cdh1 phosphorylation status has noticeable effects on these parameters, as it seems to do for ATP levels and glycoPER (extended data Fig 9h-i).

To summarize, glycolytic metabolite flux analysis with appropriate isotopically labeled tracers should be carried out in cells in which the phosphorylation status of Cdh1-T129 has been manipulated through mutagenesis.

We agree with the reviewer that tracing glycolytic metabolites in cells with altered Cdh1-T129 status is considered a gold standard approach in the field. To further validate our observations, we performed ¹³C₆-labeled glucose tracing experiments in cells entering the cell cycle, in the presence or absence of the Cdh1-T129A mutant, which cannot be phosphorylated by mTOR (see new Fig. 4l). Metabolomic analysis revealed that expression of Cdh1-T129A led to a reduction in upper/early glycolytic intermediates (G6P, F6P, FBP), along with a moderate increase in lower glycolytic metabolites (DHAP, 3PG, PEP). These changes suggest a reduced glycolytic flux and a temporary compartmentalization of the glycolytic pathway, in line with previous findings¹⁻³.

A second point is to more strongly link mTOR-dependent APC regulation to PFKFB3 degradation and to downstream alteration in glycolytic rate. A key hypothesis is that transient stabilization of PFKFB3 is what drives the spike in glycolysis. If so, expressing a PFKFB3 protein mutated in the APC degron should blunt this effect, as the basal (non-mitogen stimulated) glycolytic rate should be higher than in control cells (and, perhaps, these cells may be less quiescent).

We agree with the reviewer that testing the APC degron mutated PFKFB3 would further link APC regulation to PFKFB3 degradation and downstream alteration in glycolytic rate. To address this concern, we have used PFKFB3^{KEN} mutant, which is minimally affected by Cdh1-T129A (Extended Data Fig. 8h), and performed glycolytic rate measurements in ectopic presence of Cdh1-T129A (Extended Data Fig. 8i). Our experiments demonstrated that expression of the PFKFB3^{KEN} mutant mitigated the glycolytic suppression typically induced by CDH1-T129A (Extended Data Fig. 9j). These findings suggest that the degradation of PFKFB3 by APC/C-Cdh1 plays a considerable role in modulating glycolytic rates.

This Reviewer is aware that these requests impose significant extra work on the already large body of experiments provided by the authors. However, given the provocative and potentially highly impactful model they put forward, which in the authors' mind would supersede the myriad other ways through which mitogens can link cell proliferation and metabolism, it is essential that the key statements are thoroughly tested.

We thank the Reviewer for providing these suggestions. We agree that the added experiments provide further corroboration of our model and certainly welcome any suggestions that help us strengthen our study.

Referee #3

The authors have addressed the concerns raised in the initial submission. In particular, the connection between mTOR-dependent Cdh1 phosphorylation and APC inhibition has been strengthened. I have no further comments.

Thank you very much for taking the time to provide feedback that has strengthened the manuscript.

Literature Cited

- 1 Xu, Y. F. *et al.* Regulation of yeast pyruvate kinase by ultrasensitive allostery independent of phosphorylation. *Mol Cell* **48**, 52-62 (2012).
<https://doi.org:10.1016/j.molcel.2012.07.013>
- 2 Lowry, O. H., Carter, J., Ward, J. B. & Glaser, L. The effect of carbon and nitrogen sources on the level of metabolic intermediates in *Escherichia coli*. *J Biol Chem* **246**, 6511-6521 (1971).
- 3 Jang, C., Chen, L. & Rabinowitz, J. D. Metabolomics and Isotope Tracing. *Cell* **173**, 822-837 (2018). <https://doi.org:10.1016/j.cell.2018.03.055>

Referees' comments:

Referee #3 (Remarks to the Author):

The authors have satisfactorily addressed my remaining concerns. I recommend acceptance of this interesting and innovative manuscript.

Thank you very much for taking the time to provide feedback. Incorporating your suggestions has strengthened the manuscript.